# *Scn1a*-GFP transgenic mouse revealed Nav1.1 expression in neocortical pyramidal tract projection neurons

**Tetsushi Yamagata[1,2†], Ikuo Ogiwara[2,3†], Tetsuya Tatsukawa[2†], Toshimitsu Suzuki[1,2], Yuka Otsuka[1], Nao Imaeda[1], Emi Mazaki[2‡], Ikuyo Inoue[2§], Natsuko Tokonami[2#], Yurina Hibi[1], Shigeyoshi Itohara[4], Kazuhiro Yamakawa[1,2]\***

[1]Department of Neurodevelopmental Disorder Genetics, Institute of Brain Science, Nagoya City University Graduate School of Medical Sciences, Nagoya, Japan; [2]Laboratory for Neurogenetics, RIKEN Center for Brain Science, Wako, Japan; [3]Department of Physiology, Nippon Medical School, Tokyo, Japan; [4]Laboratory for Behavioral Genetics, RIKEN Center for Brain Science, Wako, Japan

**\*For correspondence:**
yamakawa@med.nagoya-cu.ac.jp

[†]These authors contributed equally to this work

**Present address:** [‡]International Research Center for Neurointelligence (IRCN), The University of Tokyo, Institutes for Advanced Study, Tokyo, Japan; [§]Medical-risk Avoidance based on iPS Cells Team, RIKEN Center for Advanced Intelligence Project, Kyoto, Japan; [#]Institute of Physiology, University of Zürich, Zürich, Switzerland

**Competing interest:** The authors declare that no competing interests exist.

**Abstract** Expressions of voltage-gated sodium channels Nav1.1 and Nav1.2, encoded by *SCN1A* and *SCN2A* genes, respectively, have been reported to be mutually exclusive in most brain regions. In juvenile and adult neocortex, Nav1.1 is predominantly expressed in inhibitory neurons while Nav1.2 is in excitatory neurons. Although a distinct subpopulation of layer V (L5) neocortical excitatory neurons were also reported to express Nav1.1, their nature has been uncharacterized. In hippocampus, Nav1.1 has been proposed to be expressed only in inhibitory neurons. By using newly generated transgenic mouse lines expressing *Scn1a* promoter-driven green fluorescent protein (GFP), here we confirm the mutually exclusive expressions of Nav1.1 and Nav1.2 and the absence of Nav1.1 in hippocampal excitatory neurons. We also show that Nav1.1 is expressed in inhibitory and a subpopulation of excitatory neurons not only in L5 but all layers of neocortex. By using neocortical excitatory projection neuron markers including FEZF2 for L5 pyramidal tract (PT) and TBR1 for layer VI (L6) cortico-thalamic (CT) projection neurons, we further show that most L5 PT neurons and a minor subpopulation of layer II/III (L2/3) cortico-cortical (CC) neurons express Nav1.1 while the majority of L6 CT, L5/6 cortico-striatal (CS), and L2/3 CC neurons express Nav1.2. These observations now contribute to the elucidation of pathological neural circuits for diseases such as epilepsies and neurodevelopmental disorders caused by *SCN1A* and *SCN2A* mutations.

## Editor's evaluation

Using a newly developed Scn1a promoter driven GFP mouse line, the authors convincingly show that GFP expression largely replicates the endogenous expression of Nav1.1. Additionally, they credibly identify inhibitory and excitatory neurons in the cortex that express Nav 1.1. This mouse line provides a valuable resource, especially for epilepsy and autism research, as it offers a reliable tool that can be used to identify specific cell populations that potentially cause disease-related symptoms such as seizures, ataxia, sociability deficits, learning and memory problems, and sudden unexpected death in epilepsy.

## Introduction

Voltage-gated sodium channels (VGSCs) play crucial roles in the generation and propagation of action potentials, contributing to excitability and information processing (*Catterall, 2012*). They consist

of one main pore-forming alpha- and one or two subsidiary beta-subunits that regulate kinetics or subcellular trafficking of the alpha subunits. Human has nine alpha (Nav1.1–Nav1.9) and four beta (beta-1–beta-4) subunits. Among alphas, Nav1.1, Nav1.2, Nav1.3, and Nav1.6, encoded by *SCN1A*, *SCN2A*, *SCN3A*, and *SCN8A*, respectively, are expressed in central nervous system. *SCN3A* is mainly expressed embryonically (*Brysch et al., 1991*), and *SCN1A*, *SCN2A*, and *SCN8A* are major alphas after birth. Although these three genes show mutations in a wide spectrum of neurological diseases such as epilepsy, autism spectrum disorder (ASD), and intellectual disability, two of those, *SCN1A* and *SCN2A*, are major ones (reviewed in *Yamakawa, 2016*; *Meisler et al., 2021*). To understand the circuit basis of these diseases, it is indispensable to know the detailed distributions of these molecules in the brain.

We previously reported that expressions of Nav1.1 and Nav1.2 seem to be mutually exclusive in many brain regions (*Yamagata et al., 2017*). In adult neocortex and hippocampus, Nav1.1 is dominantly expressed in medial ganglionic eminence-derived parvalbumin-positive (PV-IN) and somatostatin-positive (SST-IN) inhibitory neurons (*Ogiwara et al., 2007*; *Lorincz and Nusser, 2008*; *Ogiwara et al., 2013*; *Li et al., 2014*; *Tai et al., 2014*; *Tian et al., 2014*; *Yamagata et al., 2017*). In the neocortex, some amount of Nav1.1 is also expressed in a distinct subset of layer V (L5) excitatory neurons (*Ogiwara et al., 2013*), but their natures were unknown. In the hippocampus, Nav1.1 seems to be expressed in inhibitory but not in excitatory neurons (*Ogiwara et al., 2007*; *Ogiwara et al., 2013*). In contrast, a major amount of Nav1.2 (~95%) is expressed in excitatory neurons including the most of neocortical and all of hippocampal ones, and a minor amount is expressed in caudal ganglionic eminence-derived inhibitory neurons such as vasoactive intestinal polypeptide (VIP)-positive ones (*Lorincz and Nusser, 2010*; *Yamagata et al., 2017*; *Ogiwara et al., 2018*). However, a recent study reported that a subpopulation (more than half) of VIP-positive inhibitory neurons is Nav1.1-positive (*Goff and Goldberg, 2019*).

VGSCs are mainly localized at axons and therefore it is not always easy to identify their origins, the soma. To overcome this, here in this study we generated bacterial artificial chromosome (BAC) transgenic mouse lines that express GFP under the control of *Scn1a* promoters, and we carefully

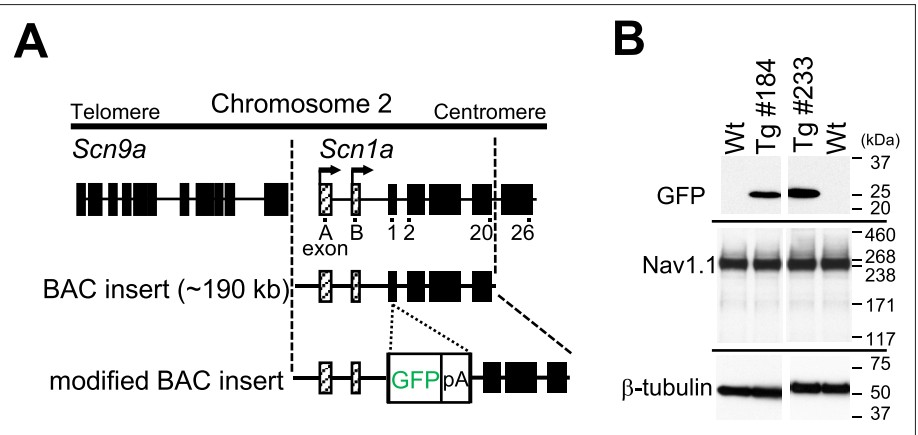

**Figure 1.** Generation of *Scn1a*-GFP mice. (**A**) Schematic representation of the modified bacterial artificial chromosome (BAC) construct containing the *Scn1a*-GFP transgene. A green fluorescent protein (GFP) reporter cassette consisting of GFP cDNA and a polyadenylation signal was inserted at the ATG initiation codon in the coding exon 1 of *Scn1a*. Filled and hatched boxes indicate the coding and non-coding exons of *Scn9a* and *Scn1a*. Arrows indicate the start sites and orientation of transcription of *Scn1a*. (**B**) Western blot analysis for *Scn1a*-GFP and endogenous Nav1.1. The whole cytosolic fractions from 5W *Scn1a*-GFP brains (lines #184 and #233) were probed with anti-GFP and their membrane fractions were probed with anti-Nav1.1 antibodies. β-Tubulin was used as an internal control. pA, polyadenylation signal; Tg, hemizygous *Scn1a*-GFP transgenic mice; Wt, wild-type littermates.

The online version of this article includes the following source data and figure supplement(s) for figure 1:

**Source data 1.** Raw and annotated immunoblots for *Figure 1B*.

**Figure supplement 1.** Green fluorescent protein (GFP) expression in the brain of two *Scn1a*-GFP transgenic mouse lines.

investigated the GFP/Nav1.1 distribution in mouse brain. Our analysis confirmed that expressions of Nav1.1 and Nav1.2 are mutually exclusive and that in neocortex Nav1.1 is expressed in both inhibitory and excitatory neurons while in hippocampus only in inhibitory but totally absent in excitatory neurons. Furthermore by using a transcription factor FEZF2 (FEZ family zinc finger protein 2 transcriptional factor), also referred to as Fezl, Fez1, Zfp312, and Fez, as a marker for L5 pyramidal tract (PT) neurons (*Inoue et al., 2004*; *Chen et al., 2005*; *Chen et al., 2008*; *Molyneaux et al., 2005*; *Lodato et al., 2014*; *Matho et al., 2021*), and a transcription factor TBR1 which suppresses FEZF2 expression and therefore does not overlap with FEZF2 (*Han et al., 2011*; *McKenna et al., 2011*; *Matho et al., 2021*), we found that most of L5 FEZF2-positive neurons are GFP-positive while L5/6 TBR1-positive neurons are largely GFP-negative and Nav1.2-positive. These results proposed that Nav1.1 is expressed in L5 PT while Nav1.2 in L5/6 non-PT neurons such as L5/6 cortico-striatal (CS) and L6 cortico-thalamic (CT) projection neurons. A majority of L2/3 excitatory neurons express Nav1.2 but a minor subpopulation are GFP-positive, suggesting that most of cortico-cortical (CC) projection neurons express Nav1.2 but the distinct minor population express Nav1.1. These results refine the expression loci of Nav1.1 and Nav1.2 in the brain and should contribute to the understanding of circuit mechanisms for diseases caused by *SCN1A* and *SCN2A* mutations.

## Results

### Generation and verification of *Scn1a*-GFP transgenic mouse lines

*Scn1a*-GFP founder mice were generated from C57BL/6J zygotes microinjected with a modified *Scn1a*-GFP BAC construct harboring all, upstream and downstream, *Scn1a* promoters (*Nakayama et al., 2010*; *Figure 1A*) (see Materials and methods for details). Western blot analysis (*Figure 1B*) and immunohistochemistry (*Figure 1—figure supplement 1*) showed robust GFP expression and mostly normal expression levels of Nav1.1 in *Scn1a*-GFP mouse lines #184 and #233. Both lines showed a similar distribution of chromogenic GFP immunosignals across the entire brain (*Figure 2A–H*), and a similar distribution was also obtained in fluorescence detection of GFP (*Figure 2I–L* and *Figure 2—figure supplement 1*). In neocortex (*Figure 2B, F, J* and *Figure 2—figure supplement 1B, F*), GFP-positive cells were distributed throughout all cortical layers. In hippocampus (*Figure 2C, G, K* and *Figure 2—figure supplement 1C, G*), cells with intense GFP signals, which are assumed to be PV-IN and SST-IN (*Ogiwara et al., 2007*; *Tai et al., 2014*) (see also Figure 8), were scattered in stratum oriens, pyramidale, radiatum, lucidum, and lacunosum-moleculare of the CA (cornu ammonis) fields, hilus and molecular layer of dentate gyrus. Of note, somata of dentate granule cells were apparently GFP-negative. CA1–3 pyramidal cells were twined around with fibrous GFP immunosignals. We previously reported that the fibrous Nav1.1 signals clinging to somata of hippocampal CA1–3 pyramidal cells were disappeared by conditional elimination of Nav1.1 in PV-INs but not in excitatory neurons, and therefore concluded that these Nav1.1-immunopositive fibers are axon terminals of PV-INs (*Ogiwara et al., 2013*). As such, GFP signals are fibrous but do not form cell shapes in the CA pyramidal cell layer (*Figure 2C, G, K* and *Figure 2—figure supplement 1C, G*), and therefore these CA pyramidal cells themselves are assumed to be GFP-negative. These observations further confirmed our previous proposal that hippocampal excitatory neurons are negative for Nav1.1 (*Ogiwara et al., 2007*; *Ogiwara et al., 2013*). In cerebellum (*Figure 2D, H, L* and *Figure 2—figure supplement 1D, H*), GFP signals appeared in Purkinje, basket, and deep cerebellar nuclei cells, again consistent to the previous reports (*Ogiwara et al., 2007*; *Ogiwara et al., 2013*). In the following analyses, we used the line #233 which shows stronger GFP signals than #184.

Quantification of Nav1.1 signals in western blot analyses of brain lysates from the *Scn1a*-GFP mice and their wild-type littermates ($N$ = 5 animals per each genotype) showed no difference between genotypes, while that of GFP somehow deviated among individual *Scn1a*-GFP mice (*Figure 2—figure supplement 2*). Fluorescence imaging of the *Scn1a*-GFP sagittal brain sections at postnatal day 15 (P15), 4-week-old (4W) and 8W showed that GFP signals continue to be intense in caudal region such as thalamus, midbrain, and brainstem (*Figure 3*), which is well consistent with our previous report of Nav1.1 protein and *Scn1a* mRNA distributions in wild-type mouse brain (*Ogiwara et al., 2007*).

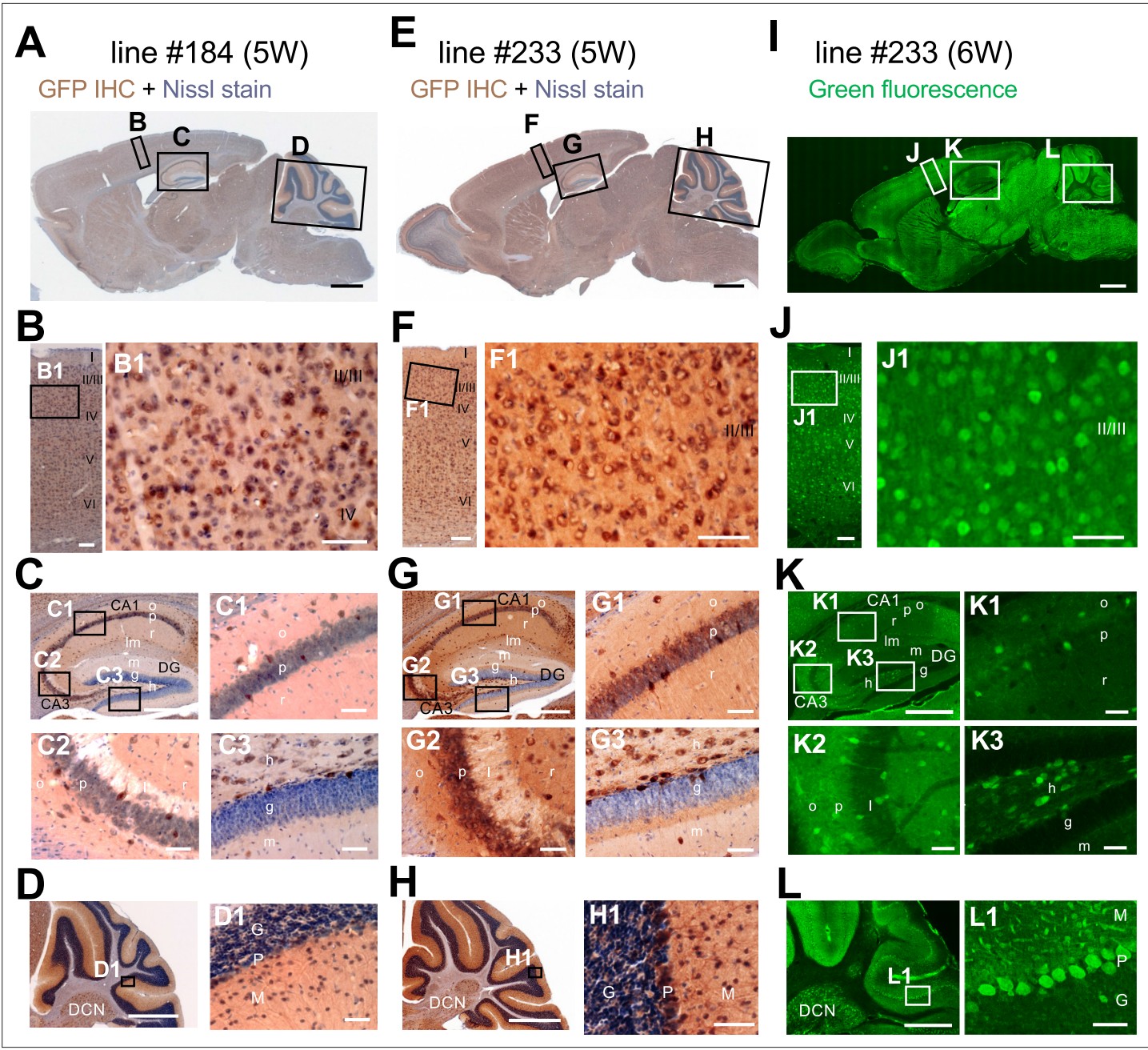

**Figure 2.** Distributions of green fluorescent protein (GFP) signals in brains are similar among *Scn1a*-GFP mouse lines. Chromogenic immunostaining of GFP (brown) with Nissl counterstaining (violet) of lines #184 and #233 (**A–H**) and GFP fluorescence images of line #233 (**I–L**) on parasagittal sections from 5W to 6W *Scn1a*-GFP brains. Boxed areas in (**A, E, I, B, F, J, C, G, K, D, H, L**) are magnified in (B–D, F–H, J–L, B1, F1, J1, C1–3, G1–3, K1–3, D1, H1, L1). The two lines (#184 and #233) showed a similar distribution pattern of GFP-expressing cells across all brain regions (**A–H**), but the signals in the line #233 are more intense than the line #184. In neocortex (**B, F, J**), GFP-expressing cells were scattered throughout the entire region. In the hippocampus (**C, G, K**), GFP-positive inhibitory neurons were sparsely distributed (see also *Figure 8*), while excitatory neurons in stratum pyramidale and stratum granulosum are GFP-negative. In cerebellum (**D, H, L**), Purkinje, basket, and deep cerebellar nuclei cells were GFP-positive. IHC, immunohistochemistry; CA, cornu ammonis; DG, dentate gyrus; o, stratum oriens; p, stratum pyramidale; r, stratum radiatum; lm, stratum lacunosum-moleculare; l, stratum lucidum; m, stratum moleculare; g, stratum granulosum; h, hilus; DCN, deep cerebellar nuclei; M, molecular layer; P, Purkinje cell layer; G, granular cell layer. Scale bars: 1 mm (**A, E, I**), 500 μm (**C, D, G, H, K, L**), 100 μm (**B, F, J**), and 50 μm (B1, C1–3, D1, F1, G1–3, H1, J1, K1–3, L1).

The online version of this article includes the following source data and figure supplement(s) for figure 2:

**Figure supplement 1.** Green fluorescent protein (GFP) distribution in the brains of two *Scn1a*-GFP mice from lines #184 and #233.

**Figure supplement 2.** Nav1.1 expression is stable while green fluorescent protein (GFP) expression varies among individual *Scn1a*-GFP mice.

*Figure 2 continued on next page*

*Figure 2 continued*

**Figure supplement 2—source data 1.** Raw and annotated immunoblots for *Figure 2—figure supplement 2*.

**Figure supplement 2—source data 2.** Numerical source data for *Figure 2—figure supplement 2*.

## Nav1.1 is expressed in both excitatory and inhibitory neurons in neocortex but only in inhibitory neurons in hippocampus

In the neocortex of *Scn1a*-GFP mouse, a large number of cells with GFP-positive somata (GFP-positive cells) were broadly distributed across all cortical layers (*Figure 3* and *Figure 3—figure supplement 1*). Intensities of GFP signals in primary somatosensory cortex (S1) at L2/3 are much higher than other areas such as primary motor cortex (M1) (*Figure 3—figure supplement 1*), however the cell population (density) of GFP-positive cells did not differ in these areas indicating that GFP signals for GFP-positive cells are stronger in S1 at L2/3. Although GFP signals are strong in PV-INs (see Figure 8), cell density of PV-INs is not specifically high at S1 area and therefore most cells with strong GFP signals in S1 at L2/3 may not be PV-INs but excitatory neurons.

In order to know the ratio of GFP-positive cells among all neurons, we further performed immunohistochemical staining using NeuN-antibody on *Scn1a*-GFP mouse at P15 and cells were counted at M1 and S1 (*Figure 3—figure supplement 2* and *Supplementary file 1a*). The NeuN staining showed that GFP-positive cells occupy 30% (L2/3), 32% (L5), and 22% (L6) of NeuN- and GFP-positive cells at P15 (*Figure 3—figure supplement 2B* and *Supplementary file 1a*). However, we noticed that sparsely distributed cells with intense GFP signals, which are assumed to be PV-INs (see Figure 8), were often NeuN-negative (*Figure 3—figure supplement 2* – arrowheads), reminiscent of a previous report that NeuN expression is absent in cerebellar inhibitory neurons such as Golgi, basket, and satellite cells in cerebellum (*Weyer and Schilling, 2003*). Therefore, NeuN-positive cells do not represent all neurons in neocortex as well. NeuN/GFP-double negative neurons could even exist and therefore above figure (*Figure 3—figure supplement 2B*) may deviate from the real ratios of GFP-positive cells among all neurons.

Next, we performed triple immunostaining of Nav1.1, GFP, and ankyrinG on brains of *Scn1a*-GFP mouse at P15. In the neocortex (*Figure 4*), axon initial segments (AISs) of cells with Nav1.1-positive somata were always Nav1.1-positive but somata of cells with Nav1.1-positive AISs were occasionally Nav1.1-negative (*Figure 4A–C*). Cell counting revealed that 17% (L2/3), 21% (L5), and 8% (L6) of neurons (cells with ankyrinG-positive AISs) were GFP-positive (*Figure 4D*, left panel and *Supplementary file 1b*). Of note, all cells with Nav1.1-positive AISs or somata were GFP-positive, but AISs or somata for only half of GFP-positive cells were Nav1.1-positive (*Figure 4D* and *Supplementary file 1c, d*), possibly due to undetectably low levels of Nav1.1 immunosignals in a subpopulation of GFP-positive cells. The above ratios of GFP-positive cells among neurons (cells with ankyrinG-positive AISs) obtained in the triple immunostaining of Nav1.1, GFP, and ankyrinG are rather discordant to those obtained in the later experiment of triple immunostaining of Nav1.2, GFP, and ankyrinG, 23% (L2/3), 30% (L5), and 21% (L6) (see below). Therefore, we additionally performed double immunostaining of GFP and ankyrinG on brains of *Scn1a*-GFP mouse at P15, and the ratios of GFP-positive cells among neurons were 30% (L2/3), 26% (L5), and 9% (L6) (*Figure 4—figure supplement 1* and *Supplementary file 1e*). Averaged ratios of GFP-positive cells among neurons of these experiments are 23% (L2/3), 26% (L5), and 13% (L6) (*Figure 4—figure supplement 2* and *Supplementary file 1f*), which are actually significantly lower than those obtained in the NeuN staining (*Figure 3—figure supplement 2* and *Supplementary file 1a*).

In contrast to the neocortex where only half of GFP-positive cells were Nav1.1-positive, in the hippocampus all GFP-positive cells were Nav1.1-positive and all Nav1.1-positive cells were GFP-positive (*Figure 5*). Actually, most of excitatory neurons such as CA1–3 pyramidal cells and dentate granule cells were GFP-negative. As described above (*Figure 2C, G*), fibrous GFP and Nav1.1 signals twining around CA1~3 pyramidal cells' somata which are assumed to be axon terminals of PV-INs were again observed (*Figure 5A, B, D*). Cell counting in the hippocampal CA1 region showed that 98% of cells with GFP-positive somata were Nav1.1-positive at their AISs and 100% of cells with Nav1.1-positive AISs were GFP-positive (*Figure 5F* and *Supplementary file 1g, h*).

Double in situ hybridization of *Scn1a* and GFP mRNAs showed that these signals well overlap in both neocortex and hippocampus of *Scn1a*-GFP mice (*Figure 6*), further supporting that the

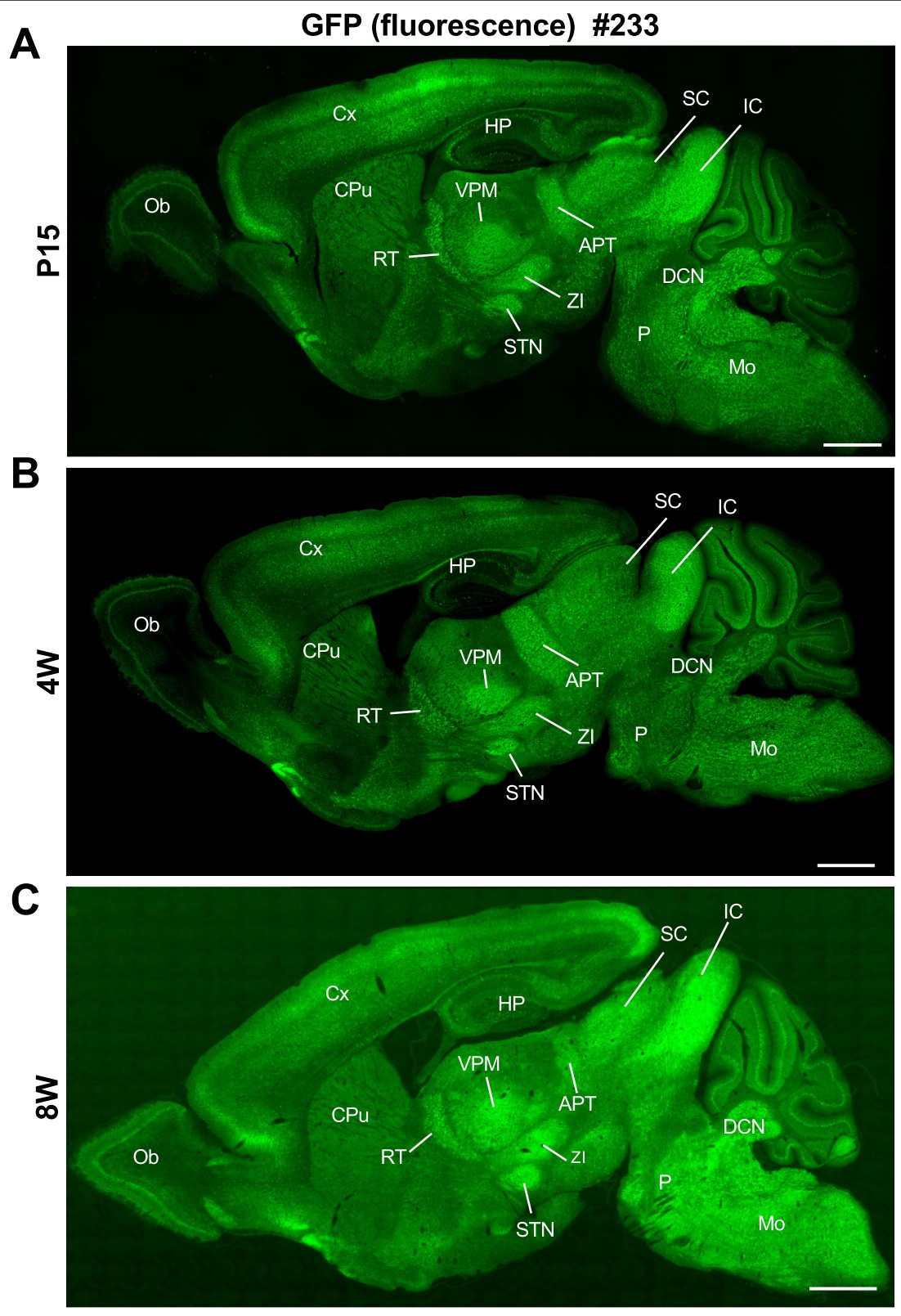

**Figure 3.** Distribution of green fluorescent protein (GFP) signals in *Scn1a*-GFP mouse brain are largely maintained through development. Fluorescent images of parasagittal sections from P15 (**A**), 4W (**B**), and 8W (**C**) *Scn1a*-GFP mouse brains (line #233). GFP signals were observed in multiple brain regions. APT, anterior pretectal nucleus; CPu, caudate putamen; Cx, cerebral cortex; DCN, deep cerebellar nuclei; HP, hippocampus; IC, inferior

*Figure 3 continued on next page*

*Figure 3 continued*

colliculus; Mo, medulla oblongata; Ob, olfactory bulb; P, pons; RT, reticular thalamic nucleus; SC, superior colliculus; STN, subthalamic nucleus; VPM, ventral posteromedial thalamic nucleus; ZI, zona incerta. Scale bars: 1 mm.

The online version of this article includes the following source data and figure supplement(s) for figure 3:

**Figure supplement 1.** Distribution of green fluorescent protein (GFP) fluorescent signals in neocortical layers of *Scn1a*-GFP mouse brain.

**Figure supplement 2.** Immunostaining for NeuN and green fluorescent protein (GFP) in the neocortex of *Scn1a*-GFP mouse.

**Figure supplement 2—source data 1.** Numerical source data for *Figure 3—figure supplement 2*.

GFP signals well represent endogenous *Scn1a*/Nav1.1 expression. Again, in neocortex *Scn1a* and GFP mRNAs seem to be expressed in a number of neurons including some of excitatory pyramidal cells, while in hippocampus they are absent in excitatory neurons such as CA1–3 pyramidal cells and dentate granule cells. All of these distributions of *Scn1a* and GFP mRNAs in *Scn1a*-GFP transgenic mouse brain are consistent to our previous report of regional distributions of *Scn1a* mRNA in wild-type mouse (*Ogiwara et al., 2007*).

To investigate the ratio of inhibitory neurons in GFP-positive cells, we generated and examined *Scn1a*-GFP and vesicular GABA transporter *Slc32a1* (*Vgat*)-Cre (*Ogiwara et al., 2013*) double transgenic mice in which *Slc32a1*-Cre is expressed in all GABAergic inhibitory neurons and visualized by floxed tdTomato transgene (*Figure 7*). In the neocortex at 4W, 23% (L2/3), 28% (L5), and 27% (L6) of GFP-positive cells were Tomato-positive inhibitory neurons and 73% (L2/3), 77% (L5), and 83% (L6) of Tomato-positive cells were GFP-positive (*Figure 7C* and *Supplementary file 1i, j*). These results suggest that a significant subpopulation of neocortical excitatory neurons also express Nav1.1. Our previous observation that Nav1.1 is expressed in callosal axons of neocortical excitatory neurons (*Ogiwara et al., 2013*) supports that a subpopulation of L2/3 CC neurons express Nav1.1. Unlike in neocortex, in the hippocampus most of GFP-positive cells were Tomato-positive, 98% (CA1) and 94% (DG), and majorities of Tomato-positive GABAergic neurons are GFP-positive, 93% (CA1) and 77% (DG). These results further confirmed that in hippocampus Nav1.1 is expressed in inhibitory neurons but not in excitatory neurons. Although somata of pyramidal cells in CA2/3 region are weakly GFP-positive in this and some other experiments (*Figure 7B* and *Figure 2—figure supplement 1G*), those were GFP-negative in other experiments (*Figures 2K and 5A, D*) and therefore the Nav1.1 expression in CA2/3 pyramidal cells would be minimal if any.

We further performed immunohistochemical staining of PV and SST in neocortex and hippocampus of *Scn1a*-GFP mice at 4W (*Figure 8*). PV and SST do not co-express in cells and do not overlap. PV-INs and SST-INs were both GFP-positive, and especially GFP signals in PV-INs were intense (*Figure 8A*). Cell counting revealed that 21% (L2/3), 37% (L5), 37% (L6), 58% (CA1), 42% (CA2/3), and 41% (DG) of GFP-positive cells were PV- or SST-positive depending on regions in neocortex and hippocampus (*Figure 8B* and *Supplementary file 1k–m*). All PV-INs were GFP-positive (*Figure 8B*, middle), and most of SST-INs were GFP-positive (*Figure 8B*, right). Comparison of these results with those of *Slc32a1*-Cre mouse (*Figure 7*) suggests that GFP-positive GABAergic neurons in neocortex are mostly PV- or SST-positive, while in hippocampus a half of those are PV/SST-negative GABAergic neurons. Higher ratios of PV- or SST-positive cells (*Figure 8B*) compared with those of *Slc32a1*-Cre-positive cells (*Figure 7C*) among GFP-positive cells would be explained by us counting a cell as PV-positive if their PV immunosignals are moderate and a significant subpopulation of such cells are known to be excitatory neurons (*Jinno and Kosaka, 2004*; *Tanahira et al., 2009*; *Matho et al., 2021*). Quantitative analysis of GFP signal intensity and area size of cells revealed that GFP signal intensities in PV-positive cells were significantly higher than those in PV-negative cells and GFP signal intensities in SST-positive cells were lower than those in PV-positive cells but similar to PV/SST-double negative cells (*Figure 9* and *Supplementary file 1n–p*). These results indicate that Nav1.1 expression level in PV-INs is significantly higher than those in excitatory neurons and PV-negative GABAergic neurons including SST-INs.

## Nav1.1 and Nav1.2 expressions are mutually exclusive in mouse brain

We previously reported that expressions of Nav1.1 and Nav1.2 seem to be mutually exclusive in multiple brain regions including neocortex, hippocampal CA1, dentate gyrus, striatum, globus pallidus, and cerebellum in wild-type mice (*Yamagata et al., 2017*). To further confirm it, here we performed triple immunostaining for Nav1.1, Nav1.2, and ankyrinG, and counted Nav1.1- or Nav1.2-immunopositive

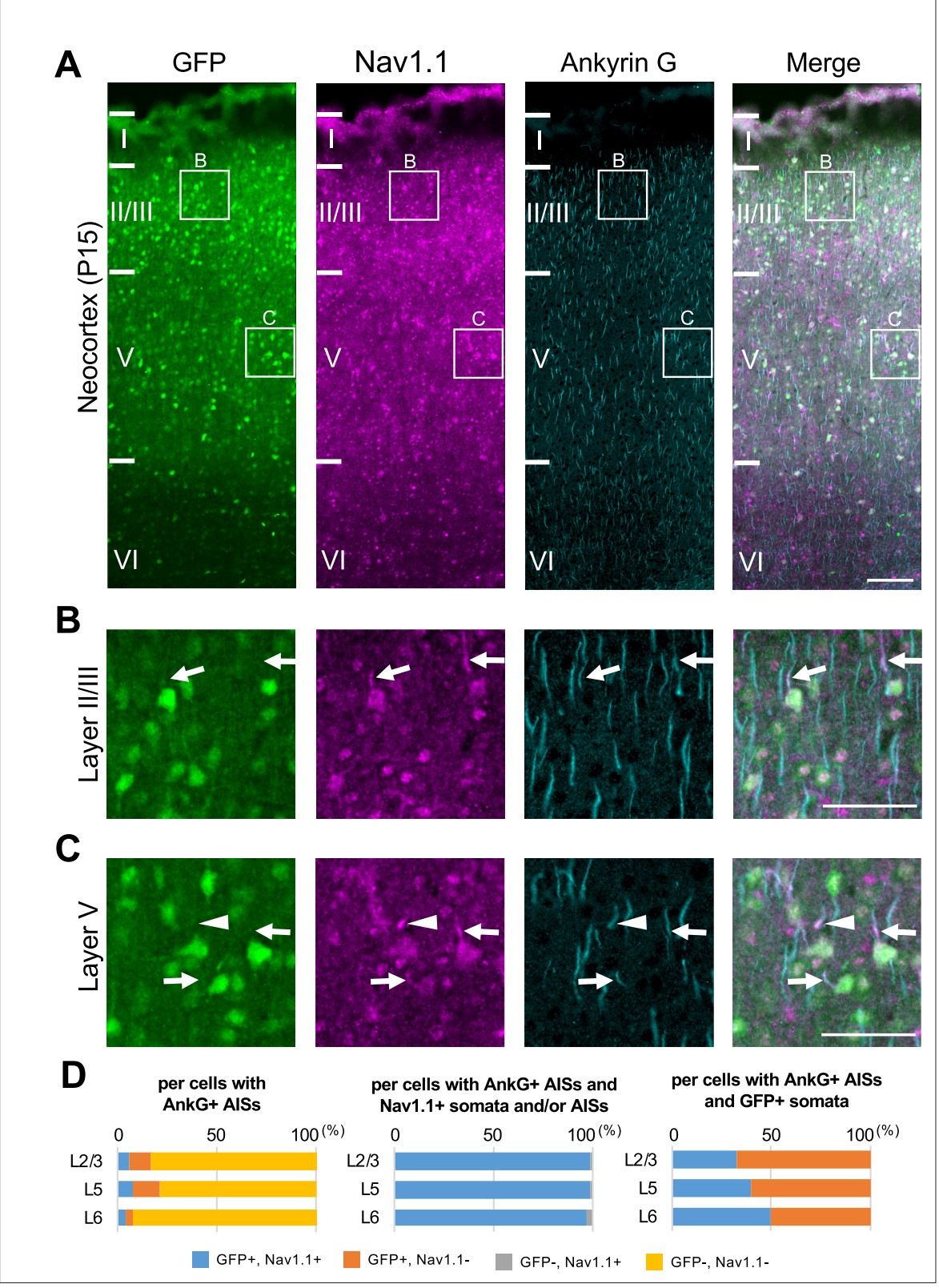

**Figure 4.** Nav1.1 expression at the axon initial segment (AIS) in the *Scn1a*-GFP mouse neocortex. (**A**) Triple immunofluorescent staining of parasagittal sections from P15 *Scn1a*-GFP mouse brain (line #233) by mouse anti-GFP (green), rabbit anti-Nav1.1 (magenta), and goat anti-ankyrinG (cyan) antibodies. Regions at primary motor cortex are shown. (**B, C**) Magnified images outlined in (**A**) are shown in (**B**) and (**C**). Arrows indicate AISs of cells with green fluorescent protein (GFP)-positive somata in which both somata and AISs are positive for Nav1.1. Arrowheads indicate AISs of cells with

*Figure 4 continued on next page*

*Figure 4 continued*

GFP-positive somata in which AISs but not somata are positive for Nav1.1. All images are oriented from pial surface (top) to callosal (bottom). Scale bars: 100 μm (**A**), 50 μm (**B, C**). (**D**) Cell counting of three *Scn1a*-GFP mice. Bar graphs indicating the percentage of cells with GFP- and Nav1.1-positive/negative somata and AISs per cells with ankyrinG-positive AISs (left panel), the percentage of cells with GFP-positive/negative somata per cells with ankyrinG-positive AISs and Nav1.1-positive somata and/or AISs (middle panel), and the percentage of cells with Nav1.1-positive/negative somata and/or AISs per cells with ankyrinG-positive AISs and GFP-positive somata (right panel) in L2/3, L5, and L6 (see also *Supplementary file 1b–d*). Only cells with ankyrinG-positive AISs were counted. Nav1.1 immunosignals were occasionally observed in somata, but in such cases Nav1.1 signals were always observed in their AISs if visible by ankyrinG staining. Note that 99% (L2/3), 99% (L5), and 97% (L6) of cells with Nav1.1-positive AISs have GFP-positive somata (middle panel), but only half or less of cells with GFP-positive somata have Nav1.1-positive AISs (right panel). L2/3, L5: neocortical layer II/III and V. AnkG, ankyrinG; +, positive; −, negative.

The online version of this article includes the following source data and figure supplement(s) for figure 4:

**Source data 1.** Numerical source data for *Figure 4D*.

**Figure supplement 1.** Immunostaining for green fluorescent protein (GFP) and ankyrinG in the neocortex of *Scn1a*-GFP mouse.

**Figure supplement 1—source data 1.** Numerical source data for *Figure 4—figure supplement 1B*.

**Figure supplement 2.** Green fluorescent protein (GFP)-positive cells are more abundant in layers II/III and V than in layer VI of the neocortex.

AISs in the neocortex of *Scn1a*-GFP mice (*Figure 10*). The staining again showed that Nav1.1 and Nav1.2 expressions are mutually exclusive in brain regions including neocortex (*Figure 10A*). The counting revealed that 5% (L2/3), 6% (L5), and 3% (L6) of AISs at P15 were Nav1.1-positive and 78% (L2/3), 69% (L5), and 69% (L6) of AISs at P15 were Nav1.2-positive in the neocortex (*Figure 10B* and *Supplementary file 1q*). Of note, less than 0.5% of AISs are Nav1.1/Nav1.2-double positive confirming that Nav1.1 and Nav1.2 do not co-express. These results are consistent with our previous study (*Yamagata et al., 2017*) and further support that expressions of Nav1.1 and Nav1.2 are mutually exclusive in mouse neocortex at least at immunohistochemical level.

As mentioned, GFP signals in *Scn1a*-GFP mouse can represent even moderate or low Nav1.1 expressions which cannot be detected by immunohistochemical staining, so some of GFP-positive cells may still express Nav1.2. To investigate whether and if so how much of GFP-positive cells have Nav1.2-positive AISs in *Scn1a*-GFP mouse neocortex, we performed triple immunohistochemical staining for Nav1.2, GFP, and ankyrinG (*Figure 11* and *Supplementary file 1r–t*). The staining showed that AISs of GFP-positive cells are largely negative for Nav1.2, and cells with Nav1.2-positive AISs are mostly GFP-negative (*Figure 11A*). Cell counting revealed that 88% (L2/3), 90% (L5), and 95% (L6) of cells with Nav1.2-positive AISs at P15 were GFP-negative (*Figure 11B*, middle panel and *Supplementary file 1s*), and 69% (L2/3), 83% (L5), and 86% (L6) of AISs of GFP-positive cells at P15 were Nav1.2-negative (*Figure 11B*, right panel and *Supplementary file 1t*). These results indicate that the co-expression of GFP and Nav1.2 would be minimal if any.

## Neocortical PT and a subpopulation of CC projection neurons express Nav1.1

Neocortical excitatory neurons can be divided into functionally distinct subpopulations, a majority of those are pyramidal cells which have axons of long-range projections such as L2/3 CC, L5 PT, L5/6 CS, and L6 CT projection neurons (*Shepherd, 2013*). Although PT neurons also project their axon collaterals to ipsilateral striatum, CS neurons project bilaterally to ipsi- and contralateral striata. A transcription factor FEZF2 is expressed in L5 PT neurons, forms their axonal projections, and defines their targets (*Inoue et al., 2004*; *Chen et al., 2005*; *Chen et al., 2008*; *Lodato et al., 2014*). Most of PT neurons are FEZF2-positive (*Molyneaux et al., 2005*; *Matho et al., 2021*). We previously reported that a subpopulation of neocortical L5 pyramidal neurons is Nav1.1-positive (*Ogiwara et al., 2013*), but their natures were unclear. To investigate those, here we performed immunohistochemical staining of FEZF2 and GFP on *Scn1a*-GFP mouse brains (*Figures 12 and 13*, *Supplementary file 1uv and w*). In L5 where a major population of FEZF2-positive cells locate (*Figure 12A*), a majority of FEZF2-positive neurons were GFP-positive (83% and 96% of FEZF2-positive neurons were GFP-positive at P15 and 4W, respectively) (*Figure 12B*, middle panels and *Supplementary file 1v*). In L2/3, FEZF2-positive cells were scarce (*Figure 12A*). In L6, a certain number of FEZF2-positive cells exist but overlaps of FEZF2 and GFP signals are much less compared to those in L5 (*Figure 12* and *Supplementary file 1u–w*). Quantitative analyses revealed that FEZF2/GFP-double positive cells in L5 showed significantly

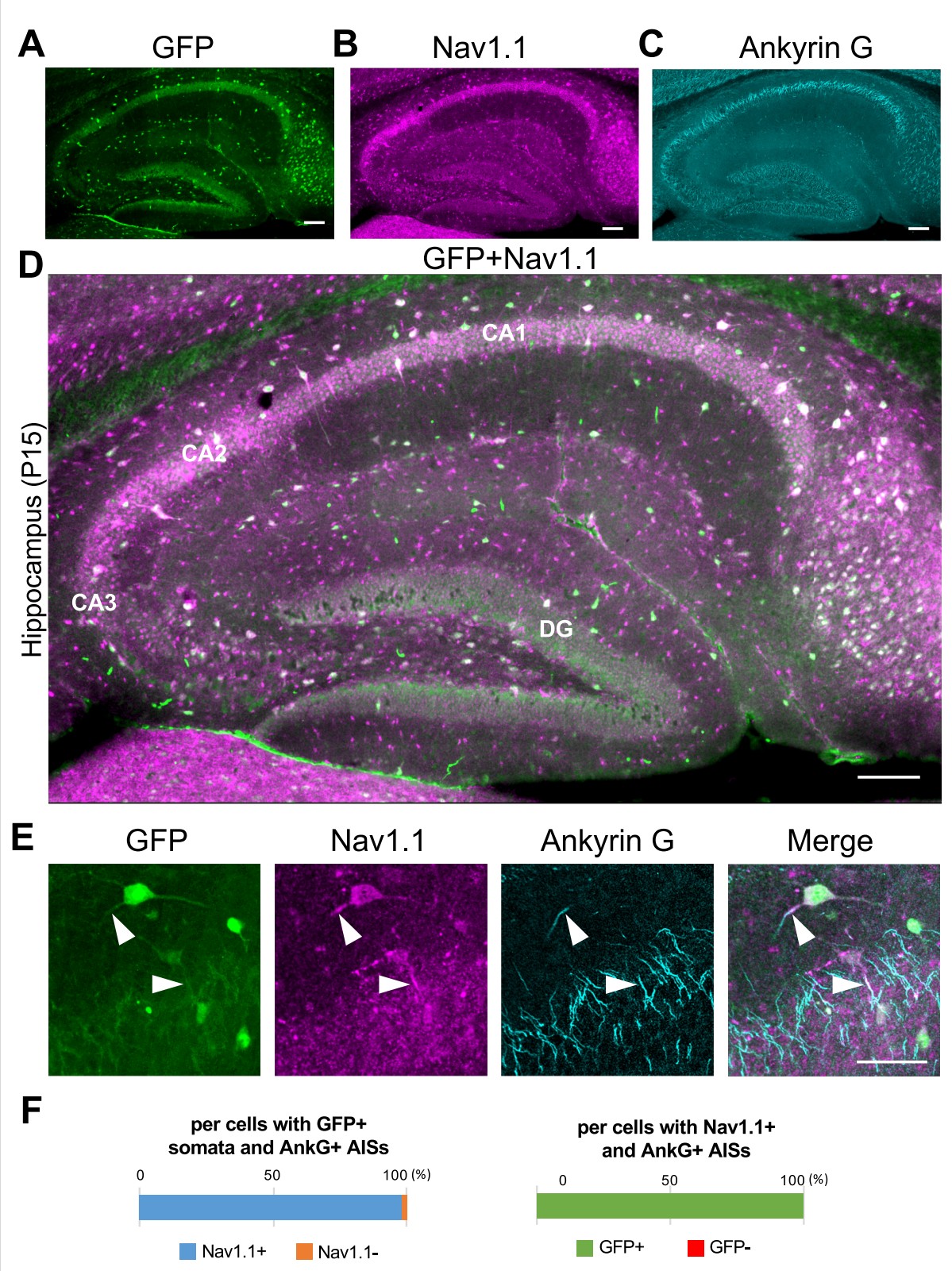

**Figure 5.** Nav1.1 expression at the axon initial segment (AIS) in the *Scn1a*-GFP mouse hippocampus. (**A–D**) Triple immunofluorescent staining of parasagittal sections from P15 *Scn1a*-GFP mouse brain (line #233) by mouse anti-GFP (green), rabbit anti-Nav1.1 (magenta), and goat anti-ankyrinG (cyan) antibodies. Regions at hippocampus were shown. Note that green fluorescent protein (GFP) and Nav1.1 immunosignals mostly overlap at somata. CA1, cornu ammonis 1; CA2, cornu ammonis 2; CA3, cornu ammonis 3; DG, dentate gyrus. Images are oriented from pial surface (top) to callosal

*Figure 5 continued on next page*

*Figure 5 continued*

(bottom). Scale bars: 100 µm. (**E**) Magnified images for co-expression of GFP and Nav1.1 in cells at CA1 region. Arrowheads indicate Nav1.1-positive AISs of GFP expression cells. Scale bar: 50 µm. (**F**) Bar graphs indicate the percentage of cells in hippocampal CA1 region with Nav1.1-positive/negative AISs per cells with GFP-positive somata and ankyrinG-positive AISs (left panel), and the percentage of cells with GFP-positive/negative somata per cells with Nav1.1/ankyrinG-double positive AISs (right panel) (see also *Supplementary file 1g, h*). Only cells with ankyrinG-positive AISs were counted. GFP/Nav1.1-double negative cells, most of which are pyramidal cells, were not counted because of the accumulated nature of their ankyrinG-positive AISs. AnkG, ankyrinG; +, positive; −, negative.

The online version of this article includes the following source data for figure 5:

**Source data 1.** Numerical source data for *Figure 5F*.

lower GFP immunosignal intensities and larger signal areas (soma sizes) compared to those of FEZF2-negative/GFP-positive cells (*Figure 13* and *Supplementary file 1x*), indicating that L5 PT neurons showed lower Nav1.1 expression compared to other neurons such as PV-INs (see also *Figure 9*). Soma sizes of GFP-positive cells in L6 were overall smaller than those of FEZF2/GFP-double positive cells in L5, and there was no statistically significant difference in size between the FEZF2-positive/negative subpopulations. However, FEZF2/GFP-double positive cells still showed lower intensity of GFP signals compared to FEZF2-negative/GFP-positive cells (*Figure 13* and *Supplementary file 1x*).

We further performed triple immunostaining of FEZF2, Nav1.1, and ankyrinG on *Scn1a*-GFP mice at P15, and found that 11% of FEZF2-positive cells have Nav1.1-positive AIS in L5 of *Scn1a*-GFP mouse neocortex (*Figure 12—figure supplement 1* and *Supplementary file 1y, z, aa*). The low ratios of FEZF2/Nav1.1-double positive cells are most possibly due to immunohistochemically undetectable low levels of Nav1.1 expression in these excitatory neurons.

We also performed triple immunostaining of FEZF2, Nav1.2, and ankyrinG on *Scn1a*-GFP mice at P15 (*Figure 12—figure supplement 2* and *Supplementary file 1ab, ac, ad, ae*). The staining showed that 20% of neurons (cells with ankyrinG-positive AISs) in L5 are FEZF2-positive and a half of L5 FEZF2-positive cells have Nav1.2-positive AISs. Together with the observation that most of FEZF2-positive cells are GFP-positive (*Figure 12*), these results indicate that a subpopulation of FEZF2-positive PT neurons may express both Nav1.1 and Nav1.2.

We additionally performed triple immunostaining of FEZF2, GFP, and Nav1.2 on *Scn1a*-GFP mice at P15 (*Figure 12—figure supplement 3* and *Supplementary file 1af*), showing that in L5 74% of FEZF2-positive cells are GFP-positive but a majority of their AISs are Nav1.2-negative. The ratios of Nav1.2-positive cells among FEZF2-positive cells obtained in the triple immunostaining of FEZF2, GFP, and Nav1.2 (*Figure 12—figure supplement 3*) are 27% (L5) and 40% (L6). These results further support the above notion that a subpopulation of FEZF2-positive PT neurons may express both Nav1.1 and Nav1.2. Although further studies such as retrograde tracking analyses are required to confirm and figure out the detailed circuits, all these results propose that the majority of L5 PT neurons express Nav1.1.

## The majority of CT, CS, and CC projection neurons express Nav1.2

TBR1 (T-box brain 1 transcription factor) is a negative regulator of FEZF2 and therefore not expressed in the PT neurons, all of which are known to be FEZF2-positive (*Chen et al., 2008*; *Han et al., 2011*; *McKenna et al., 2011*). TBR1 is predominantly expressed in L6 CT neurons and subpopulations of L2/3 and L5 non-PT excitatory neurons instead (*Han et al., 2011*; *McKenna et al., 2011*; *Matho et al., 2021*). To further elucidate the distributions of GFP (Nav1.1)-expressing neurons in neocortex, we performed immunohistochemical staining of TBR1 on *Scn1a*-GFP mice (*Figure 14* and *Supplementary file 1ag, ah, ai*) and quantitated GFP signal intensities and area sizes of cells (*Figure 15* and *Supplementary file 1aj*). TBR1-positive cells were predominant in neocortical L6, some in L5 and a few in L2/3. In L5, contrary to the high ratios of GFP-positive cells among FEZF2-positive cells (83% at P15 and 96% at 4W) (*Figure 12B*), the ratios of GFP-positive cells among TBR1-positive cells are quite low (11% at P15 and 5% at 4W) (*Figure 14B*, middle panels and *Supplementary file 1ah*). Soma sizes of TBR1/GFP-double positive cells were smaller than those of TBR1-negative/GFP-positive cells (*Figure 15*, middle panel and *Supplementary file 1aj*). In L6 where a major population of TBR1-positive neurons locate, the ratios of GFP-positive cells among TBR1-positive cells are still low (15% at P15 and 26% at 4W) (*Figure 14B*, middle panels and *Supplementary file 1ah*). Soma sizes of TBR1/GFP-double positive cells were also smaller than those of TBR1-negative/GFP-positive cells, and TBR1/GFP-double positive cells showed lower intensity of GFP immunosignals compared to

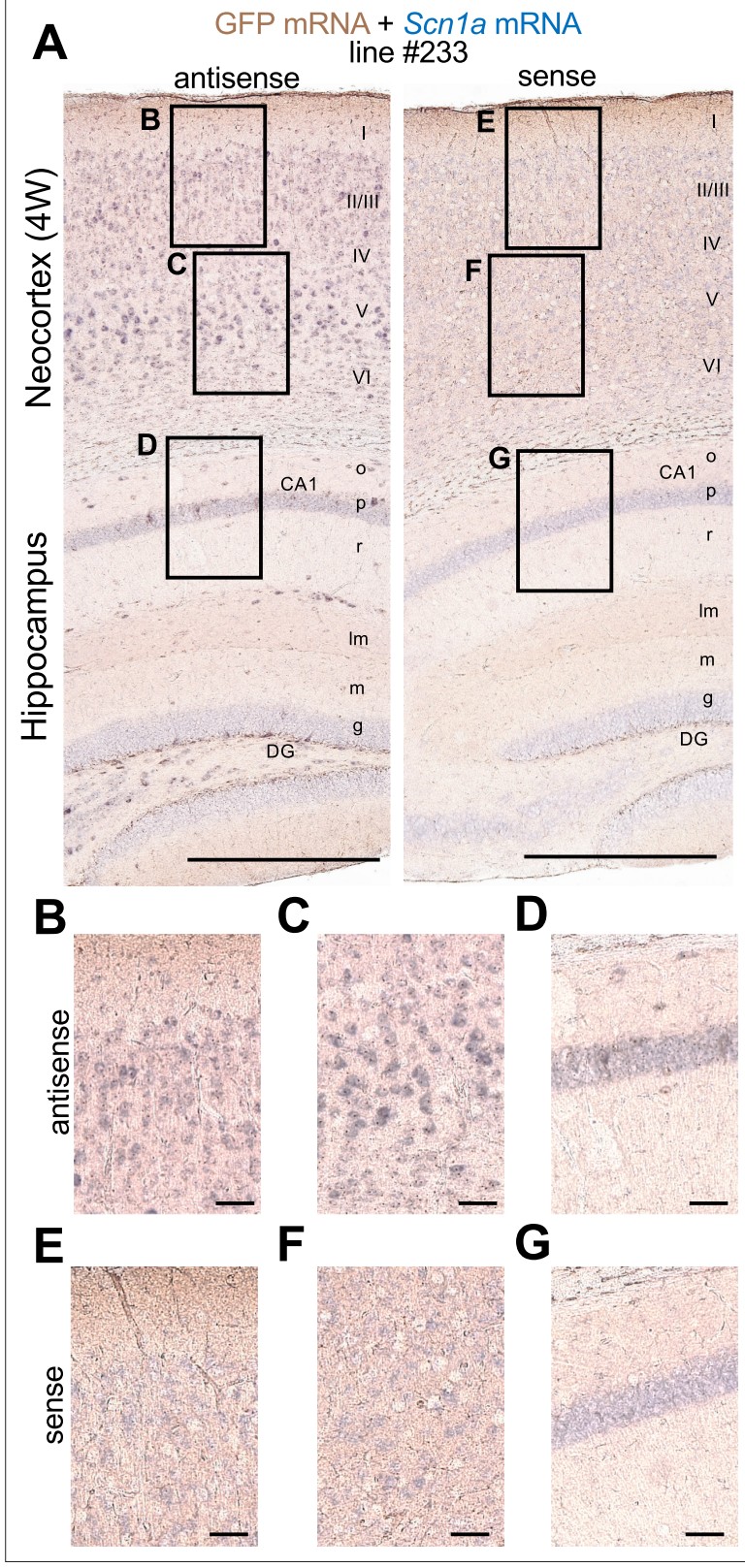

**Figure 6.** Green fluorescent protein (GFP) and *Scn1a* mRNAs expression mainly overlap in *Scn1a*-GFP mouse brain. Double in situ hybridization for *Scn1a*-GFP transgene mRNA and endogenous *Scn1a* mRNA on parasagittal sections from 4W *Scn1a*-GFP brains (line #233). (**A**) Sections were hybridized with antisense (left) and sense (right) RNA probes for GFP transgene (brown) and endogenous *Scn1a* (blue) mRNA species and chromogenically stained.

*Figure 6 continued on next page*

*Figure 6 continued*

Magnified images outlined in (**A**) are shown in (**B–D**) for antisense probes, and (**E–G**) for sense probes. o, stratum oriens; p, stratum pyramidale; r, stratum radiatum; lm, stratum lacunosum-moleculare; m, stratum moleculare; g, stratum granulosum, CA1, cornu ammonis 1; DG, dentate gyrus. Scale bars: 500 μm (**A**), 50 μm (**B–G**).

TBR1-negative/GFP-positive cells (*Figure 15*, right panel and *Supplementary file 1aj*). We additionally performed triple immunostaining for TBR1, Nav1.1, and ankyrinG on *Scn1a*-GFP mice (*Figure 14—figure supplement 1* and *Supplementary file 1ak, al, am*). Notably, the ratios of Nav1.1-positive cells among TBR1-positive cells are 0% in all layers (*Figure 14—figure supplement 1B*, right panel and *Supplementary file 1am*). These results indicate that the major population of TBR1-positive cells do not express Nav1.1.

To investigate whether TBR1-positive cells express Nav1.2, we performed triple immunostaining of TBR1, Nav1.2, and ankyrinG on *Scn1a*-GFP mice (*Figure 14—figure supplement 2* and *Supplementary file 1an, ao, ap, aq*). In L5, contrary to the low ratios of GFP-positive cells among TBR1-positive cells (11% at P15 and 5% at 4W) (*Figure 14B*, middle panels and *Supplementary file 1aq*), the ratios of Nav1.2-positive cells among TBR1-positive cells are high (69% (L2/3), 69%(L5), and 69% (L6) at P15) (*Figure 14—figure supplement 2B*, right-upper panel and *Supplementary file 1ap*). The ratios of TBR1-positive cells among Nav1.2-positive cells are 29% (L2/3), 53% (L5), and 62% (L6) at P15 (*Figure 14—figure supplement 2B*, middle upper panel and *Supplementary file 1ao*).

We further performed triple immunostaining for TBR1, Nav1.2, and GFP on *Scn1a*-GFP mice (*Figure 14—figure supplement 3* and *Supplementary file 1ar*) and found that most (88%) of L6 TBR1-positive cells are GFP-negative.

Taken together, these results indicate that most TBR1-positive neurons including L6 CT neurons do not express Nav1.1 but expresses Nav1.2.

As a whole, above results showed that a minor subpopulation of L2/3 CC and L5 PT neurons express Nav1.1 while the majority of L2/3 CC, L5/6 CS, and L6 CT neurons express Nav1.2. A breakdown of L5 neuron subtypes is specifically described in *Figure 14—figure supplement 4*.

## Discussion

In our present study, we developed *Scn1a* promoter-driven GFP mice in which the expression of GFP replicates that of Nav1.1. All PV-INs and most of SST-INs were GFP-positive in the neocortex and hippocampus of the *Scn1a*-GFP mouse, being consistent with the previous reports of Nav1.1 expression in those inhibitory neurons (*Ogiwara et al., 2007*; *Lorincz and Nusser, 2008*; *Ogiwara et al., 2013*; *Li et al., 2014*; *Tai et al., 2014*; *Tian et al., 2014*; *Yamagata et al., 2017*). All Nav1.1-positive cells were GFP-positive. Reversely all GFP-positive cells were also Nav1.1-positive in the hippocampus, but in the neocortex only a half of GFP-positive cells were Nav1.1-positive. This is largely because in the hippocampus Nav1.1 expression is restricted to inhibitory neurons, but in the neocortex Nav1.1 is expressed not only in inhibitory but also in a subpopulation of excitatory neurons in which Nav1.1 expression is low and not easily detected immunohistochemically by anti-Nav1.1 antibodies. In neocortex, one-third of GFP-positive cells were GABAergic cells such as PV-INs and SST-INs, and the remaining two-third were excitatory neurons. GFP signals were especially intense in PV-positive cells indicating strong Nav1.1 expression in those cells, while GFP signals in SST-positive cells were similar to those in excitatory neurons. In addition, extensive immunostaining analyses using projection neuron markers FEZF2 and TBR1 together with anti-Nav1.2 antibody also revealed that most L5 PT neurons and a minor subpopulation of L2/3 CC neurons express Nav1.1 while the majority of L6 CT, L5/6 CS, and L2/3 CC neurons express Nav1.2.

The above observations should contribute to understanding of neural circuits responsible for diseases such as epilepsy and neurodevelopmental disorders caused by *SCN1A* and *SCN2A* mutations. Dravet syndrome is a sporadic intractable epileptic encephalopathy characterized by early onset (6 months to 1 year after birth) epileptic seizures, intellectual disability, autistic features, ataxia, and increased risk of sudden unexpected death in epilepsy (SUDEP). De novo loss-of-function mutations of *SCN1A* are found in more than 80% of the patients (*Claes et al., 2001*; *Sugawara et al., 2002*; *Fujiwara et al., 2003*; *Dravet et al., 2005*; *Depienne et al., 2009*; *Meng et al., 2015*). In mice, loss-of-function *Scn1a* mutations caused clinical features reminiscent of Dravet syndrome, including

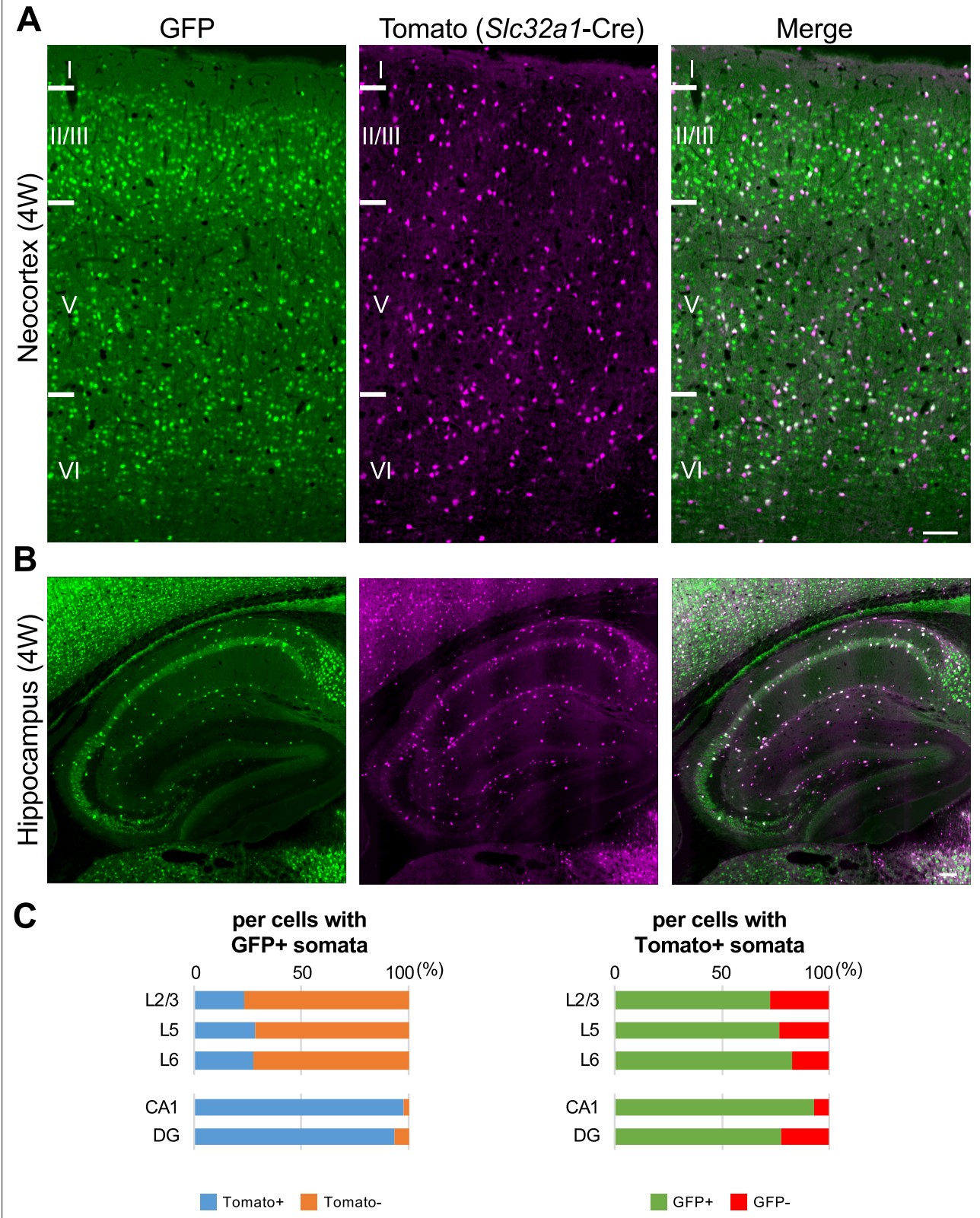

**Figure 7.** One-third of green fluorescent protein (GFP)-positive cells in neocortex are inhibitory neurons, but most of GFP-positive cells in hippocampus are inhibitory neurons. (**A, B**) GFP (green) and Tomato (magenta) fluorescent images of parasagittal sections from 4W *Scn1a*-GFP/*Slc32a1*-cre/Ai14 mouse. Regions at primary motor cortex (**A**) and hippocampus (**B**) are shown. Scale bar: 100 µm. (**C**) Bar graphs indicate the percentage of cells with Tomato-positive/negative somata per cells with GFP-positive somata (left panel) (see also ***Supplementary file 1i***) and the percentage of cells with

*Figure 7 continued on next page*

*Figure 7 continued*

GFP-positive/negative somata per cells with Tomato-positive somata (right panel) (see also *Supplementary file 1j*) in L2/3, L5, L6, CA1, and DG. Cells in primary motor cortex and hippocampus of *Scn1a*-GFP mouse at 4W were counted. L2/3, L5, L6, CA1, and DG: neocortical layer II/III, V, VI, cornu ammonis 1, dentate gyrus. +, positive; −, negative.

The online version of this article includes the following source data for figure 7:

**Source data 1.** Numerical source data for *Figure 7C*.

early-onset epileptic seizures, hyperactivity, learning and memory deficits, reduced sociability and ataxic gaits and premature sudden death (*Yu et al., 2006*; *Ogiwara et al., 2007*; *Oakley et al., 2009*; *Cao et al., 2012*; *Han et al., 2012*; *Kalume et al., 2013*; *Ito et al., 2013*). As also shown in the present study, Nav1.1 is densely localized at AISs of inhibitory cells such PV-IN (*Ogiwara et al., 2007*; *Ogiwara et al., 2013*; *Li et al., 2014*; *Tai et al., 2014*) and selective elimination of Nav1.1 in PV-IN in mice leads to epileptic seizures, sudden death, and deficits in social behavior and spatial memory (*Ogiwara et al., 2013*; *Tatsukawa et al., 2018*). It is thus plausible that Nav1.1 haplo-deficiency in PV-IN plays a pivotal role in the pathophysiology of many clinical aspects of Dravet syndrome. Notably, mice with selective Nav1.1 reduction in global inhibitory neurons were at a greater risk of lethal seizure than heterozygote null mice (Nav1.1 KO/+), and the mortality risk of mice with Nav1.1 haplo-deficiency in inhibitory neurons was significantly decreased or improved with additional Nav1.1 reduction in dorsal telencephalic excitatory neurons (*Ogiwara et al., 2013*). These observations indicate beneficial effects of Nav1.1 deficiency in excitatory neurons for epileptic seizures and sudden death. Because of the absence of Nav1.1 in hippocampal excitatory neurons (*Ogiwara et al., 2007*; *Ogiwara et al., 2013*; *Yamagata et al., 2017*) and the present study, the ameliorating effect was most possibly caused by Nav1.1 haploinsufficiency in neocortical excitatory neurons. *Kalume et al., 2013* reported that parasympathetic hyperactivity is observed in Nav1.1 haplo-deficient mice and it causes ictal bradycardia and finally result in seizure-associated sudden death.

Our present finding of Nav1.1 expression in L5 PT projection neurons which innervate the vagus nerve may possibly elucidate the ameliorating effects of Nav1.1 haploinsufficiency in neocortical excitatory neurons for sudden death of Nav1.1 haplo-deficient mice and may contribute to the understanding of the neural circuit for SUDEP in patients with Dravet syndrome. Further studies including retrograde tracing and electrophysiological analyses are needed.

We previously proposed that impaired CS excitatory neurotransmission causes epilepsies in *Scn2a* haplo-deficient mouse (*Miyamoto et al., 2019*). Our present finding of Nav1.2 expression in CS neurons is consistent and further support the proposal. Because *SCN2A* has been well established as one of top genes which show de novo loss-of-function mutations in patients with ASD (*Hoischen et al., 2014*; *Johnson et al., 2016*) and because impaired striatal function was suggested in multiple ASD animal models (*Fuccillo, 2016*), our finding of Nav1.2 expression in CS neurons may also contribute to the understanding of neural circuit for ASD caused by *SCN2A* mutations.

In summary, the present investigations using a newly developed *Scn1a* promoter-driven GFP mice together with anti-Nav1.1/Nav1.2 antibodies and neocortical neuron markers revealed the cellular expression of Nav1.1 and Nav1.2 in more detail. Further developments of mice containing fluorescent protein reporters driven by promoters for other sodium channel genes and combinatorial analyses of those mice are needed to segregate and redefine their unique functional roles in diverse neuronal populations of complexed neural circuits.

## Materials and methods

### Animal work statement

All animal experimental protocols were approved by the Animal Experiment Committee of Nagoya City University and RIKEN Center for Brain Science. Mice were handled in accordance with the guidelines of the Animal Experiment Committee.

### Mice

*Scn1a*-GFP BAC transgenic mice were generated as follows. A murine BAC clone RP23-232A20 containing the *Scn1a* locus was obtained from the BACPAC Resource Center (https://bacpacresources.

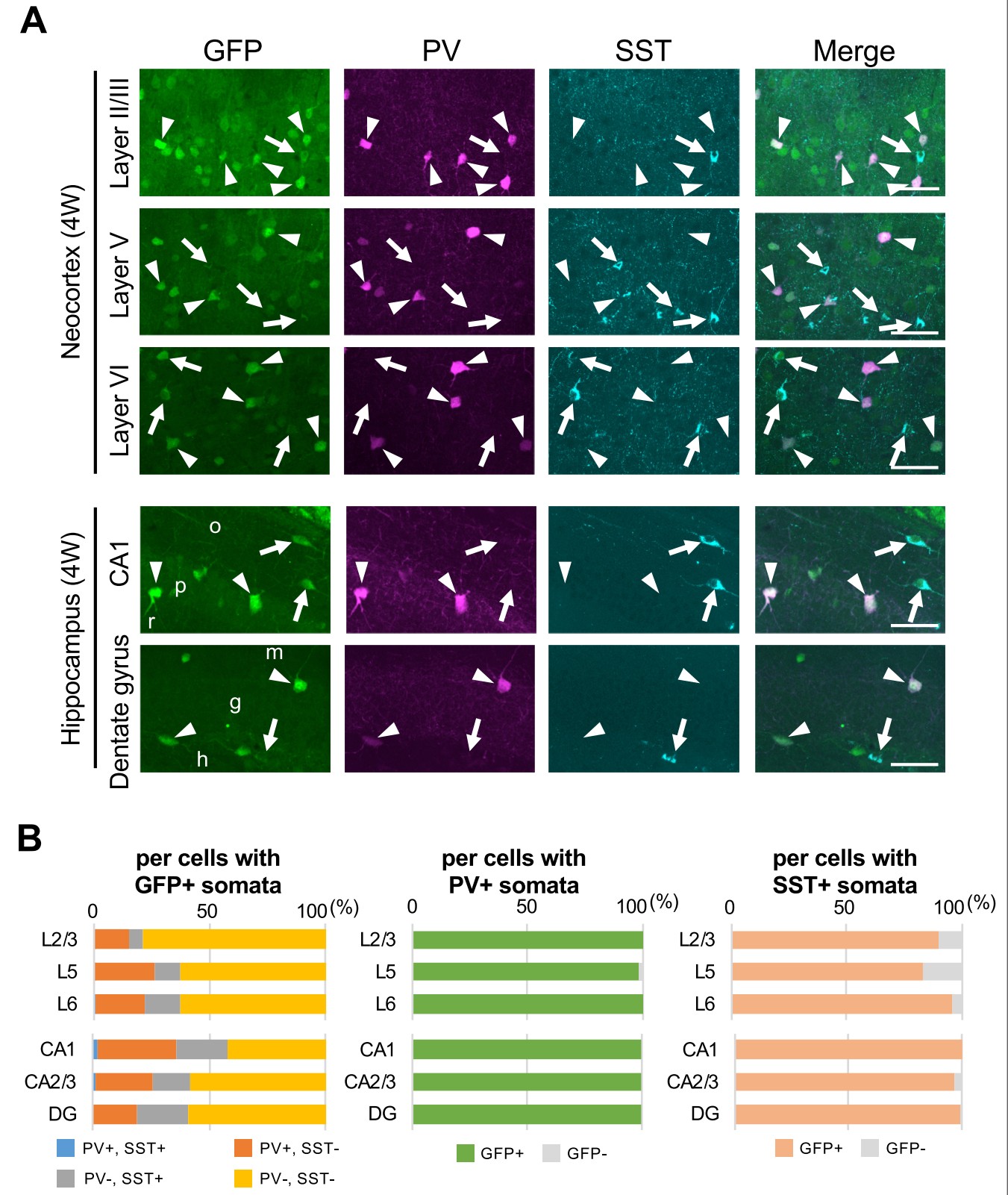

**Figure 8.** Parvalbumin- or somatostatin-positive inhibitory neurons are green fluorescent protein (GFP)-positive in *Scn1a*-GFP mouse neocortex and hippocampus. (**A**) Triple immunofluorescent staining of parasagittal sections from 4W *Scn1a*-GFP mouse (line #233) by mouse anti-GFP (green), rabbit anti-parvalbumin (PV) (magenta), and goat anti-somatostatin (SST) (cyan) antibodies. Regions at neocortex and hippocampus are shown. Merged images were shown in the right columns. Arrows indicate SST/GFP-double positive cells. Arrowheads indicate PV/GFP-double positive. o, stratum

*Figure 8 continued on next page*

*Figure 8 continued*

oriens; p, stratum pyramidale; r, stratum radiatum; h, hilus; g, stratum granulosum; m, stratum moleculare. All images are oriented from pial surface (top) to callosal (bottom). Scale bars: 50 µm. (**B**) Bar graphs indicate the percentage of cells with PV- and SST-positive/negative somata per cells with GFP-positive somata (left panel) (see also *Supplementary file 1k*), the percentage of cells with GFP-positive/negative somata per cells with PV-positive somata (middle panel) (see also *Supplementary file 1l*), and the percentage of cells with GFP-positive/negative somata per cells with SST-positive somata (right panel) (see also *Supplementary file 1m*) in L2/3, L5, L6, CA1, CA2/3, and DG. Cells in neocortex and hippocampus of *Scn1a*-GFP mouse at 4W were counted. L2/3, L5, L6, CA1, CA2/3, and DG: neocortical layer II/III, V, VI, cornu ammonis 1, 2 plus 3, dentate gyrus. +, positive; −, negative.

The online version of this article includes the following source data for figure 8:

**Source data 1.** Numerical source data for *Figure 8B*.

org). A GFP reporter cassette, comprising a red-shifted variant GFP cDNA and a downstream polya-denylation signal derived from pIRES2-EGFP (Takara Bio), was inserted in-frame into the initiation codon of the *Scn1a* coding exon 1 using the Red/ET Recombineering kit (Gene Bridges), according to the manufacturer's instructions. A correctly modified BAC clone verified using PCR and restriction mapping was digested with *Sac*II, purified using CL-4B sepharose (GE Healthcare), and injected into pronuclei of C57BL/6J zygotes. Mice carrying the BAC transgene were identified using PCR with primers: mScn1a_TG_check_F1, 5′-TGTTCTCCACGTTTCTGGTT-3′, mScn1a_TG_check_R1, 5′-TTAG CCTTCTCTTCTGCAATG-3′, and EGFP_R1, 5′-GCTCCTGGACGTAGCCTTC-3′ that detect the wild-type *Scn1a* allele as an internal control (186 bp) and the inserted transgene (371 bp). Of 15 inde-pendent founder lines that were crossed with C57BL/6J mice, 12 lines successfully transmitted the transgene to their progeny. Of 12 founders, two lines (#184 and 233) that display much stronger green fluorescent intensity compared with other lines were selected, and maintained on a congenic C57BL/6J background. The mouse lines had normal growth and development. The line #233 has been deposited to the RIKEN BioResource Research Center (https://web.brc.riken.jp/en/) for distri-bution under the registration number RBRC10241. *Slc32a1*-Cre BAC transgenic mice and loxP flanked transcription terminator cassette CAG promotor-driven tdTomato transgenic mice Ai14 (B6. Cg-*Gt(ROSA)26Sor*^tm14(CAG-tdTomato)Hze^/J, Stock No: 007914, The Jackson Laboratory, USA) were main-tained on a C57BL/6J background. To generate triple mutant mice (*Scn1a*-GFP, *Slc32a1*-Cre, Ai14), heterozygous *Scn1a*-GFP and *Slc32a1*-Cre mice were mated with homozygous *Rosa26*-tdTomato transgenic mice.

## Western blot analysis

Mouse brains at 5W were isolated and homogenized in homogenization buffer [(0.32 M sucrose, 10 mM 4-(2-hydroxyethyl)-1-piperazineethanesulfonic acid (HEPES), 2 mM ethylenediaminetetraacetic acid (EDTA), and 1× complete protease inhibitor cocktail (Roche Diagnostics), pH 7.4)], and centri-fuged for 15 min at $1000 \times g$. The supernatants were next centrifuged for 30 min at $30,000 \times g$. The resulting supernatants were designated as the cytosol fraction. The pellets were subsequently resuspended in lysis buffer (50 mM HEPES and 2 mM EDTA, pH 7.4) and centrifuged for 30 min at $30,000 \times g$. The resulting pellets, designated as the membrane fraction, were dissolved in 2 M Urea, 1× NuPAGE reducing agent (Thermo Fisher Scientific) and 1× NuPAGE LDS sample buffer (Thermo Fisher Scientific). The cytosol and membrane fractions were separated on the NuPAGE Novex Tris-acetate 3–8% gel (Thermo Fisher Scientific) or the PAG mini SuperSep Ace Tris-glycine 5–20% gel (FUJIFILM Wako Pure Chemical), and transferred to nitrocellulose membranes (Bio-Rad). Membranes were probed with the rabbit anti-Nav1.1 (250 ng/ml; IO1, *Ogiwara et al., 2007*), chicken anti-GFP (1:5000; ab13970, Abcam), and mouse anti-β tubulin (1:10,000; T0198, Sigma-Aldrich) antibodies, and incubated with the horseradish peroxidase-conjugated goat anti-rabbit IgG (1:2000; sc-2004, Santa Cruz Biotechnology), rabbit anti-chicken IgY (1:1000; G1351, Promega), and goat anti-mouse IgG (1:5000; W4011, Promega) antibodies. Blots were detected using the enhanced chemiluminescence reagent (PerkinElmer). The intensity of the Nav1.1 immunosignals was quantified using the Image Studio Lite software (LI-COR, Lincoln, NE, USA) and normalized to the level of β-tubulin or glyceralde-hyde 3-phosphate dehydrogenase.

## Histochemistry

Mice were deeply anesthetized, perfused transcardially with 4% paraformaldehyde (PFA) in phosphate-buffered saline (PBS) (10 mM phosphate buffer, 2.7 mM KCl, and 137 mM NaCl, pH 7.4)

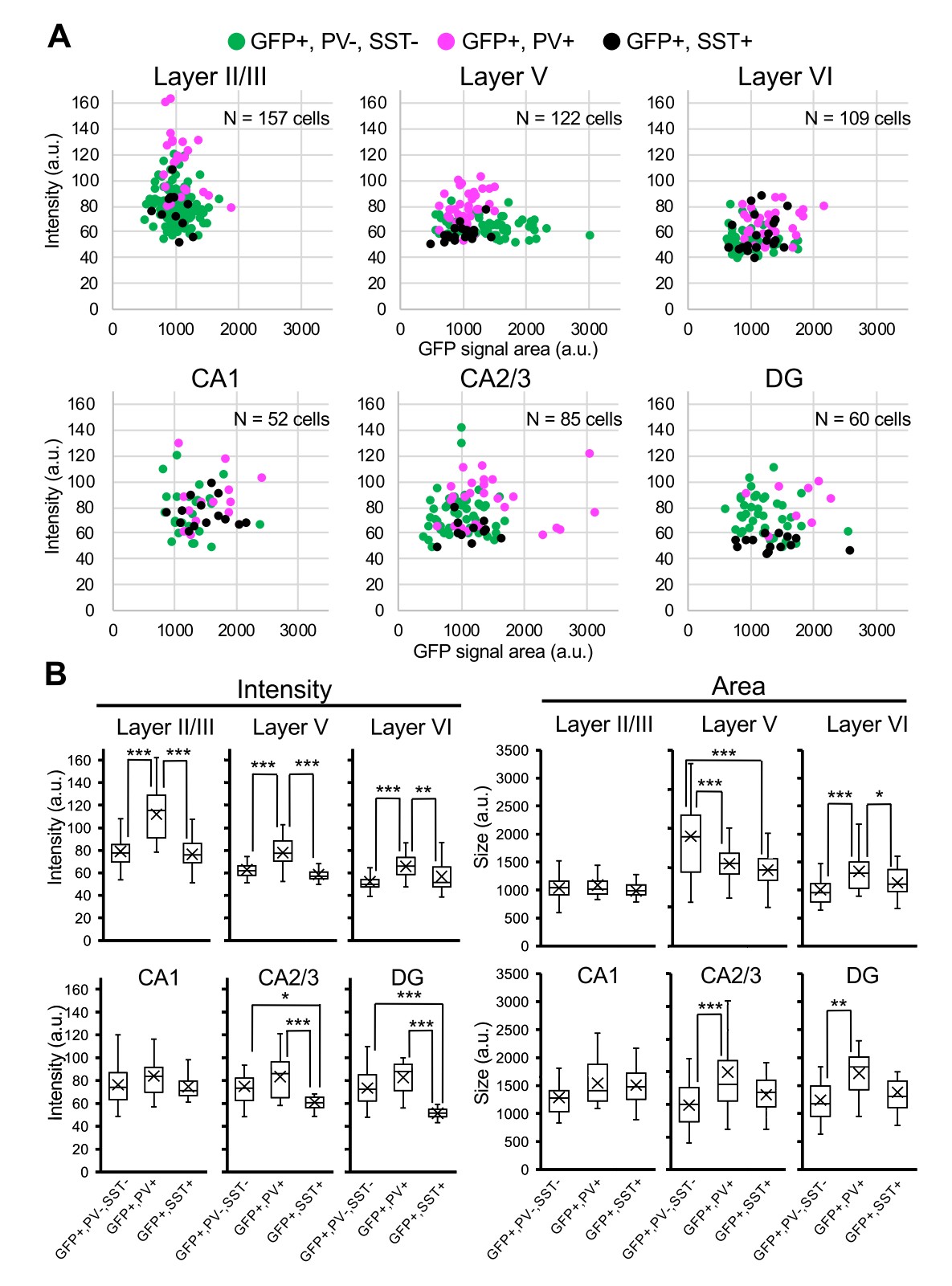

**Figure 9.** Green fluorescent protein (GFP) signals in parvalbumin-positive inhibitory neurons are higher than PV-negative/GFP-positive cells in *Scn1a*-GFP mouse neocortex. (**A**) Scatter plots of intensities and area sizes of GFP immunosignals in GFP-positive cells with PV- or SST-positive or -negative somata. Cells at primary motor cortex (upper panels) and hippocampus (lower panels) in parasagittal sections from 4W *Scn1a*-GFP mouse (line #233) were analyzed. PV-positive (magenta circles) or SST-positive (black circles) and -negative (green circles) cells in neocortical L2/3, L5, and L6 or

*Figure 9 continued on next page*

*Figure 9 continued*

hippocampal CA1, CA2/3, and DG are plotted (see also **Supplementary file 1n**). (**B**) Box plots represent values for the intensity and area size in each cell type (see also **Supplementary file 1o, p**). Cross marks indicate mean values in each cell type. Statistical significance was assessed using one-way analysis of variance (ANOVA) followed by Tukey–Kramer post hoc multiple comparison test. *p < 0.05, **p < 0.01, ***p < 0.001. Note that GFP signal intensities of PV/GFP-double positive cells were significantly higher than that of SST/GFP-double positive cells and PV/SST-negative/GFP-positive cells (all layers), while GFP signal intensities of SST/GFP-double positive cells were similar to PV/SST-negative/GFP-positive cells in neocortex. In hippocampus, GFP signal intensities of SST/GFP-double positive cells were significantly lower than that of SST-negative/GFP-positive cells at CA2/3 and DG. CA1, CA2/3, and DG: cornu ammonis 1, 2 plus 3, dentate gyrus. a.u., arbitrary unit; +, positive; −, negative.

The online version of this article includes the following source data for figure 9:

**Source data 1.** Numerical source data for **Figure 9**.

or periodate-lysine-4% PFA (PLP). Brains were removed from the skull and post-fixed. For fluorescent imaging, PFA-fixed brains were cryoprotected with 30% sucrose in PBS, cut in 30 µm parasagittal sections, and mounted on glass slides. The sections on glass slides were treated with TrueBlack Lipofuscin Autofluorescence Quencher (Biotium) to reduce background fluorescence. For immunostaining, frozen parasagittal sections (30 µm) were blocked with 4% BlockAce (DS Pharma Biomedical) in PBS for 1 hr at room temperature (RT), and incubated with rat anti-GFP (1:500; GF090R, Nacalai Tesque). The sections were then incubated with the secondary antibodies conjugated with biotin (1:200; Vector Laboratories). The antibody–antigen complexes were visualized using the Vectastain Elite ABC kit (Vector Laboratories) with Metal enhanced DAB substrate (34065, PIERCE). For immunofluorescent staining, we prepared 6 µm parasagittal sections from paraffin embedded PLP-fixed brains of mice. The sections were processed as previously described (**Yamagata et al., 2017**). Following antibodies were used to detect GFP, Nav1.1, Nav1.2, TBR1, FEZF2, ankyrinG, NeuN, parvalbumin, and somatostatin; mouse anti-GFP antibodies (1:500; 11814460001, Roche Diagnostics), anti-Nav1.1 antibodies (1:10,000; rabbit IO1, 1:500; goat SC-16031, Santa Cruz Biotechnology), anti-Nav1.2 antibodies (1:1000; rabbit ASC-002, Alomone Labs; goat SC-31371, Santa Cruz Biotechnology), rabbit anti-TBR1 antibody (1:1000; ab31940, Abcam or 1:500; SC-376258, Santa Cruz Biotechnology), rabbit anti-FEZF2 antibody (1:500; #18997, IBL), ankyrinG antibodies (1:500; mouse SC-12719, rabbit SC-28561; goat, SC-31778, Santa Cruz Biotechnology), mouse anti-NeuN biotin-conjugated antibody (1:2000; MAB377B, Millipore), rabbit anti-parvalbumin (1:5000; PC255L, Merck), and rabbit anti-somatostatin (1:5000; T-4103, Peninsula Laboratories, 1:1000; SC-7819, Santa Cruz Biotechnology) antibodies. As secondary antibodies, Alexa Fluor Plus 488-, 555-, 594-, and 647-conjugated antibodies (1:1000; A32723, A32766, A32794, A32754, A32849, A32795, A32787, Thermo Fisher Scientific) were used. To detect NeuN, Alexa 647-conjugated streptavidin (1:1000; S21374, Thermo Fisher Science) was used. Images were captured using fluorescence microscopes (BZ-8100 and BZ-X710, Keyence), and processed with Adobe Photoshop Elements 10 (Adobe Systems) and BZ-X analyzer (Keyence).

## Fluorescence and immunofluorescence quantification

For quantification of inhibitory neurons in GFP-positive cells, we used *Scn1a*-GFP/*Slc32a1*-Cre/Ai14 mice at 4W. We acquired multiple color images of primary motor cortex and hippocampus from three parasagittal sections per animal. Six images per region of interest were manually counted and summarized using Adobe Photoshop Elements 10 and Excel (Microsoft). On immunofluorescence quantification, we used *Scn1a*-GFP mice at P15 and/or 4W for the quantification of immunosignals. For quantification of GFP-, NeuN-, PV-, SST-, Nav1.1-, Nav1.2-, FEZF2-, or TBR1-positive cells, we acquired multiple color images of primary motor cortex and hippocampus from three parasagittal sections per animal. Six to nine images per region of interest were manually quantified and summarized. For quantification of PV-, SST-, FEZF2-, or TBR1-positive cells, intensity and area size of GFP fluorescent signals were measured by Fiji software. Statistical analyses were performed by one-way analysis of variance followed by Tukey–Kramer post hoc multiple comparison test using Kyplot 6.0 (KyensLab Inc). p value smaller than 0.05 was considered statistically significant. Data are presented as the mean ± standard error of the mean.

## In situ hybridization

Frozen sections (30 µm) of PFA-fixed mouse brains at 4W were incubated in 0.3% $H_2O_2$ in PBS for 30 min at RT to quench endogenous peroxidases, and mounted on glass slides. The sections on slides

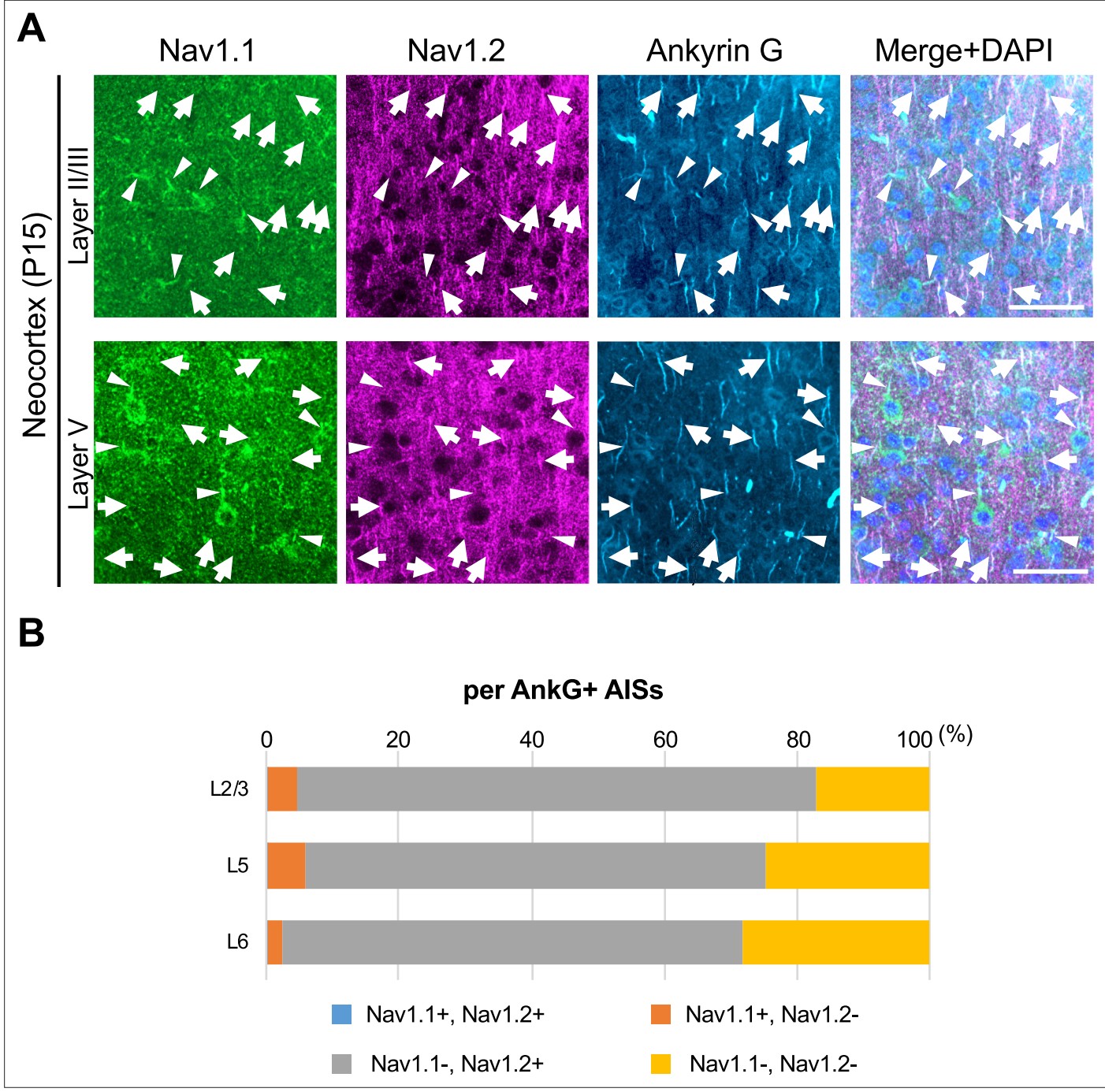

**Figure 10.** Nav1.1 and Nav1.2 are mutually exclusive at axon initial segments (AISs) in mouse brain. (**A**) Triple immunofluorescent staining on parasagittal sections from *Scn1a*-GFP mouse at P15 by rabbit anti-Nav1.1 (green), goat anti-Nav1.2 (magenta), and mouse anti-ankyrinG (cyan) antibodies. Merged images are shown in the right panels. Arrows indicate Nav1.2-positive AISs. Arrowheads indicate Nav1.1-positive AISs. Note that there are no Nav1.1/Nav1.2-double positive AISs. Images are oriented from pial surface (top) to callosal (bottom). Scale bars: 50 μm. (**B**) Bar graphs indicating the percentage of Nav1.1- and Nav1.2-positive/negative AISs per AISs detected by ankyrinG staining in L2/3, L5, and L6 of *Scn1a*-GFP mice. Note that Nav1.1/Nav1.2-double positive AISs were less than 0.5% of all AISs in these layers (see ***Supplementary file 1q***). L2/3, L5, L6: neocortical layer II/III, V, VI. AnkG, ankyrinG; +, positive; −, negative.

The online version of this article includes the following source data for figure 10:

**Source data 1.** Numerical source data for *Figure 10B*.

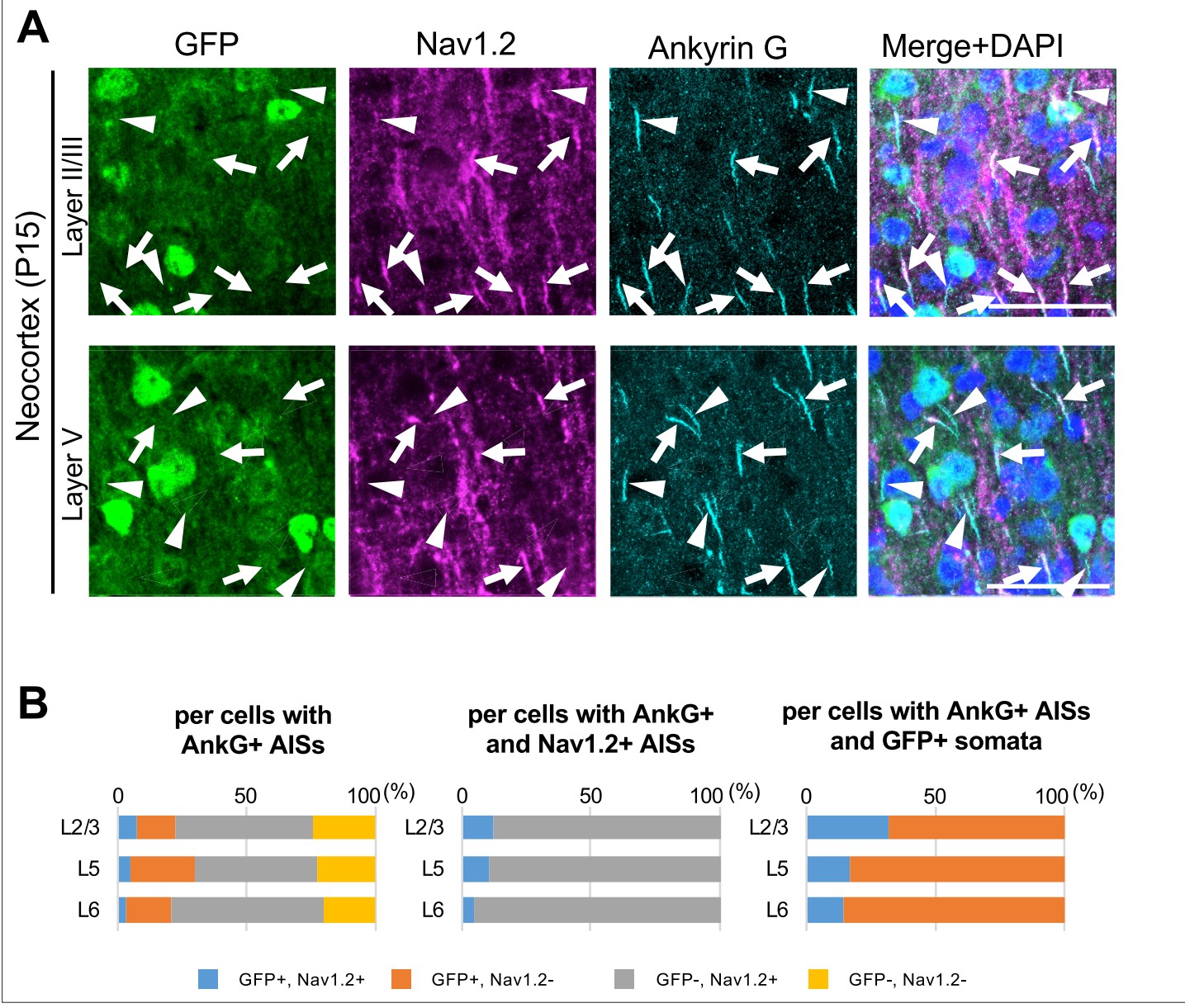

**Figure 11.** Cells with Nav1.2-positive axon initial segments (AISs) are mostly green fluorescent protein (GFP)-negative in *Scn1a*-GFP mouse neocortex. (**A**) Triple immunofluorescent staining of parasagittal sections from P15 *Scn1a*-GFP mouse brain (line #233) by mouse anti-GFP (green), goat anti-Nav1.2 (magenta), and rabbit anti-ankyrinG (cyan) antibodies. Merged images of the signals are shown in the right panels. Arrows indicate Nav1.2-positive AISs of cells with GFP-negative somata. Arrowheads indicate Nav1.2-negative AISs of cells with GFP-positive somata. All images are oriented from pial surface (top) to callosal (bottom). Scale bars: 50 μm. (**B**) Bar graphs indicating the percentage of cells with GFP- and Nav1.2-positive/negative somata and AISs per cells with ankyrinG-positive AISs (left panel) (see also *Supplementary file 1r*), the percentage of cells with GFP-positive/negative somata per cells with ankyrinG/Nav1.2-double positive AISs (middle panel) (see also *Supplementary file 1s*), and the percentage of cells with Nav1.2-positive/negative AISs per cells with ankyrinG-positive AISs and GFP-positive somata (right panel) (see also *Supplementary file 1t*) in L2/3, L5, and L6. Note that 88% (L2/3), 90% (L5), and 95% (L6) of cells with Nav1.2-positive AISs have GFP-negative somata (middle panel), and 68% (L2/3), 83% (L5), and 86% (L6) of cells with GFP-positive somata have Nav1.2-negative AISs (right panel). L2/3, L5, L6: neocortical layer II/III, V, VI. AnkG, ankyrinG; +, positive; −, negative.

The online version of this article includes the following source data for figure 11:

**Source data 1.** Numerical source data for *Figure 11B*.

were UV irradiated with 1250 mJ/cm$^2$ (Bio-Rad), permeabilized with 0.3% Triton X-100 in PBS for 15 min at RT, and digested with 1 μg/ml proteinase K (Nacalai Tesque) in 10 mM Tris–HCl and 1 mM EDTA, pH 8.0, for 30 min at 37°C, washed twice with 100 mM glycine in PBS for 5 min at RT, fixed with 4% formaldehyde in PBS for 5 min at RT, and acetylated with 0.25% acetic anhydride in 100 mM

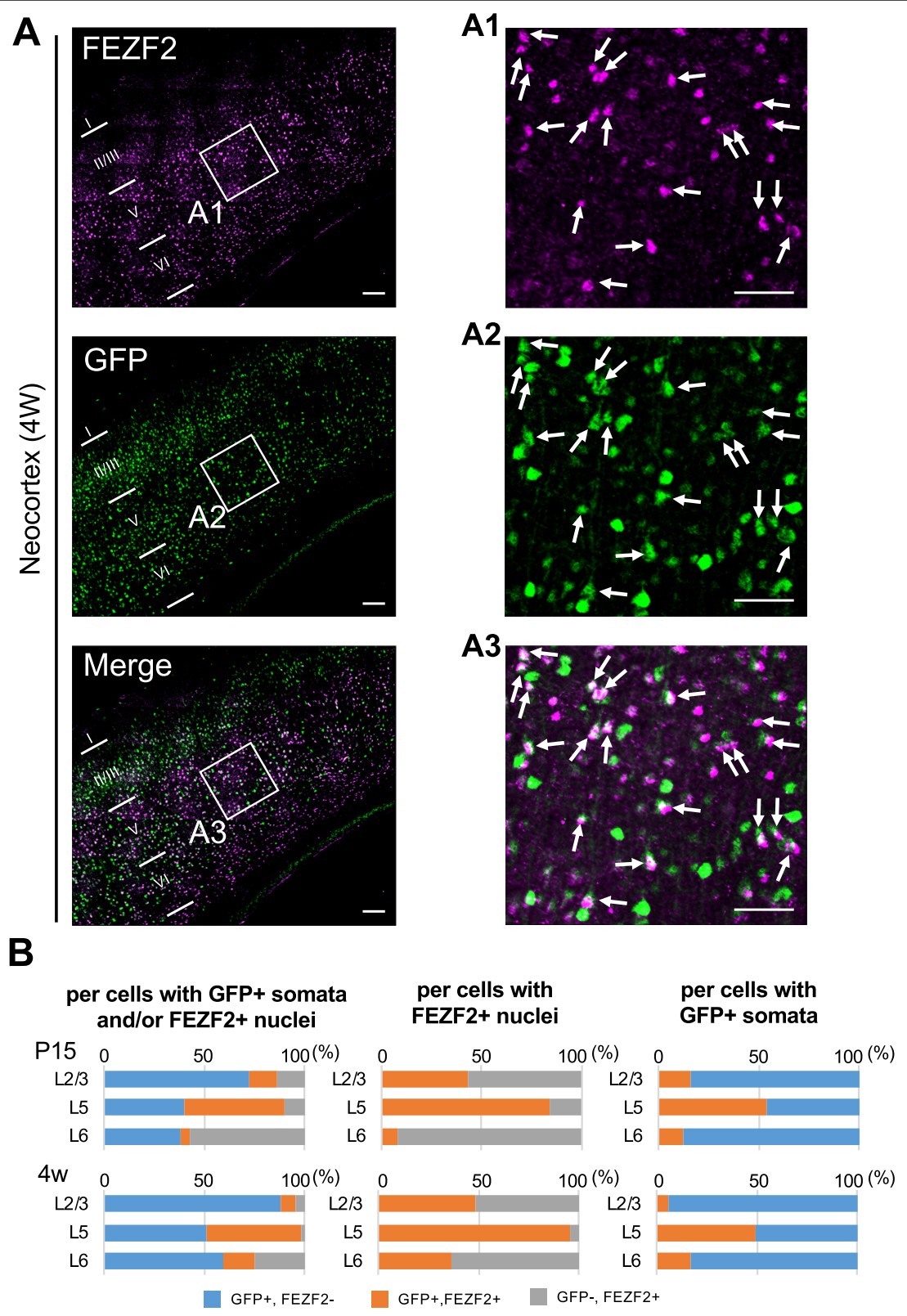

**Figure 12.** Cells positive for FEZF2 are mostly green fluorescent protein (GFP)-positive in L5 of *Scn1a*-GFP mouse neocortex. (**A**) Double immunostaining of FEZF2 and GFP in neocortex of 4W *Scn1a*-GFP mouse (line #233) by rabbit anti-FEZF2 (magenta) and mouse anti-GFP (green) antibodies. Arrows indicate FEZF2/GFP-double positive cells. Magnified images outlined in (**A**) are shown in (**A1–A3**). Note that FEZF2 signals mostly overlap with GFP signals in L5. Many of the remained GFP-positive/FEZF2-negative cells have intense GFP signals and are assumed to be inhibitory

*Figure 12 continued on next page*

*Figure 12 continued*

neurons (see **Figure 8**). Scale bars: 100 µm (**A**), 50 µm (**A1–A3**). (**B**) Bar graphs indicating the percentage of cells with FEZF2- and GFP-positive/negative nuclei and somata per cells with GFP-positive somata and/or FEZF2-positive nuclei (left panels) (see also **Supplementary file 1u**), the percentage of cells with GFP-positive/negative somata per cells with FEZF2-positive nuclei (middle panels) (see also **Supplementary file 1v**), and the percentage of cells with FEZF2-positive/negative nuclei per cells with GFP-positive somata (right panels) (see also **Supplementary file 1w**) in L2/3, L5, and L6. Cells at primary motor cortex of *Scn1a*-GFP mouse at P15 and 4W were counted. Note that 83% (P15) and 96% (4W) of cells with FEZF2-positive cells are GFP-positive in L5 (middle panels), but a half of cells with GFP-positive cells are FEZF2-positive in L5 (right panel). L2/3, L5, L6: neocortical layer II/III, V, VI. +, positive; −, negative.

The online version of this article includes the following source data and figure supplement(s) for figure 12:

**Source data 1.** Numerical source data for **Figure 12B**.

**Figure supplement 1.** Some FEZF2-positive cells in L5 of the *Scn1a*-GFP mouse neocortex have Nav1.1-positive axon initial segment (AIS).

**Figure supplement 1—source data 1.** Numerical source data for **Figure 12—figure supplement 1B**.

**Figure supplement 2.** A half of FEZF2-positive cells at L5 have Nav1.2-positive axon initial segment (AIS) of *Scn1a*-GFP mouse neocortex.

**Figure supplement 2—source data 1.** Numerical source data for **Figure 12—figure supplement 2B**.

**Figure supplement 3.** Most of the FEZF2/green fluorescent protein (GFP)-double positive cells are Nav1.2-negative in L5 of *Scn1a*-GFP mouse neocortex.

**Figure supplement 3—source data 1.** Numerical source data for **Figure 12—figure supplement 3B**.

triethanolamine, pH 8.0. After acetylation, the sections were washed twice with 0.1 M phosphate buffer, pH 8.0, incubated in a hybridization buffer [(50% formamide, 5× Sodium Chloride-Sodium Phosphate-EDTA buffer (SSPE), 0.1% Sodium dodecyl sulfate (SDS), and 1 mg/ml Yeast tRNA (Roche Diagnostics))] containing the Avidin solution (Vector Laboratories) for 2 hr at 60°C, and hybridized with 2 µg/ml digoxigenin (DIG)- and dinitrophenol (DNP)-labeled probes in a hybridization buffer

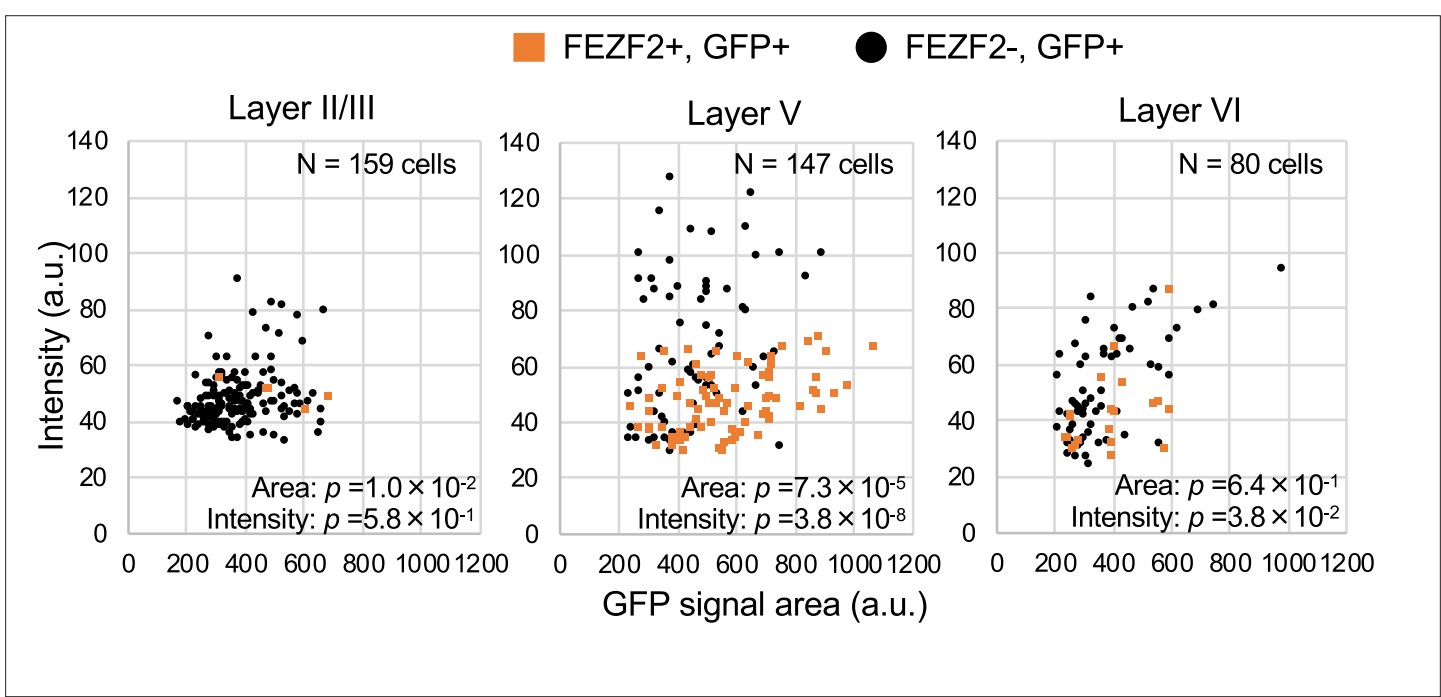

**Figure 13.** FEZF2-positive cells have lower green fluorescent protein (GFP) signal intensities in *Scn1a*-GFP mouse neocortex. Scatter plots of intensities and area sizes of GFP immunosignals in GFP-positive cells with FEZF2-positive/negative nuclei. Cells at primary motor cortex in parasagittal sections from 4W *Scn1a*-GFP mouse (line #233) were analyzed. FEZF2-positive (orange squares) and negative (black circles) cells in neocortical L2/3, L5, and L6 are plotted (see also **Supplementary file 1x**). Note that GFP signal intensities of FEZF2/GFP-double positive cells were significantly lower than that of FEZF2-negative/GFP-positive cells (L5, L6), and signal area size of FEZF2/GFP-double positive cells was significantly larger than that of FEZF2-negative/GFP-positive cells (L2/3, L5). Statistical significance was assessed using *t*-test. a.u., arbitrary unit; +, positive; −, negative.

The online version of this article includes the following source data for figure 13:

**Source data 1.** Numerical source data for **Figure 13**.

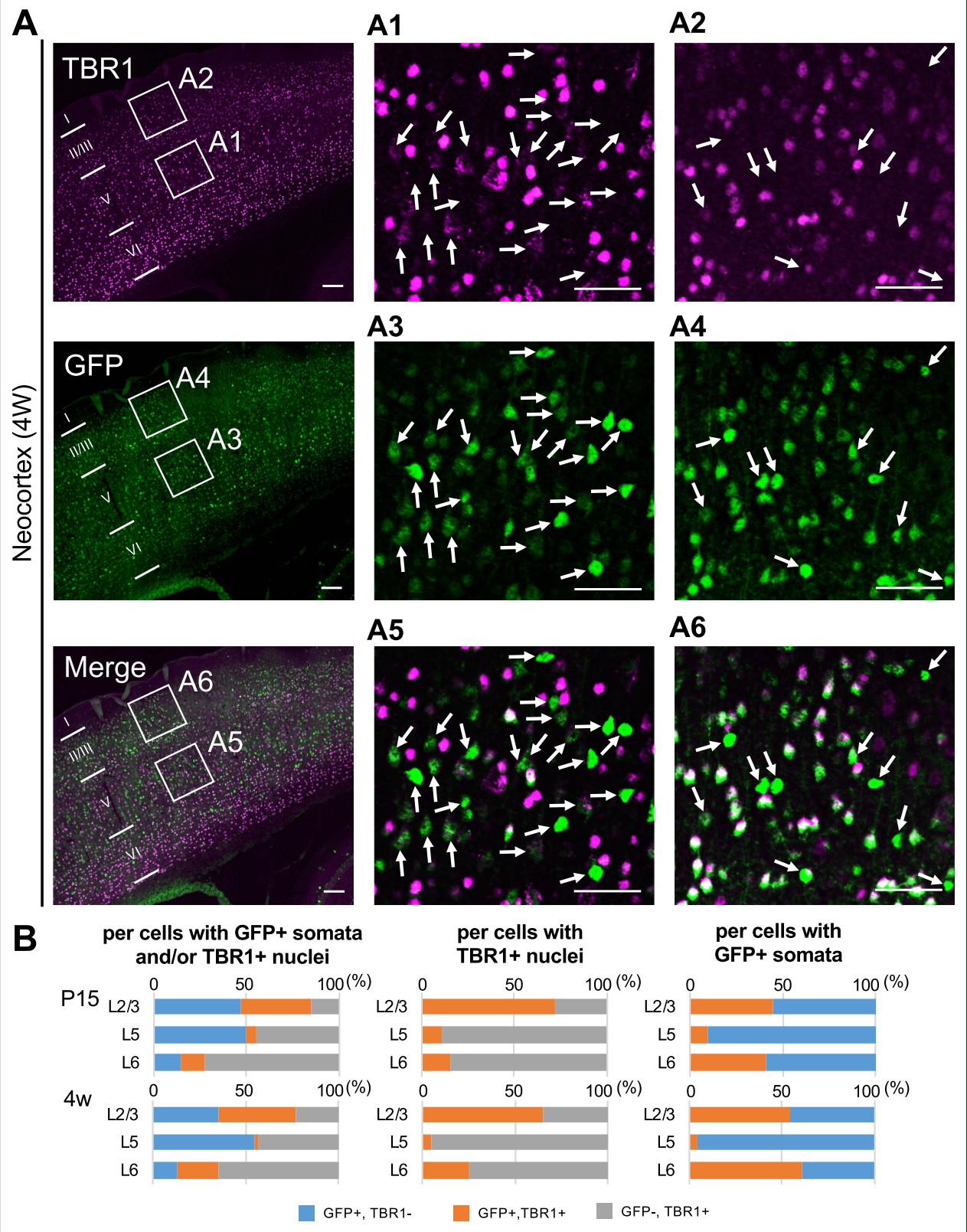

**Figure 14.** Green fluorescent protein (GFP)-positive cells were mostly negative for TBR1 in L5 of *Scn1a*-GFP mouse neocortex. (**A**) Double immunostaining of TBR1 and GFP in neocortex of 4W *Scn1a*-GFP mouse (line #233) detected by mouse rabbit anti-TBR1 (magenta) and anti-GFP (green) antibodies. Arrows indicate TBR1-negative/GFP-positive cells. Magnified images outlined in (**A**) are shown in (**A1–A6**). Note that at L5 GFP-positive cells were mostly TBR1-negative but at L2/3 more than half of GFP-positive cells were TBR1-positive. Scale bars: 100 μm (**A**), 50 μm (**A1–A6**). (**B**) Bar

*Figure 14 continued on next page*

*Figure 14 continued*

graphs indicating the percentage of cells with TBR1- and GFP-positive/negative nuclei and somata per GFP-positive cells and/or TBR1-positive nuclei (left panels) (see also *Supplementary file 1ag*), the percentage of cells with GFP-positive/negative somata per cells with TBR1-positive nuclei (middle panels) (see also *Supplementary file 1ah*), and the percentage of cells with TBR1-positive/negative nuclei per cells with GFP-positive somata (right panels) (see also *Supplementary file 1ai*) in L2/3, L5, and L6. Cells in primary motor cortex of *Scn1a*-GFP mouse at P15 and 4W were counted. Note that 86% (P15) and 95% (4W) of cells with TBR1-positive cells are GFP-negative in L5 (middle panels), and 90% (P15) and 96% (4W) of cells with GFP-positive cells are TBR1-negative in L5 (right panel). L2/3, L5, L6: neocortical layer II/III, V, VI. +, positive; −, negative.

The online version of this article includes the following source data and figure supplement(s) for figure 14:

**Source data 1.** Numerical source data for *Figure 14B*.

**Figure supplement 1.** Axon initial segments (AISs) of TBR1-positive cells are Nav1.1-negative in *Scn1a*-GFP mouse neocortex.

**Figure supplement 1—source data 1.** Numerical source data for *Figure 14—figure supplement 1B*.

**Figure supplement 2.** 70% of TBR1-positive cells have Nav1.2-positive axon initial segment (AIS) in *Scn1a*-GFP mouse neocortex.

**Figure supplement 2—source data 1.** Numerical source data for *Figure 14—figure supplement 2B*.

**Figure supplement 3.** TBR1/Nav1.2-double positive cells are green fluorescent protein (GFP)-negative in L5 and L6 in *Scn1a*-GFP mouse neocortex.

**Figure supplement 3—source data 1.** Numerical source data for *Figure 14—figure supplement 3B*.

**Figure supplement 4.** Distributions of Nav1.1 (green fluorescent protein, GFP) and Nav1.2 in neocortical layer V revealed by the analysis of *Scn1a*-GFP mouse.

containing the Biotin solution (Vector Laboratories) overnight at 60°C in a humidified chamber. The hybridized sections were washed with 50% formamide in 2× Saline-sodium citrate buffer (SSC) for 15 min at 50°C twice, incubated in TNE (1 mM EDTA, 500 mM NaCl, 10 mM Tris–HCl, pH 8.0) for 10 min at 37°C, treated with 20 µg/ml RNase A (Nacalai Tesque) in TNE for 15 min at 37°C, washed 2× SSC twice for 15 min each at 37°C twice and 0.2× SSC twice for 15 min each at 37°C. After washing twice in a high stringency buffer (10 mM Tris, 10 mM EDTA, and 500 mM NaCl, pH 8.0) for 10 min

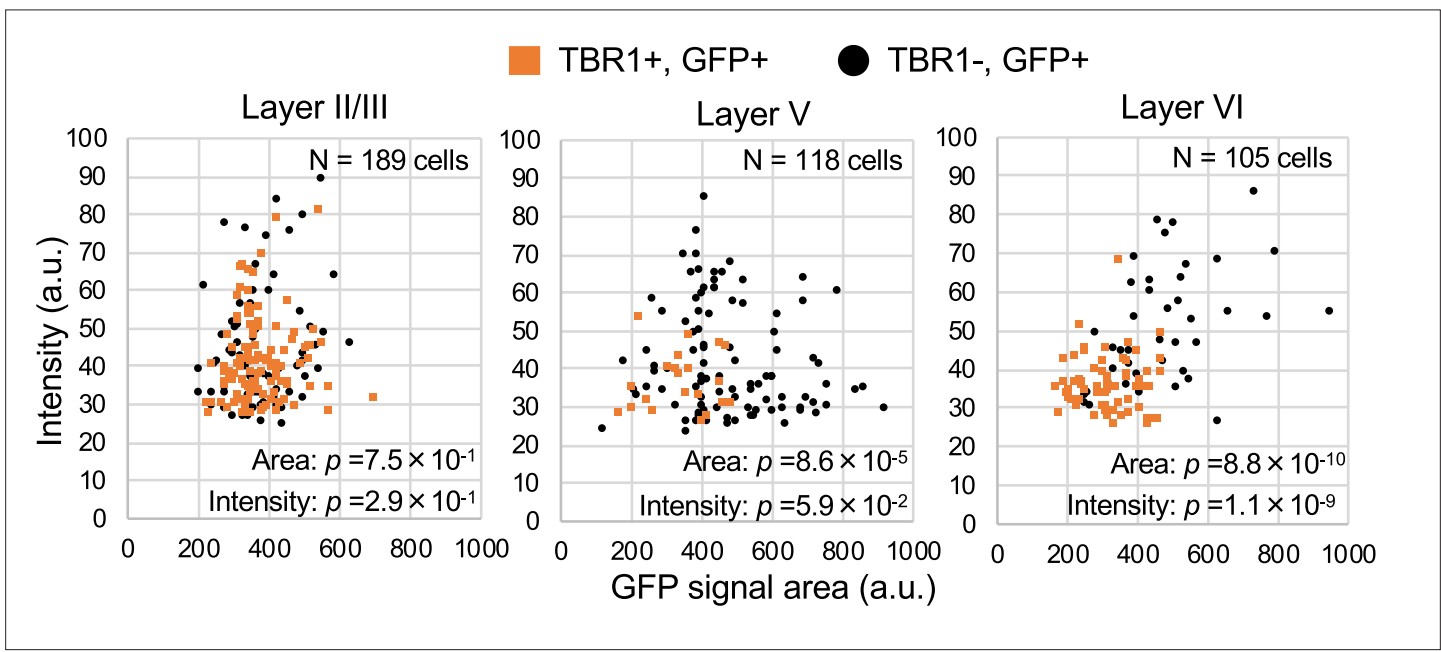

**Figure 15.** TBR1-positive cells have lower green fluorescent protein (GFP) signal intensities in *Scn1a*-GFP mouse neocortex. Scatter plots of intensities and area sizes of GFP immunosignals in GFP-positive cells with TBR1-positive/negative nuclei. Cells at primary motor cortex in parasagittal sections from 4W *Scn1a*-GFP mouse (line #233) were analyzed. TBR1-positive (orange squares) and negative (black circles) cells in neocortical L2/3, L5, and L6 are plotted (see also *Supplementary file 1aj*). Note that GFP signal intensities of TBR1/GFP-double positive cells were significantly lower than that of TBR1-negative/GFP-positive cells (L6), and signal area size of TBR1/GFP-double positive cells was significantly smaller than that of TBR1-negative/GFP-positive cells (L5, L6). Statistical significance was assessed using *t*-test. a.u., arbitrary unit; +, positive; −, negative.

The online version of this article includes the following source data for figure 15:

**Source data 1.** Numerical source data for *Figure 15*.

each at RT, the sections were blocked with a blocking buffer [20 mM Tris and 150 mM NaCl, pH 7.5 containing 0.05% Tween-20, 4% BlockAce (DS Pharma Biomedical) and 0.5× Blocking reagent (Roche Diagnostics)] for 1 hr at RT, and incubated with the alkaline phosphatase-conjugated sheep anti-DIG (1:500; 11093274910, Roche Diagnostics) and biotinylated rabbit anti-DNP (1:100; BA-0603, Vector Laboratories) antibodies in a blocking buffer overnight at 4 °C, followed by incubation with the biotinylated goat anti-rabbit antibody (1:200; BA-1000, Vector Laboratories) in a blocking buffer at RT for 1–2 hr. The probes were visualized using the NBT/BCIP kit (Roche Diagnostics), VECTASTAIN Elite ABC kit (Vector Laboratories), and ImmPACT DAB substrate (Vector Laboratories).

The DIG-labeled RNA probes for *Scn1a* designed to target the 3'-untranlated region (nucleotides 6488–7102 from accession number NM_001313997.1) were described previously (*Ogiwara et al., 2007*), and synthesized using the MEGAscript transcription kits (Thermo Fisher Scientific) with DIG-11-UTP (Roche Diagnostics). The DNP-labeled RNA probes for GFP were derived from the fragment corresponding to nucleotides 1256–1983 in pIRES2-EGFP (Takara Bio), and prepared using the MEGAscript transcription kits (Thermo Fisher Scientific) with DNP-11-UTP (PerkinElmer).

## Acknowledgements

We thank Dr. Yaguchi (Laboratory for Behavioral Genetics) and the staff members at the Research Resources Division of RIKEN Center for Brain Science for technical assistance in generating *Scn1a*-GFP BAC Tg mice and Dr. Kaneda (Nippon Medical School) for his support. This study was supported in part by MEXT/JSPS KAKENHI JP20H03566, JP17H01564, JP16H06276, JP23H02799, AMED JP18dm0107092, RIKEN Center for Brain Science (KY); Takeda Science Foundation, Kiyokun Foundation (IO and KY); MEXT/JSPS KAKENHI JP19790747, JP21791020, JP16K15564, JP19K08284 (IO); and Japan Epilepsy Research Foundation (IO and TT).

## Additional information

### Funding

| Funder | Grant reference number | Author |
| --- | --- | --- |
| Japan Society for the Promotion of Science | JP20H03566 | Kazuhiro Yamakawa |
| RIKEN Center for Brain Science | | Kazuhiro Yamakawa |
| Japan Agency for Medical Research and Development | JP18dm0107092 | Kazuhiro Yamakawa |
| Japan Society for the Promotion of Science | JP16H06276 | Kazuhiro Yamakawa |
| Ministry of Education, Culture, Sports, Science and Technology | JP17H01564 | Kazuhiro Yamakawa |
| Takeda Science Foundation | | Ikuo Ogiwara Kazuhiro Yamakawa |
| Kiyokun Foundation | | Ikuo Ogiwara Kazuhiro Yamakawa |
| Japan Society for the Promotion of Science | JP19790747 | Ikuo Ogiwara |
| Japan Society for the Promotion of Science | JP21791020 | Ikuo Ogiwara |
| Japan Society for the Promotion of Science | JP16K15564 | Ikuo Ogiwara |
| Japan Society for the Promotion of Science | JP19K08284 | Ikuo Ogiwara |

| Funder | Grant reference number | Author |
|--------|------------------------|--------|
| Japan Epilepsy Research Foundation | | Ikuo Ogiwara<br>Tetsuya Tatsukawa |
| Japan Society for the Promotion of Science | JP23H02799 | Kazuhiro Yamakawa |

The funders had no role in study design, data collection, and interpretation, or the decision to submit the work for publication.

## Author contributions

Tetsushi Yamagata, Conceptualization, Data curation, Formal analysis, Validation, Investigation, Visualization, Writing – original draft, Writing – review and editing; Ikuo Ogiwara, Data curation, Funding acquisition, Visualization, Writing – review and editing; Tetsuya Tatsukawa, Data curation, Funding acquisition, Visualization; Toshimitsu Suzuki, Data curation, Visualization, Writing – review and editing; Yuka Otsuka, Nao Imaeda, Emi Mazaki, Data curation, Visualization; Ikuyo Inoue, Natsuko Tokonami, Yurina Hibi, Data curation; Shigeyoshi Itohara, Resources, Supervision; Kazuhiro Yamakawa, Conceptualization, Supervision, Funding acquisition, Validation, Visualization, Writing – original draft, Project administration, Writing – review and editing

## Author ORCIDs

Ikuo Ogiwara http://orcid.org/0000-0003-4826-1456
Kazuhiro Yamakawa http://orcid.org/0000-0002-1478-4390

## Ethics

All animal experimental protocols were approved by the Animal Experiment Committee of Nagoya City University (#19-032) and RIKEN Center for Brain Science (W2019-1-005). Mice were handled in accordance with the guidelines of the Animal Experiment Committee.

## Decision letter and Author response

Decision letter https://doi.org/10.7554/eLife.87495.sa1
Author response https://doi.org/10.7554/eLife.87495.sa2

---

# Additional files

## Supplementary files

• Supplementary file 1. Tables of cell counting data for *Figures 3–5*, *Figures 7–15*, and *Figure 3—figure supplement 2*, *Figure 4—figure supplements 1 and 2*, *Figure 12—figure supplements 1–3*, *Figure 14—figure supplements 1–3*.

• MDAR checklist

## Data availability

All data generated or analyzed during this study are included in the manuscript and supporting file; Source Data files have been provided for Figure 1, 3–5, 7–15 and Figure 2—figure supplement 2, Figure 3—figure supplement 2, Figure 4—figure supplement 1, Figure 12—figure supplement 1–3, Figure 14—figure supplement 1–3.

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
