## [Editor Report]

Using a newly developed Scn1a promoter driven GFP mouse line, the authors convincingly show that GFP expression largely replicates the endogenous expression of Nav1.1. Additionally, they credibly identify inhibitory and excitatory neurons in the cortex that express Nav 1.1. This mouse line provides a valuable resource, especially for epilepsy and autism research, as it offers a reliable tool that can be used to identify specific cell populations that potentially cause disease-related symptoms such as seizures, ataxia, sociability deficits, learning and memory problems, and sudden unexpected death in epilepsy.

---

## [Decision Letter]

**Decision letter after peer review:**

Thank you for submitting your article "Scn1a-GFP transgenic mouse revealed Nav1.1 expression in neocortical pyramidal tract projection neurons" for consideration by *eLife*. Your article has been reviewed by 3 peer reviewers, one of whom is a member of our Board of Reviewing Editors, and the evaluation has been overseen by John Huguenard as the Senior Editor. The following individual involved in review of your submission has agreed to reveal their identity: Priscilla Kolibea Mante (Reviewer #2).

Essential revision:

1. For figure 1, please add a figure showing quantification for GFP and endogenous SCN1A protein levels in WT and BAC transgenic animals, with preferably more than four animals in each group. This quantification will strengthen the claim that endogenous levels of Scn1a are not altered (increased/decreased) in your BAC mouse.

2. To substantiate the statements of figure1E-F, please quantify and apply the relevant statistical analyses for the double-staining of endogenous Nav1.1 and Scn1a-GFP at the AISs, using robust sampling such as >40 AISs per region, and from multiple animals. This quantification can be added to the results proportion of the manuscript.

3. In the neocortex (Figure 2B, F, J), GFP-positive cells do not appear to be sparsely distributed throughout all cortical layers. Please include counts of the number of (GFP+ NeuN+/ NeuN) cells across the cortical layers for either the 184 or 233 BAC line.

4. The data in Figure 3 is not convincing as it is not a fair comparison of Nav1.1 and Nav1.2 expression in the AIS. Namely because GFP is mostly somatic while Nav1.2 is primarily in the AIS. Please provide quantification with relevant statistical analyses of Nav1.1 and Nav1.2 co-expression in AISs. One approach would be triple- immunostaining for Nav1.1, Nav1.2 and ankyrin, with appropriate samples sizes and replication.

5. The authors assume that the FEZF2 and TBR1 positive neurons that are GFP negative are Nav1.2 positive. Please apply triple-immunolabeling for (a) FEZF2, GFP and NAV1.2 and (b) TBR1, GFP and NAV1.2, with quantification and the relevant statistical analyses, to determine the proportion of FEZF2+ and TBR1+ neurons that are both GFP- and Nav1.2+ at either P15 or P28 for cortical layers 2/3, 5 and 6.

6. Please either remove Figure 6 or provide quantitative data to support the conclusions of the figure. This quantitative data may include a viral- based strategy to label, PT, CC(a-c), CT and iCS neuronal subsets. Followed by quantifications for each virally-labeled cell type for: Nav1.1+FEZF2+; Nav1.1-FEZF2+; Nav1.1+TBR1+; Nav1.1-TBR1+; Nav1.2+FEZF2+; Nav1.2-FEZF2+; Nav1.2+TBR1+; Nav1.2-TBR1+ neurons.

7. Please remove all references to how SUDEP might happen in Dravet Syndrome, including Figure 7. Although the hypothesis proposed is intriguing, this section weakens the paper, as it is purely speculative and not directly supported by the data presented in the manuscript.

8. For Supplemental figure 1, please add quantifications for the populations of PV+GFP+ and SST+GFP+ neurons in the neocortex and hippocampal regions analyzed in Supplementary figure 1. This data may provide quantitative support for the statement that all GFP cells in the hippocampus are inhibitory.

9. For Supplemental figure S2, please add to the figure, quantification and the relevant statistical analyses, for AIS quantifications of Nav 1.2+ ankyrin+ TBR1+, Nav 1.2- ankyrin+ TBR1+, Nav1.1+ ankyrin+ TBR1+, Nav1.1- ankyrin+ TBR1+ neurons.

10. Please specify in the methods section:

1. How many sections were reviewed per animal per age.

2. How many images per section.

3. Please clearly specify the brain region where sections and images originated.

4. Please indicate in each case whether the sections are sagittal or coronal.

11. Please review all the comments from the three manuscript referees to identify areas within the manuscript where grammatical errors and mislabeling of figures were observed. These errors must be corrected.

*Reviewer #1 (Recommendations for the authors):*

1. This manuscript contains numerous, very long sentences that are difficult to read and understand. Some examples are included below.

A. This sentence is too long.

Kalume and colleagues (2013) reported that sudden death in Nav1.1 haplo-deficient mice occurred immediately after generalized tonic-clonic seizures and ictal bradycardia or slower heart rate, that the cardiac and sudden death phenotypes were reproduced in mice with forebrain GABAergic neuron-specific, but not cardiac neuron-specific heterozygous elimination of Scn1a, and that the ictal bradycardia and sudden death were suppressed by atropine, a competitive antagonist for muscarinic acetylcholine receptors, and therefore counteracts against parasympathetic nervous system.

B. This sentence is too long.

Here in this study we revealed that Nav1.1 is expressed in PT neurons. Together with the above mentioned parasympathetic hyperactivity (Kalume et al., 2013) and the ameliorating effect of Nav1.1 haploinsufficiency in neocortical excitatory excitatory neurons (Ogiwara et al., 2013) observed in the sudden death of Nav1.1 haplo-deficient mice, the pathological neural circuit was assumed to be as follows, "Nav1.1 haploinsufficiency in neocortical PV-IN fails to suppress and therefore activates PT neurons as well as subsequent parasympathetic neurons and consequently suppresses heart activity and results in cardiac arrest" (Figure 7), in which Nav1.1 is remained in full amount in PT neurons of the mice with inhibitory neurons-specific haploelimination of Nav1.1 and therefore the lethality of the mice was further aggravated.

2. This manuscript contain some sentences that utilize the same words multiple times in one sentence. An example is included below.

A. In contrast, a major amount of Nav1.2 (~95%) is expressed in excitatory neurons including the major population of neocortical and all of hippocampal ones, and a minor amount is expressed in caudal ganglionic eminence (CGE)-derived inhibitory neurons such as vasoactive intestinal polypeptide (VIP)-positive ones (VIP-IN) (Lorincz and Nusser 2010; Yamagata et al., 2017; Ogiwara et al., 2018), however a recent study reported that a subpopulation (mostly half) of VIP-IN are Nav1.1-positive (Goff and Goldberg, 2019).

3. Please see additional suggested edits below.

A. Please edit the title to Figure 3 to state GFP-positive cells.

B. The word "mimicking" is misspelt in the sentence below.

We therefore performed immunohistochemical investigations of Nav1.1-mimicing GFP signals in Scn1a-GFP mice by using FEZF2 and TBR1 as projection neuron markers (Figure 4 and Figure 5).

C. Fluorescence imaging showed relatively intense GFP signals in the caudal brain portions (Figure 1D), which is well consistent with the previously reported regional distribution of Nav1.1 protein and Scn1a mRNA in wild-type mouse brain (Ogiwara et al., 2007).

D. In the sentence below, consider the use of another segue word than instead.

Instead, most AISs of TBR1-positive neurons are Nav1.2-positive (Supplemental figure S2).

E. The meaning of the sentence below is unclear and should be re-written.

A majority of neocortical excitatory projection neurons in L5 consist of pyramidal tract (PT) and intratelencephalic cortico-striatal (iCS) neurons, here we define iCS rather than CS because PT also occasionally innervate striatum, and those in L6 are mostly cortico-thalamic (CT) neurons (Shepherd, 2013).

Science Comments

1. In Figure 1B the β-tubulin blot does not appear to be a re-probe of the blot for Nav1.1 as the lanes for the β-tubulin blot are not clearly visible.

2. Using western blot or a similar technique, please quantify GFP and SCN1A protein levels in multiple (n>8-10) WT and BAC transgenic animals. This quantification will strengthen the claim that endogenous levels of Scn1a are not altered (increased/decreased) in your BAC mouse.

3. This statement is unfounded from the images.

"Immunohistochemical double-staining of Nav1.1 and GFP revealed Nav1.1-signals at axon initial segments (AISs) of most neocortical and hippocampal GFP-positive cells (Figure 1E, F)".

Please quantify the double-staining of Nav 1.1 and GFP at the AISs in a subset ( ~100 cells per region in 4-5 mice) of neocortical and hippocampal neurons.

4. Please include matched images i.e. at the same brain level for Figure 2A and Figure 2E.

5. The GFP-positive cells do not appear to be sparsely distributed throughout all cortical layers. Please include counts for the number of (GFP+ NeuN+/ NeuN) cells across the cortical layers for either the 184 or 233 S.

6. Only chromogenic staining seemed to be shown for both the 184 and 233 Scn1a-GFP lines. Thus, please re-write the sentence below:

The Scn1a-GFP lines #184 and 233 showed a similar distribution of GFP-signals, immuno-reactive and fluorescent signals as well, across the entire brain (Figure 1C, 2A-L).

as:

The Scn1a-GFP lines #184 and 233 showed a similar distribution of chromogenic GFP-signals across the entire brain (Figure 1C, 2A-D, 2E-2H).

7. Please replace this statement: These data indicate that GFP signals in the Scn1a-GFP lines faithfully reflect endogenous Scn1a/Nav1.1 expression.

with

These data indicate that GFP signals in the Scn1a-GFP lines may phenocopy endogenous Scn1a/Nav1.1 expression in the neocortex, hippocampus, and cerebellum.

8. The data in figure 3 is not convincing as it is difficult to distinguish the AIS of GFP- positive and non-GFP positive cells. Thus, it is unclear whether the AISs of GFP-positive cells are Nav1.2-negative.

9. In figures S1A, S1E, and S1AF, some cells with "dense" and "less intense" staining appear to be stained for GFP and an inhibitory marker. Likewise, some cells have prominent GFP staining but are not labeled for either SST or PV. To strengthen your claim, the data presented in supplementary figure S1 should include quantification of brain slices that are triple labeled for PV, SST and GFP. Cell counts should then be included for the fraction of GFP cells that are PV+ and SST+.

10. Objective and quantitative definitions for "dense" and "less" intense" GFP staining should be established. Next, the fraction of inhibitory, and excitatory cells that have "dense" and "less" intense" GFP staining should been quantified. Without such criteria, it is difficult to conclude that FEZF2 or TBR1 signals were found in cells with less intense GFP signals.

11. Please provide quantification to support the statement made in the manuscript that most AISs of TBR1-positive neurons are Nav1.2-positive.

*Reviewer #2 (Recommendations for the authors):*

There is an excessive use of abbreviations which makes the reading of both the results and the discussion quite difficult. This can be minimized as much as possible.

The arrow heads in Figure 3 do not clearly point towards what authors indicate they are pointing towards.

Figure 3 shows that Nav1.2 is not expressed in the axon initial segments of GRP-positive cells in the cortex and other brain regions. Indeed the authors state that data for the "other brain regions" are not shown but it will be great if they could mention of some of these brain regions.

The authors state that "most AIS of TBRI-positive neurons are Nav1.2 positive" and go on to show only one supplemental figure (Supplemental Figure S2). The conclusions drawn from Figure 5 and Figure 6 were supported by percentage expression patterns in Supplemental Table 1. The authors should include a similar supplementary table to support the data shown in Supplemental Figure S2, since a conclusive statement is being made from that figure.

There are a few grammatical errors that need addressing:

Page 5 Paragraph 1: "Human has nine alphas (Nav1.1~Nav1.9) and four betas (β-1~β-4)."

Results Paragraph 2: "….and GFP fluorescent signals are well negative in CA pyramidal and Dentate granule cells….."

Results Paragraph 4: "We therefore examined whether Nav1.2 is expressed in of cells expressing GFP and found that….."

Results, Final Paragraph: The first sentence needs a rewrite. The use of "mostly half" is inappropriate and does not lend to understanding. The last sentence also needs a rewrite "…..the remained Nav1.1-negative are assumed to be…"

Discussion: The last two paragraphs of the discussion could benefit from a rewrite which seeks to simplify the ideas being conveyed. These are really long sentences which reduce clarity of the statements being made. There are also multiple grammatical errors.

*Reviewer #3 (Recommendations for the authors):*

This new transgenic mice is very exciting and an important contribution to the field of epilepsy mechanisms.

My main suggestion for the manuscript is to remove the portion about how SUDEP might happen in Dravet Syndrome, including Figure 7. This section is purely speculative and is not necessary for the main objectives of the paper: Presenting the transgenic mouse model and further characterizing the excitatory cortical neurons in which Nav1.1 is expressed. I think the hypothesis proposed is intriguing, and as mentioned in the public review the transgenic mice will be useful in future experiments testing it. But right now I feel it weakens the paper by ending on a note that is not directly supported by the data presented in the manuscript (we don't see confirmation that the problem is cortical disinhibition and not dysfunction of hypothalamic neurons that also express Nav1.1 and would represent the bulk of input to the autonomic brainstem centers with the corticobulbar projections representing a minority and might even just go to the portion of the nucleus ambiguus in charge of pharyngeal muscles).

While I agree with the authors that statistical methods are not necessary for a characterization study, the methods section would benefit from additional details. How many sections were reviewed per animal per age? How many images per section? There is a mention in the Results section of the FEZF2 neurons being quantified in the primary motor cortex. Were all images taken in this region? Were the sections sagital or coronal?

The subheading "Nav1.1 is expressed in pyramidal tract while Nav1.2 in cortico-striatal and cortico-thalamic projection neurons" of the Results section would benefit from a quick review of the main projections from each layer at the top. Referring to figure 6 earlier would also make it easier for the reader to follow the results presented later in the section. Figure 6 is a great way to summarize the main points of the latter experiments by the way.

The authors assume that the FEZF2 and TBR1 positive neurons that are GFP negative are Nav1.2 positive. I assume this was not confirmed experimentally because the antibody for both transcription factors and Nav1.2 are all rabbit derived, but if other antibodies are available this would be an important point to confirm through IHC.

I would suggest doing a double labelling of PV and SST in Supplemental Figure 1 to show that all GFP cells in the hippocampus are inhibitory.

In Figure 1, it looks like the left arrowhead in the GFP panel of E is right on top of the neuron's AIS; unlike in the Merge panel where the neurons seems to be completely to the right of the arrowhead

In figure 2, the stratum pyramidale in CA1 and CA3 od panel C seems to have a lot of background and it is very hard to see the truly positive cells

In Figure 3, because the GFP is mostly somatic and Nav1.2/ankyrin G are located in the AIS, the current magnification of the figure makes it difficult to interpret. Consider using the same magnification as in Figure 1 panels E and F. With the current magnification it seems like one cell in the top left corner is positive for both GFP and Nav1.2

In the legend of Figure 5 for consistency keep the order of labeling consistent when describing the arrows and asterisks (i.e. mention the GFP status first in all of them). In the description of the asterisks it mentions CFP/TBR1-double positive cells. That CFP should be GFP.

In Supplemental Table S1 please add the conventions for the a, b, and c superscripts. For consistency remove the line between P15 and 4W in the L2/3 rows and consider adding a line between the L5 and L6 rows.

There are occasional grammar and misspelling issues throughout the manuscript.

---

## [Author Response]

Essential revision:1. For figure 1, please add a figure showing quantification for GFP and endogenous SCN1A protein levels in WT and BAC transgenic animals, with preferably more than four animals in each group. This quantification will strengthen the claim that endogenous levels of Scn1a are not altered (increased/decreased) in your BAC mouse.

In response to the editor’s comment, we performed western blot quantification to reveal protein levels of Nav1.1 and GFP in the brains of five *Scn1a*-GFP and five wild-type mice. As a result, we were able to confirm that there was no change in the amount of Nav1.1 in *Scn1a*-GFP mouse compared to the wild type (Supplementary figure S3A). However, we detected up to a 3-fold range of change in the amount of GFP protein, the reason is so far unknown (Supplementary figure S3B).

Accordingly, we revised the manuscript as follows;

Lines #129-132 at the Results section;

“Quantification of Nav1.1 signals in western blot analyses of brain lysates from the *Scn1a*-GFP mice and their wild-type littermates (N = 5 animals per each genotype) showed no difference between genotypes, while that of GFP somehow deviated among individual *Scn1a*-GFP mice (Supplementary figure S3).”

2. To substantiate the statements of figure1E-F, please quantify and apply the relevant statistical analyses for the double-staining of endogenous Nav1.1 and Scn1a-GFP at the AISs, using robust sampling such as >40 AISs per region, and from multiple animals. This quantification can be added to the results proportion of the manuscript.

In response to the editor’s comment, we performed triple immunostaining using anti-Nav1.1, anti-GFP and anti-ankyrinG antibodies and quantified Nav1.1-positive AISs (>40) and somata in cerebral motor cortex and hippocampus of three *Scn1a*-GFP mice (Figures 4 and 5).

Accordingly, we revised the manuscript as follows;

Lines #161-170 at the Results section;

“Next, we performed triple-immunostaining of Nav1.1, GFP and ankyrinG on brains of *Scn1a*-GFP mouse at P15. In the neocortex (Figure 4), axon initial segments (AISs) of cells with Nav1.1-positive somata were always Nav1.1-positive but somata of cells with Nav1.1-positive AISs were occasionally Nav1.1-negative (Figures 4A-C). Cell counting revealed that 17% (L2/3), 21% (L5) and 8% (L6) of neurons (cells with ankyrinG-positive AISs) were GFP-positive (Figure 4D-left panel and Supplementary table S2). Of note, all cells with Nav1.1-positive AISs or somata were GFP-positive, but AISs or somata for only half of GFP-positive cells were Nav1.1-positive (Figure 4D and Supplementary tables S3, S4), possibly due to undetectably low levels of Nav1.1 immunosignals in a subpopulation of GFP-positive cells.”

Lines #181-189 at the Results section;

“In contrast to the neocortex where only half of GFP-positive cells were Nav1.1-positive, in the hippocampus all GFP-positive cells were Nav1.1-positive and all Nav1.1-positive cells were GFP-positive (Figure 5). Actually, most of excitatory neurons such as CA1~3 pyramidal cells and dentate granule cells were GFP-negative. As described above (Figures 2C, G), fibrous GFP and Nav1.1 signals twining around CA1~3 pyramidal cells' somata which are assumed to be axon terminals of PV-INs were again observed (Figures 5A, B, D). Cell counting in the hippocampal CA1 region showed that 98% of cells with GFP-positive somata were Nav1.1-positive at their AISs and 100% of cells with Nav1.1-positive AISs were GFP-positive (Figure 5F and Supplementary tables S7, S8).”

Lines #362-372 at the Discussion section;

“In our present study, we developed *Scn1a* promoter-driven GFP mice in which the expression of GFP mimics that of Nav1.1. All PV-INs and most of SST-INs were GFP-positive in the neocortex and hippocampus of the *Scn1a*-GFP mouse, being consistent with the previous reports of Nav1.1 expression in those inhibitory neurons (Ogiwara et al., 2007; Lorincz and Nusser, 2008; Ogiwara et al., 2013; Li et al., 2014; Tai et al., 2014; Tian et al., 2014; Yamagata et al., 2017). All Nav1.1-positive cells were GFP-positive. Reversely all GFP-positive cells were also Nav1.1-positive in the hippocampus, but in the neocortex only a half of GFP-positive cells were Nav1.1-positive. This is largely because in the hippocampus Nav1.1 expression is restricted to inhibitory neurons, but in the neocortex Nav1.1 is expressed not only in inhibitory but also in a subpopulation of excitatory neurons in which Nav1.1 expression is low and not easily detected immunohistochemically by anti-Nav1.1 antibodies.”

Lines #520-534 at the Materials and methods section;

"Fluorescence and Immunofluorescence quantification

For quantification of inhibitory neurons in GFP-positive cells, we used *Scn1a*-GFP/*Vgat*-Cre/*Rosa26*-tdTomamto mice at 4W. We acquired multiple color images of primary motor cortex and hippocampus from three parasagittal sections per animal. Six images per region of interest were manually counted and summarized using Adobe Photoshop Elements 10 and Excel (Microsoft). On immunofluorescence quantification, we used *Scn1a*-GFP mice at P15 and/or 4W for the quantification of immunosignals. For quantification of GFP, NeuN, PV, SST, Nav1.1, Nav1.2, FEZF2 or TBR1-positive cells, we acquired multiple color images of primary motor cortex and hippocampus from three parasagittal sections per animal. Six~nine images per region of interest were manually quantified and summarized. For quantification of PV, SST, FEZF2 or TBR1-positive cells, intensity and area size of GFP fluorescent signals were measured by Fiji software. Statistical analyses were performed by one-way ANOVA followed by Tukey–Kramer post-hoc multiple comparison test using Kyplot 6.0 (KyensLab Inc). P-value smaller than 0.05 was considered statistically significant. Data are presented as the mean ± standard error of the mean (SEM).”

3. In the neocortex (Figure 2B, F, J), GFP-positive cells do not appear to be sparsely distributed throughout all cortical layers. Please include counts of the number of (GFP+ NeuN+/ NeuN) cells across the cortical layers for either the 184 or 233 BAC line.

In response to the editor’s comment, we performed immunostaining of NeuN using *Scn1a*-GFP mouse line #233 and counted cell numbers (Supplementary figure S5 and Supplementary table S1). In the neocortex, 20~30% of NeuN-positive cells were GFP-positive. However, we noticed that sparsely-distributed cells with intense GFP-signals, which are PV-INs, were often NeuN-negative. NeuN-positive cells therefore do not represent all neurons in neocortex and above figure should deviate from the real ratios of GFP-positive cells among all neurons. Instead, we estimated the ratio of GFP-positive cells among all neurons in the experiments of GFP and ankyrinG stainings (see Figures 4, 11, S6, S7).

Accordingly, we revised the manuscript as follows;

Lines #149-180 at the Results section;

“In order to know the ratio of GFP-positive cells among all neurons, we further performed immunohistochemical staining using NeuN-antibody on *Scn1a*-GFP mouse at P15 and cells were counted at M1 and S1 (Supplementary figure S5 and Supplementary table S1). The NeuN staining showed that GFP-positive cells occupy 30% (L2/3), 32% (L5) and 22% (L6) of NeuN- or GFP-positive cells at P15 (Supplementary figure S5B and Supplementary table S1). However, we noticed that sparsely-distributed cells with intense GFP-signals, which are assumed to be PV-INs (see Figure 8), were often NeuN-negative (Supplementary figure S5A – arrowheads), reminiscent of a previous report that NeuN expression is absent in cerebellar inhibitory neurons such as Golgi, basket and satellite cells in cerebellum (Weyer et al., 2003). Therefore, NeuN-positive cells do not represent all neurons in neocortex as well. NeuN/GFP-double negative neurons could even exist and therefore above figure (Supplementary figure S5B) may deviate from the real ratios of GFP-positive cells among all neurons.

Next, we performed triple-immunostaining of Nav1.1, GFP and ankyrinG on brains of *Scn1a*-GFP mouse at P15. In the neocortex (Figure 4), axon initial segments (AISs) of cells with Nav1.1-positive somata were always Nav1.1-positive but somata of cells with Nav1.1-positive AISs were occasionally Nav1.1-negative (Figures 4A-C). Cell counting revealed that 17% (L2/3), 21% (L5) and 8% (L6) of neurons (cells with ankyrinG-positive AISs) were GFP-positive (Figure 4D-left panel and Supplementary table S2). Of note, all cells with Nav1.1-positive AISs or somata were GFP-positive, but AISs or somata for only half of GFP-positive cells were Nav1.1-positive (Figure 4D and Supplementary tables S3, S4), possibly due to undetectably low levels of Nav1.1 immunosignals in a subpopulation of GFP-positive cells. The above ratios of GFP-positive cells among neurons (cells with ankyrinG-positive AISs) obtained in the triple-immunostaining of Nav1.1, GFP and ankyrinG are rather discordant to those obtained in the later experiment of triple-immunostaining of Nav1.2, GFP and ankyrinG, 23% (L2/3), 30% (L5) and 21% (L6) (see Figure 11). Therefore, we additionally performed double-immunostaining of GFP and ankyrinG on brains of *Scn1a*-GFP mouse at P15, and the ratios of of GFP-positive cells among neurons were 30% (L2/3), 26% (L5) and 9% (L6) (Supplementary figure S6 and Supplementary table S5). Averaged ratios of GFP-positive cells among neurons of Figures 4D, 11B and Supplementary figure S6B are 23% (L2/3), 26% (L5) and 13% (L6) (Supplementary figure S7 and Supplementary table S6), which are actually significantly lower than those obtained in the NeuN-staining (Supplementary figure S5 and Supplementary table S1).”

Line #502-504 at the Materials and methods section;

"Following antibodies were used to detect GFP, Nav1.1, Nav1.2, TBR1, FEZF2, ankyrinG, NeuN, parvalbumin and somatostatin; …”

Line #510-511 at the Materials and methods section;

"… mouse anti-NeuN biotin conjugated antibody (1:2,000; MAB377B, Millipore), …”

Line #515-516 at the Materials and methods section;

"To detect NeuN, Alexa-647 conjugated streptavidin (1:1,000; S21374, Thermo Fisher Science) was used."

Lines #520-534 at the Materials and methods section;

"Fluorescence and Immunofluorescence quantification

For quantification of inhibitory neurons in GFP-positive cells, we used *Scn1a*-GFP/*Vgat*-Cre/*Rosa26*-tdTomamto mice at 4W. We acquired multiple color images of primary motor cortex and hippocampus from three parasagittal sections per animal. Six images per region of interest were manually counted and summarized using Adobe Photoshop Elements 10 and Excel (Microsoft). On immunofluorescence quantification, we used *Scn1a*-GFP mice at P15 and/or 4W for the quantification of immunosignals. For quantification of GFP, NeuN, PV, SST, Nav1.1, Nav1.2, FEZF2 or TBR1-positive cells, we acquired multiple color images of primary motor cortex and hippocampus from three parasagittal sections per animal. Six~nine images per region of interest were manually quantified and summarized. For quantification of PV, SST, FEZF2 or TBR1-positive cells, intensity and area size of GFP fluorescent signals were measured by Fiji software. Statistical analyses were performed by one-way ANOVA followed by Tukey–Kramer post-hoc multiple comparison test using Kyplot 6.0 (KyensLab Inc). P-value smaller than 0.05 was considered statistically significant. Data are presented as the mean ± standard error of the mean (SEM)."

4. The data in Figure 3 is not convincing as it is not a fair comparison of Nav1.1 and Nav1.2 expression in the AIS. Namely because GFP is mostly somatic while Nav1.2 is primarily in the AIS. Please provide quantification with relevant statistical analyses of Nav1.1 and Nav1.2 co-expression in AISs. One approach would be triple- immunostaining for Nav1.1, Nav1.2 and ankyrin, with appropriate samples sizes and replication.

In response to the editor’s comment, we performed triple-immunostaining for Nav1.1, Nav1.2 and ankyrin G (Figure 10 and Supplementary table S17). The analysis revealed that Nav1.1/Nav1.2 double-positive AISs were less than 0.5% of ankyrin G-positive AIS, and most of AIS were Nav1.2-positive in neocortex.

Accordingly, we revised the manuscript as follows;

Lines #237-262 at the Results section;

“Nav1.1 and Nav1.2 expressions are mutually exclusive in mouse brain

We previously reported that expressions of Nav1.1 and Nav1.2 seem to be mutually-exclusive in multiple brain regions including neocortex, hippocampal CA1, dentate gyrus, striatum, globus pallidus, and cerebellum in wild-type mice (Yamagata et al., 2017). To further confirm it, here we performed triple immunostaining for Nav1.1, Nav1.2 and ankyrinG, and counted Nav1.1- or Nav1.2-immunopositive AISs in the neocortex of *Scn1a*-GFP mice (Figure 10). The staining again showed that Nav1.1 and Nav1.2 expressions are mutually exclusive in brain regions including neocortex (Figure 10A). The counting revealed that 5% (L2/3), 6% (L5) and 3% (L6) of AISs at P15 were Nav1.1-positive and 78% (L2/3), 69% (L5) and 69% (L6) of AISs at P15 were Nav1.2-positive in the neocortex (Figure 10B and Supplementary table S17). Of note, less than 0.5% of AISs are Nav1.1/Nav1.2-double positive confirming that Nav1.1 and Nav1.2 do not co-express. These results are consistent with our previous study (Yamagata et al., 2017) and further support that expressions of Nav1.1 and Nav1.2 are mutually exclusive in mouse neocortex at least at immunohistochemical level.

As mentioned, GFP signals in *Scn1a*-GFP mouse can represent even moderate or low Nav1.1 expressions which cannot be detected by immunohistochemical staining, so some of GFP-positive cells may still express Nav1.2. To investigate whether and if so how much of GFP-positive cells have Nav1.2-positive AISs in *Scn1a*-GFP mouse neocortex, we performed triple immunohistochemical staining for Nav1.2, GFP and ankyrinG (Figure 11 and Supplementary tables S18-S20). The staining showed that AISs of GFP-positive cells are largely negative for Nav1.2, and cells with Nav1.2-positive AISs are mostly GFP-negative (Figure 11A). Cell counting revealed that 88% (L2/3), 90% (L5) and 95% (L6) of cells with Nav1.2-positive AISs at P15 were GFP-negative (Figure 11B-middle panel and Supplementary table S19), and 69% (L2/3), 83% (L5), and 86% (L6) of AISs of GFP-positive cells at P15 were Nav1.2-negative (Figure 11B-right panel and Supplementary table S20). These results indicate that the co-expression of GFP and Nav1.2 would be minimal if any.”

Lines #520-534 at the Materials and methods section;

"Fluorescence and Immunofluorescence quantification

For quantification of inhibitory neurons in GFP-positive cells, we used *Scn1a*-GFP/*Vgat*-Cre/*Rosa26*-tdTomamto mice at 4W. We acquired multiple color images of primary motor cortex and hippocampus from three parasagittal sections per animal. Six images per region of interest were manually counted and summarized using Adobe Photoshop Elements 10 and Excel (Microsoft). On immunofluorescence quantification, we used *Scn1a*-GFP mice at P15 and/or 4W for the quantification of immunosignals. For quantification of GFP, NeuN, PV, SST, Nav1.1, Nav1.2, FEZF2 or TBR1-positive cells, we acquired multiple color images of primary motor cortex and hippocampus from three parasagittal sections per animal. Six ~ nine images per region of interest were manually quantified and summarized. For quantification of PV, SST, FEZF2 or TBR1-positive cells, intensity and area size of GFP fluorescent signals were measured by Fiji software. Statistical analyses were performed by one-way ANOVA followed by Tukey–Kramer post-hoc multiple comparison test using Kyplot 6.0 (KyensLab Inc). P-value smaller than 0.05 was considered statistically significant. Data are presented as the mean ± standard error of the mean (SEM)."

5. The authors assume that the FEZF2 and TBR1 positive neurons that are GFP negative are Nav1.2 positive. Please apply triple-immunolabeling for (a) FEZF2, GFP and NAV1.2 and (b) TBR1, GFP and NAV1.2, with quantification and the relevant statistical analyses, to determine the proportion of FEZF2+ and TBR1+ neurons that are both GFP- and Nav1.2+ at either P15 or P28 for cortical layers 2/3, 5 and 6.

In response to the editor’s comment, we performed triple immunostaining for FEZF2, GFP and Nav1.2 (Supplementary figure S10 and Supplementary table S32) and that for TBR1, GFP and Nav1.2 (Supplementary figure S13 and Supplementary table S44) using three *Scn1a*-GFP mice at P15. These experiments together with additional new data revealed more detailed relationships between GFP (Nav1.1), Nav1.2, FEZF2 and TBR1.

Accordingly, we revised the manuscript as follows;

Lines #264-359 at the Results section;

"Neocortical pyramidal tract and a subpopulation of cortico-cortical projection neurons express Nav1.1

Neocortical excitatory neurons can be divided into functionally distinct subpopulations, a majority of those are pyramidal cells which have axons of long-range projections such as L2/3 CC, L5 PT, L5/6 CS and L6 CT projection neurons (Shepherd, 2013). […] As a whole, above results showed that a minor subpopulation of L2/3 CC and L5 PT neurons express Nav1.1 while the majority of L2/3 CC, L5/6 CS and L6 CT neurons express Nav1.2. A breakdown of L5 neuron species containing PT neurons is specifically described (Supplementary figure S14).”

Lines #520-534 at the Materials and methods section;

"Fluorescence and Immunofluorescence quantification

For quantification of inhibitory neurons in GFP-positive cells, we used *Scn1a*-GFP/*Vgat*-Cre/*Rosa26*-tdTomamto mice at 4W. We acquired multiple color images of primary motor cortex and hippocampus from three parasagittal sections per animal. Six images per region of interest were manually counted and summarized using Adobe Photoshop Elements 10 and Excel (Microsoft). On immunofluorescence quantification, we used *Scn1a*-GFP mice at P15 and/or 4W for the quantification of immunosignals. For quantification of GFP, NeuN, PV, SST, Nav1.1, Nav1.2, FEZF2 or TBR1-positive cells, we acquired multiple color images of primary motor cortex and hippocampus from three parasagittal sections per animal. Six~nine images per region of interest were manually quantified and summarized. For quantification of PV, SST, FEZF2 or TBR1-positive cells, intensity and area size of GFP fluorescent signals were measured by Fiji software. Statistical analyses were performed by one-way ANOVA followed by Tukey–Kramer post-hoc multiple comparison test using Kyplot 6.0 (KyensLab Inc). P-value smaller than 0.05 was considered statistically significant. Data are presented as the mean ± standard error of the mean (SEM).”

6. Please either remove Figure 6 or provide quantitative data to support the conclusions of the figure. This quantitative data may include a viral- based strategy to label, PT, CC(a-c), CT and iCS neuronal subsets. Followed by quantifications for each virally-labeled cell type for: Nav1.1+FEZF2+; Nav1.1-FEZF2+; Nav1.1+TBR1+; Nav1.1-TBR1+; Nav1.2+FEZF2+; Nav1.2-FEZF2+; Nav1.2+TBR1+; Nav1.2-TBR1+ neurons.

We agree with the editor’s comment, and we removed the original Figure 6.

7. Please remove all references to how SUDEP might happen in Dravet Syndrome, including Figure 7. Although the hypothesis proposed is intriguing, this section weakens the paper, as it is purely speculative and not directly supported by the data presented in the manuscript.

We agree with the editor’s comment, and therefore we removed the original Figure 7 and speculative hypothesis of neural circuit for SUDEP in Dravet syndrome in the discussion.

Accordingly, we revised the manuscript as follows;

Lines 381-429 at the Discussion section;

“The above observations should contribute to understanding of neural circuits responsible for diseases such as epilepsy and neurodevelopmental disorders caused by *SCN1A* and *SCN2A* mutations. […] Further developments of transgenic mice for other sodium channel genes' promoter-driven reporter molecules and combinatorial analyses of those mice are awaited to segregate and redefine their unique functional roles in each of highly diverse neuronal species and complexed neural circuits.”

8. For Supplemental figure 1, please add quantifications for the populations of PV+GFP+ and SST+GFP+ neurons in the neocortex and hippocampal regions analyzed in Supplementary figure 1. This data may provide quantitative support for the statement that all GFP cells in the hippocampus are inhibitory.

In response to the editor’s comment, we counted the number of GFP-positive cells with PV-positive and SST-positive soma in neocortex and hippocampus (Figure 8 and Supplementary tables S11-13). In addition, to investigate the ratio of inhibitory neurons in GFP-positive cells or vice versa more accurately we further generated and examined *Scn1a*-GFP and vesicular GABA transporter (*Vgat*)-Cre double transgenic mice in which *Vgat*-Cre is expressed in all GABAergic inhibitory neurons (Figure 7 and Supplementary tables S9 and S10). These experiments surely verified that all GFP cells in the hippocampus were inhibitory.

Accordingly, we revised the manuscript as follows;

Lines #198-235 at the Results section;

“To investigate the ratio of inhibitory neurons in GFP-positive cells, we generated and examined *Scn1a*-GFP and vesicular GABA transporter (*Vgat*)-Cre (Ogiwara et al., 2013) double transgenic mice in which *Vgat*-Cre is expressed in all GABAergic inhibitory neurons and visualized by floxed tdTomato transgene (Figure 7). […] These results indicate that Nav1.1 expression level in PV-INs is significantly higher than those in excitatory neurons and PV-negative GABAergic neurons including SST-INs.”

Lines #458-464 at the Materials and methods section;

"*Vgat*-Cre BAC transgenic mice and loxP flanked transcription terminator cassette CAG promotor driven tdTomato transgenic (*Rosa26*-tdTomato; B6.Cg- Gt(ROSA)26Sortm14(CAG-tdTomato)Hze/J, Stock No: 007914, The Jackson Laboratory, USA) mice were maintained on a C57BL/6J background. To generate triple mutant mice (*Scn1a*-GFP^gfp/-^, *Vgat*-Cre^cre/-^, *Rosa26*-tdTomamto^tomato/-^), heterozygous *Scn1a*-GFP and *Vgat*-Cre mice were mated with homozygous *Rosa26*-tdTomato transgenic mice.”

Lines #520-534 at the Materials and methods section;

"Fluorescence and Immunofluorescence quantification

For quantification of inhibitory neurons in GFP-positive cells, we used *Scn1a*-GFP/*Vgat*-Cre/*Rosa26*-tdTomamto mice at 4W. We acquired multiple color images of primary motor cortex and hippocampus from three parasagittal sections per animal. Six images per region of interest were manually counted and summarized using Adobe Photoshop Elements 10 and Excel (Microsoft). On immunofluorescence quantification, we used *Scn1a*-GFP mice at P15 and/or 4W for the quantification of immunosignals. For quantification of GFP, NeuN, PV, SST, Nav1.1, Nav1.2, FEZF2 or TBR1-positive cells, we acquired multiple color images of primary motor cortex and hippocampus from three parasagittal sections per animal. Six~nine images per region of interest were manually quantified and summarized. For quantification of PV, SST, FEZF2 or TBR1-positive cells, intensity and area size of GFP fluorescent signals were measured by Fiji software. Statistical analyses were performed by one-way ANOVA followed by Tukey–Kramer post-hoc multiple comparison test using Kyplot 6.0 (KyensLab Inc). P-value smaller than 0.05 was considered statistically significant. Data are presented as the mean ± standard error of the mean (SEM).”

9. For Supplemental figure S2, please add to the figure, quantification and the relevant statistical analyses, for AIS quantifications of Nav 1.2+ ankyrin+ TBR1+, Nav 1.2- ankyrin+ TBR1+, Nav1.1+ ankyrin+ TBR1+, Nav1.1- ankyrin+ TBR1+ neurons.10. Please specify in the methods section:

In response to the editor’s comment, we counted cell numbers in a newly-performed triple staining of TBR1, Nav1.1 and ankyrinG (Supplementary figure S11 and Supplementary tables S37-39) and that in the triple staining of TBR1, Nav1.2 and ankyrinG (Supplementary figure S12 and Supplementary tables S40-43).

Accordingly, we revised the manuscript as follows;

Lines #336-350 at the Results section;

“We additionally performed triple immunostaining for TBR1, Nav1.1, and ankyrinG on *Scn1a*-GFP mice (Supplementary figure S11 and Supplementary tables S37-S39). Notably, the ratios of Nav1.1-positive cells among TBR1-positive cells are 0% in all layers (Supplementary figure S11B-right panel and Supplementary table S39). These results indicate that the major population of TBR1-positive cells do not express Nav1.1.

To investigate whether TBR1-positive cells express Nav1.2, we performed triple immunostaining of TBR1, Nav1.2 and ankyrinG on *Scn1a*-GFP mice (Supplementary figure S12 and Supplementary tables S40-S43). In L5, contrary to the low ratios of GFP-positive cells among TBR1-positive cells (11% at P15 and 5% at 4W) (Figure 14B-middle panels and Supplementary table S34), the ratios of Nav1.2-positive cells among TBR1-positeve cells are high (69% (L2/3), 69%(L5) and 69% (L6) at P15) (Supplementary figure S12B-right-upper panel and Supplementary table S42). The ratios of TBR1-positive cells among Nav1.2-positive cells are 29% (L2/3), 53% (L5) and 62% (L6) at P15 (Supplementary figure S12B-middle upper panel and Supplementary table S41).”

Lines #520-534 at the Materials and methods section;

"Fluorescence and Immunofluorescence quantification

For quantification of inhibitory neurons in GFP-positive cells, we used *Scn1a*-GFP/*Vgat*-Cre/*Rosa26*-tdTomamto mice at 4W. We acquired multiple color images of primary motor cortex and hippocampus from three parasagittal sections per animal. Six images per region of interest were manually counted and summarized using Adobe Photoshop Elements 10 and Excel (Microsoft). On immunofluorescence quantification, we used *Scn1a*-GFP mice at P15 and/or 4W for the quantification of immunosignals. For quantification of GFP, NeuN, PV, SST, Nav1.1, Nav1.2, FEZF2 or TBR1-positive cells, we acquired multiple color images of primary motor cortex and hippocampus from three parasagittal sections per animal. Six~nine images per region of interest were manually quantified and summarized. For quantification of PV, SST, FEZF2 or TBR1-positive cells, intensity and area size of GFP fluorescent signals were measured by Fiji software. Statistical analyses were performed by one-way ANOVA followed by Tukey–Kramer post-hoc multiple comparison test using Kyplot 6.0 (KyensLab Inc). P-value smaller than 0.05 was considered statistically significant. Data are presented as the mean ± standard error of the mean (SEM).”

1. How many sections were reviewed per animal per age.2. How many images per section.3. Please clearly specify the brain region where sections and images originated.4. Please indicate in each case whether the sections are sagittal or coronal.

In response to the editor’s comment, we revised the manuscript as follows;

Lines #520-534 at the Materials and methods section;

"Fluorescence and Immunofluorescence quantification

For quantification of inhibitory neurons in GFP-positive cells, we used *Scn1a*-GFP/*Vgat*-Cre/*Rosa26*-tdTomamto mice at 4W. We acquired multiple color images of primary motor cortex and hippocampus from three parasagittal sections per animal. Six images per region of interest were manually counted and summarized using Adobe Photoshop Elements 10 and Excel (Microsoft). On immunofluorescence quantification, we used *Scn1a*-GFP mice at P15 and/or 4W for the quantification of immunosignals. For quantification of GFP, NeuN, PV, SST, Nav1.1, Nav1.2, FEZF2 or TBR1-positive cells, we acquired multiple color images of primary motor cortex and hippocampus from three parasagittal sections per animal. Six~nine images per region of interest were manually quantified and summarized. For quantification of PV, SST, FEZF2 or TBR1-positive cells, intensity and area size of GFP fluorescent signals were measured by Fiji software. Statistical analyses were performed by one-way ANOVA followed by Tukey–Kramer post-hoc multiple comparison test using Kyplot 6.0 (KyensLab Inc). P-value smaller than 0.05 was considered statistically significant. Data are presented as the mean ± standard error of the mean (SEM).”

Lines #491-493 at the Materials and methods section;

"For fluorescent imaging, PFA-fixed brains were cryoprotected with 30% sucrose in PBS, cut in 30 µm parasagittal sections, and mounted on glass slides."

Lines #495-497 at the Materials and methods section;

"For immunostaining, frozen parasagittal sections (30 µm) were blocked with 4% BlockAce (DS Pharma Biomedical) in PBS for 1 hour at room temperature (RT), and incubated with rat anti-GFP (1:500; GF090R, Nacalai Tesque)."

Lines #499-502 at the Materials and methods section;

"The antibody-antigen complexes were visualized using the Vectastain Elite ABC kit (Vector Laboratories) with Metal enhanced DAB substrate (34065, PIERCE). For immunofluorescent staining, we prepared 6 µm parasagittal sections from paraffin embedded PLP-fixed brains of mice."

Lines #502-516 at the Materials and methods section;

"Following antibodies were used to detect GFP, Nav1.1, Nav1.2, TBR1, FEZF2, ankyrinG, NeuN, parvalbumin and somatostatin; mouse anti-GFP antibodies (1:500; 11814460001, Roche Diagnostics), anti-Nav1.1 antibodies (1:10,000; rabbit IO1, 1:500; goat SC-16031, Santa Cruz Biotechnology), anti-Nav1.2 antibodies (1:1,000; rabbit ASC-002, Alomone Labs; goat SC-31371, Santa Cruz Biotechnology), rabbit anti-TBR1 antibody (1:1,000; ab31940, Abcam or 1:500; SC-376258, Santa Cruz Biotechnology), rabbit anti-FEZF2 antibody (1:500; #18997, IBL), ankyrinG antibodies (1:500; mouse SC-12719, rabbit SC-28561; goat, SC-31778, Santa Cruz Biotechnology), mouse anti-NeuN biotin conjugated antibody (1:2,000; MAB377B, Millipore), rabbit anti-parvalbumin (1:5,000; PC255L, Merck) and rabbit anti-somatostatin (1:5,000; T-4103, Peninsula Laboratories, 1:1,000; SC-7819, Santa Cruz Biotechnology) antibodies. As secondary antibodies, Alexa Fluor Plus 488, 555, 594 and 647 conjugated antibodies (1:1,000; A32723, A32766, A32794, A32754, A32849, A32795, A32787, Thermo Fisher Scientific) were used. To detect NeuN, Alexa-647 conjugated streptavidin (1:1,000; S21374, Thermo Fisher Science) was used.”

11. Please review all the comments from the three manuscript referees to identify areas within the manuscript where grammatical errors and mislabeling of figures were observed. These errors must be corrected.

We have corrected grammatical errors and mislabeling of figures indicated by the reviewers and revised the manuscript.

Reviewer #1 (Recommendations for the authors):1. This manuscript contains numerous, very long sentences that are difficult to read and understand. Some examples are included below.A. This sentence is too long.Kalume and colleagues (2013) reported that sudden death in Nav1.1 haplo-deficient mice occurred immediately after generalized tonic-clonic seizures and ictal bradycardia or slower heart rate, that the cardiac and sudden death phenotypes were reproduced in mice with forebrain GABAergic neuron-specific, but not cardiac neuron-specific heterozygous elimination of Scn1a, and that the ictal bradycardia and sudden death were suppressed by atropine, a competitive antagonist for muscarinic acetylcholine receptors, and therefore counteracts against parasympathetic nervous system.

In response to the reviewer's comment, we have revised the sentence as follows;

Lines #406-409 at the Discussion section;

“Kalume and colleagues (2013) reported that parasympathetic hyperactivity is observed in Nav1.1 haplo-deficient mice and it causes ictal bradycardia and finally result in seizure-associated sudden death.”

B. This sentence is too long.Here in this study we revealed that Nav1.1 is expressed in PT neurons. Together with the above mentioned parasympathetic hyperactivity (Kalume et al., 2013) and the ameliorating effect of Nav1.1 haploinsufficiency in neocortical excitatory excitatory neurons (Ogiwara et al., 2013) observed in the sudden death of Nav1.1 haplo-deficient mice, the pathological neural circuit was assumed to be as follows, "Nav1.1 haploinsufficiency in neocortical PV-IN fails to suppress and therefore activates PT neurons as well as subsequent parasympathetic neurons and consequently suppresses heart activity and results in cardiac arrest" (Figure 7), in which Nav1.1 is remained in full amount in PT neurons of the mice with inhibitory neurons-specific haploelimination of Nav1.1 and therefore the lethality of the mice was further aggravated.

In response to the editor's comment (#7), we removed this sentence in the revised manuscript.

2. This manuscript contain some sentences that utilize the same words multiple times in one sentence. An example is included below.A. In contrast, a major amount of Nav1.2 (~95%) is expressed in excitatory neurons including the major population of neocortical and all of hippocampal ones, and a minor amount is expressed in caudal ganglionic eminence (CGE)-derived inhibitory neurons such as vasoactive intestinal polypeptide (VIP)-positive ones (VIP-IN) (Lorincz and Nusser 2010; Yamagata et al., 2017; Ogiwara et al., 2018), however a recent study reported that a subpopulation (mostly half) of VIP-IN are Nav1.1-positive (Goff and Goldberg, 2019).

In response to the reviewer's comment, we revised the manuscript as follows;

Lines #72-76 at the Introduction section;

“In contrast, a major amount of Nav1.2 (~95%) is expressed in excitatory neurons including the most of neocortical and all of hippocampal ones, and a minor amount is expressed in caudal ganglionic eminence-derived inhibitory neurons such as vasoactive intestinal polypeptide (VIP)-positive ones (Lorincz and Nusser 2010; Yamagata et al., 2017; Ogiwara et al., 2018).”

3. Please see additional suggested edits below.A. Please edit the title to Figure 3 to state GFP-positive cells.

We have replaced the original Figure 3 with a new Figure 11 and titled as follows;

Lines #884-885 at the Figure legends section;

“Figure 11. Cells with Nav1.2-positive AISs are mostly GFP-negative in *Scn1a*-GFP mouse neocortex.”

B. The word "mimicking" is misspelt in the sentence below.We therefore performed immunohistochemical investigations of Nav1.1-mimicing GFP signals in Scn1a-GFP mice by using FEZF2 and TBR1 as projection neuron markers (Figure 4 and Figure 5).

The sentence pointed out by the reviewer has been removed from the revised manuscript.

C. Fluorescence imaging showed relatively intense GFP signals in the caudal brain portions (Figure 1D), which is well consistent with the previously reported regional distribution of Nav1.1 protein and Scn1a mRNA in wild-type mouse brain (Ogiwara et al., 2007).

We have changed the original Figure 1D to a new Figure 3 in the revision. The sentence was revised as follows;

Lines #132-136 at the Results section;

“Fluorescence imaging of the *Scn1a*-GFP sagittal brain sections at postnatal day 15 (P15), 4-week-old (4W) and 8W showed that GFP-signals continue to be intense in caudal region such as thalamus, mid brain, and brainstem (Figure 3), which is well consistent with our previous report of Nav1.1 protein and *Scn1a* mRNA distributions in wild-type mouse brain (Ogiwara et al., 2007).”

D. In the sentence below, consider the use of another segue word than instead.Instead, most AISs of TBR1-positive neurons are Nav1.2-positive (Supplemental figure S2).

The sentence pointed out by the reviewer has been removed.

E. The meaning of the sentence below is unclear and should be re-written.A majority of neocortical excitatory projection neurons in L5 consist of pyramidal tract (PT) and intratelencephalic cortico-striatal (iCS) neurons, here we define iCS rather than CS because PT also occasionally innervate striatum, and those in L6 are mostly cortico-thalamic (CT) neurons (Shepherd, 2013).

In response to the reviewer's comment, we revised the text as follows;

Lines #266-270 at the Results section;

“Neocortical excitatory neurons can be divided into functionally distinct subpopulations, a majority of those are pyramidal cells which have axons of long-range projections such as L2/3 CC, L5 PT, L5/6 CS and L6 CT projection neurons (Shepherd, 2013). Although PT neurons also project their axon collaterals to ipsilateral striatum, CS neurons project bilaterally to ipsi- and contralateral striata.”

Science Comments1. In Figure 1B the β-tubulin blot does not appear to be a re-probe of the blot for Nav1.1 as the lanes for the β-tubulin blot are not clearly visible.

In response to the reviewer’s comment, we adjusted levels of signal intensities and prepared a new Figure 1B in which the lanes for β-tubulin blot are visibly separated.

2. Using western blot or a similar technique, please quantify GFP and SCN1A protein levels in multiple (n>8-10) WT and BAC transgenic animals. This quantification will strengthen the claim that endogenous levels of Scn1a are not altered (increased/decreased) in your BAC mouse.

In response to the reviewer’s comment, we performed western blot quantification to reveal protein levels of Nav1.1 and GFP in the brains of five *Scn1a*-GFP and five wild-type mice. As a result, we were able to confirm that there was no change in the amount of Nav1.1 in *Scn1a*-GFP mouse compared to the wild type (Supplementary figure S3A). However, we detected up to a 3-fold range of change in the amount of GFP protein, the reason is so far unknown (Supplementary figure S3B).

Accordingly, we revised the manuscript as follows;

Lines #129-132 at the Results section;

“Quantification of Nav1.1 signals in western blot analyses of brain lysates from the *Scn1a*-GFP mice and their wild-type littermates (N = 5 animals per each genotype) showed no difference between genotypes, while that of GFP somehow deviated among individual *Scn1a*-GFP mice (Supplementary figure S3).”

3. This statement is unfounded from the images."Immunohistochemical double-staining of Nav1.1 and GFP revealed Nav1.1-signals at axon initial segments (AISs) of most neocortical and hippocampal GFP-positive cells (Figure 1E, F)".Please quantify the double-staining of Nav 1.1 and GFP at the AISs in a subset ( ~100 cells per region in 4-5 mice) of neocortical and hippocampal neurons.

In response to the reviewer’s comment, we performed triple immunostaining using anti-Nav1.1, anti-GFP and anti-ankyrinG antibodies and quantitatively analyzed the Nav1.1-positive AIS and soma in the motor cortex (82~124 of Nav1.1-positive AISs were counted in each layer, Figure 4 and Supplementary tables S2, S3, S4) and hippocampus (114 of Nav1.1-positive AISs were counted in CA1 area, Figure 5 and Supplementary tables S7, S8) of three *Scn1a*-GFP mice.

Accordingly, we revised the manuscript as follows;

Lines #161-180 at the Results section;

“Next, we performed triple-immunostaining of Nav1.1, GFP and ankyrinG on brains of *Scn1a*-GFP mouse at P15. In the neocortex (Figure 4), axon initial segments (AISs) of cells with Nav1.1-positive somata were always Nav1.1-positive but somata of cells with Nav1.1-positive AISs were occasionally Nav1.1-negative (Figures 4A-C). Cell counting revealed that 17% (L2/3), 21% (L5) and 8% (L6) of neurons (cells with ankyrinG-positive AISs) were GFP-positive (Figure 4D-left panel and Supplementary table S2). Of note, all cells with Nav1.1-positive AISs or somata were GFP-positive, but AISs or somata for only half of GFP-positive cells were Nav1.1-positive (Figure 4D and Supplementary tables S3, S4), possibly due to undetectably low levels of Nav1.1 immunosignals in a subpopulation of GFP-positive cells. The above ratios of GFP-positive cells among neurons (cells with ankyrinG-positive AISs) obtained in the triple-immunostaining of Nav1.1, GFP and ankyrinG are rather discordant to those obtained in the later experiment of triple-immunostaining of Nav1.2, GFP and ankyrinG, 23% (L2/3), 30% (L5) and 21% (L6) (see Figure 11). Therefore, we additionally performed double-immunostaining of GFP and ankyrinG on brains of *Scn1a*-GFP mouse at P15, and the ratios of of GFP-positive cells among neurons were 30% (L2/3), 26% (L5) and 9% (L6) (Supplementary figure S6 and Supplementary table S5). Averaged ratios of GFP-positive cells among neurons of Figures 4D, 11B and Supplementary figure S6B are 23% (L2/3), 26% (L5) and 13% (L6) (Supplementary figure S7 and Supplementary table S6), which are actually significantly lower than those obtained in the NeuN-staining (Supplementary figure S5 and Supplementary table S1).”

Lines #367-372 at the Discussion section;

“All Nav1.1-positive cells were GFP-positive. Reversely all GFP-positive cells were also Nav1.1-positive in the hippocampus, but in the neocortex only a half of GFP-positive cells were Nav1.1-positive. This is largely because in the hippocampus Nav1.1 expression is restricted to inhibitory neurons, but in the neocortex Nav1.1 is expressed not only in inhibitory but also in a subpopulation of excitatory neurons in which Nav1.1 expression is low and not easily detected immunohistochemically by anti-Nav1.1 antibodies.”

Lines #520-534 at the Materials and methods section;

"Fluorescence and Immunofluorescence quantification

For quantification of inhibitory neurons in GFP-positive cells, we used *Scn1a*-GFP/*Vgat*-Cre/*Rosa26*-tdTomamto mice at 4W. We acquired multiple color images of primary motor cortex and hippocampus from three parasagittal sections per animal. Six images per region of interest were manually counted and summarized using Adobe Photoshop Elements 10 and Excel (Microsoft). On immunofluorescence quantification, we used *Scn1a*-GFP mice at P15 and/or 4W for the quantification of immunosignals. For quantification of GFP, NeuN, PV, SST, Nav1.1, Nav1.2, FEZF2 or TBR1-positive cells, we acquired multiple color images of primary motor cortex and hippocampus from three parasagittal sections per animal. Six~nine images per region of interest were manually quantified and summarized. For quantification of PV, SST, FEZF2 or TBR1-positive cells, intensity and area size of GFP fluorescent signals were measured by Fiji software. Statistical analyses were performed by one-way ANOVA followed by Tukey–Kramer post-hoc multiple comparison test using Kyplot 6.0 (KyensLab Inc). P-value smaller than 0.05 was considered statistically significant. Data are presented as the mean ± standard error of the mean (SEM).”

4. Please include matched images i.e. at the same brain level for Figure 2A and Figure 2E.

We could not prepare the matched chromogenic images because of a paucity of available mice, but instead we added new GFP fluorescence images showing the distribution of GFP-positive cells at a matched position in *Scn1a*-GFP lines #184 and #233 (Supplementary figure S2).

Accordingly, we revised the manuscript as follows;

Lines #107-115 at the Results section;

"Both lines showed a similar distribution of chromogenic GFP immunosignals across the entire brain (Figures 2A-H), and a similar distribution was also obtained in fluorescence detection of GFP (Figures 2I-L and Supplementary figure S2). In neocortex (Figures 2B, F, J and Supplementary figures S2B, F), GFP-positive cells were distributed throughout all cortical layers. In hippocampus (Figures 2C, G, K and Supplementary figures S2C, G), cells with intense GFP signals, which are assumed to be PV-IN and SST-IN (Ogiwara et al., 2007; Tai et al., 2014) (see also Figure 8), were scattered in stratum oriens, pyramidale, radiatum, lucidum and lacunosum-moleculare of the CA (cornu ammonis) fields, hilus and molecular layer of dentate gyrus.”

Lines #125-127 at the Results section;

"In cerebellum (Figures 2D, H, L and Supplementary figures S2D, H), GFP signals appeared in Purkinje, basket, and deep cerebellar nuclei cells, again consistent to the previous reports (Ogiwara et al., 2007; Ogiwara et al., 2013).”

5. The GFP-positive cells do not appear to be sparsely distributed throughout all cortical layers. Please include counts for the number of (GFP+ NeuN+/ NeuN) cells across the cortical layers for either the 184 or 233 S.

In response to the editor’s comment, we performed immunostaining of NeuN using *Scn1a*-GFP mouse line #233 and counted cell numbers (Supplementary figure S5 and Supplementary table S1). In the neocortex, 20~30% of NeuN-positive cells were GFP-positive. However, we noticed that sparsely-distributed cells with intense GFP-signals, which are PV-INs, were often NeuN-negative. NeuN-positive cells therefore do not represent all neurons in neocortex and above figure should deviate from the real ratios of GFP-positive cells among all neurons. Instead, we estimated the ratio of GFP-positive cells among all neurons in the experiments of GFP and ankyrinG stainings (see Figures 4, 11, S6, S7).

Accordingly, we revised the manuscript as follows;

Lines #149-180 at the Results section;

“In order to know the ratio of GFP-positive cells among all neurons, we further performed immunohistochemical staining using NeuN-antibody on *Scn1a*-GFP mouse at P15 and cells were counted at M1 and S1 (Supplementary figure S5 and Supplementary table S1). The NeuN staining showed that GFP-positive cells occupy 30% (L2/3), 32% (L5) and 22% (L6) of NeuN- or GFP-positive cells at P15 (Supplementary figure S5B and Supplementary table S1). However, we noticed that sparsely-distributed cells with intense GFP-signals, which are assumed to be PV-INs (see Figure 8), were often NeuN-negative (Supplementary figure S5A – arrowheads), reminiscent of a previous report that NeuN expression is absent in cerebellar inhibitory neurons such as Golgi, basket and satellite cells in cerebellum (Weyer et al., 2003). Therefore, NeuN-positive cells do not represent all neurons in neocortex as well. NeuN/GFP-double negative neurons could even exist and therefore above figure (Supplementary figure S5B) may deviate from the real ratios of GFP-positive cells among all neurons.

Next, we performed triple-immunostaining of Nav1.1, GFP and ankyrinG on brains of *Scn1a*-GFP mouse at P15. In the neocortex (Figure 4), axon initial segments (AISs) of cells with Nav1.1-positive somata were always Nav1.1-positive but somata of cells with Nav1.1-positive AISs were occasionally Nav1.1-negative (Figures 4A-C). Cell counting revealed that 17% (L2/3), 21% (L5) and 8% (L6) of neurons (cells with ankyrinG-positive AISs) were GFP-positive (Figure 4D-left panel and Supplementary table S2). Of note, all cells with Nav1.1-positive AISs or somata were GFP-positive, but AISs or somata for only half of GFP-positive cells were Nav1.1-positive (Figure 4D and Supplementary tables S3, S4), possibly due to undetectably low levels of Nav1.1 immunosignals in a subpopulation of GFP-positive cells. The above ratios of GFP-positive cells among neurons (cells with ankyrinG-positive AISs) obtained in the triple-immunostaining of Nav1.1, GFP and ankyrinG are rather discordant to those obtained in the later experiment of triple-immunostaining of Nav1.2, GFP and ankyrinG, 23% (L2/3), 30% (L5) and 21% (L6) (see Figure 11). Therefore, we additionally performed double-immunostaining of GFP and ankyrinG on brains of *Scn1a*-GFP mouse at P15, and the ratios of of GFP-positive cells among neurons were 30% (L2/3), 26% (L5) and 9% (L6) (Supplementary figure S6 and Supplementary table S5). Averaged ratios of GFP-positive cells among neurons of Figures 4D, 11B and Supplementary figure S6B are 23% (L2/3), 26% (L5) and 13% (L6) (Supplementary figure S7 and Supplementary table S6), which are actually significantly lower than those obtained in the NeuN-staining (Supplementary figure S5 and Supplementary table S1).”

Line #502-504 at the Materials and methods section;

"Following antibodies were used to detect GFP, Nav1.1, Nav1.2, TBR1, FEZF2, ankyrinG, NeuN, parvalbumin and somatostatin; …”

Line #510-511 at the Materials and methods section;

"… mouse anti-NeuN biotin conjugated antibody (1:2,000; MAB377B, Millipore), …”

Line #515-516 at the Materials and methods section;

"To detect NeuN, Alexa-647 conjugated streptavidin (1:1,000; S21374, Thermo Fisher Science) was used."

Lines #520-534 at the Materials and methods section;

"Fluorescence and Immunofluorescence quantification

For quantification of inhibitory neurons in GFP-positive cells, we used *Scn1a*-GFP/*Vgat*-Cre/*Rosa26*-tdTomamto mice at 4W. We acquired multiple color images of primary motor cortex and hippocampus from three parasagittal sections per animal. Six images per region of interest were manually counted and summarized using Adobe Photoshop Elements 10 and Excel (Microsoft). On immunofluorescence quantification, we used *Scn1a*-GFP mice at P15 and/or 4W for the quantification of immunosignals. For quantification of GFP, NeuN, PV, SST, Nav1.1, Nav1.2, FEZF2 or TBR1-positive cells, we acquired multiple color images of primary motor cortex and hippocampus from three parasagittal sections per animal. Six~nine images per region of interest were manually quantified and summarized. For quantification of PV, SST, FEZF2 or TBR1-positive cells, intensity and area size of GFP fluorescent signals were measured by Fiji software. Statistical analyses were performed by one-way ANOVA followed by Tukey–Kramer post-hoc multiple comparison test using Kyplot 6.0 (KyensLab Inc). P-value smaller than 0.05 was considered statistically significant. Data are presented as the mean ± standard error of the mean (SEM)."

6. Only chromogenic staining seemed to be shown for both the 184 and 233 Scn1a-GFP lines. Thus, please re-write the sentence below:The Scn1a-GFP lines #184 and 233 showed a similar distribution of GFP-signals, immuno-reactive and fluorescent signals as well, across the entire brain (Figure 1C, 2A-L).as:The Scn1a-GFP lines #184 and 233 showed a similar distribution of chromogenic GFP-signals across the entire brain (Figure 1C, 2A-D, 2E-2H).

We have added new GFP fluorescence images showing the distribution of GFP-positive cells at a matched position in *Scn1a*-GFP lines #184 and #233 (Supplementary figure S2).

Accordingly, we have revised the manuscript as follows;

Lines #107-128 at the Results section;

“Both lines showed a similar distribution of chromogenic GFP immunosignals across the entire brain (Figures 2A-H), and a similar distribution was also obtained in fluorescence detection of GFP (Figures 2I-L and Supplementary figure S2). In neocortex (Figures 2B, F, J and Supplementary figures S2B, F), GFP-positive cells were distributed throughout all cortical layers. In hippocampus (Figures 2C, G, K and Supplementary figures S2C, G), cells with intense GFP signals, which are assumed to be PV-IN and SST-IN (Ogiwara et al., 2007; Tai et al., 2014) (see also Figure 8), were scattered in stratum oriens, pyramidale, radiatum, lucidum and lacunosum-moleculare of the CA (cornu ammonis) fields, hilus and molecular layer of dentate gyrus. Of note, somata of dentate granule cells were apparently GFP-negative. CA1~3 pyramidal cells were twined around with fibrous GFP immunosignals. We previously reported that the fibrous Nav1.1-signals clinging to somata of hippocampal CA1~3 pyramidal cells were disappeared by conditional elimination of Nav1.1 in PV-INs but not in excitatory neurons, and therefore concluded that these Nav1.1-immunopositive fibers are axon terminals of PV-INs (Ogiwara et al., 2013). As such, GFP signals are fibrous but do not form cell shapes in the CA pyramidal cell layer (Figures 2C, G, K and Supplementary figures S2C, G), and therefore these CA pyramidal cells themselves are assumed to be GFP-negative. These observations further confirmed our previous proposal that hippocampal excitatory neurons are negative for Nav1.1 (Ogiwara et al., 2007; Ogiwara et al., 2013). In cerebellum (Figures 2D, H, L and Supplementary figures S2D, H), GFP signals appeared in Purkinje, basket, and deep cerebellar nuclei cells, again consistent to the previous reports (Ogiwara et al., 2007; Ogiwara et al., 2013). In the following analyses, we used the line #233 which shows stronger GFP signals than #184.”

7. Please replace this statement: These data indicate that GFP signals in the Scn1a-GFP lines faithfully reflect endogenous Scn1a/Nav1.1 expression.withThese data indicate that GFP signals in the Scn1a-GFP lines may phenocopy endogenous Scn1a/Nav1.1 expression in the neocortex, hippocampus, and cerebellum.

After an intensive revision, the sentence pointed out by the reviewer has been removed.

8. The data in figure 3 is not convincing as it is difficult to distinguish the AIS of GFP- positive and non-GFP positive cells. Thus, it is unclear whether the AISs of GFP-positive cells are Nav1.2-negative.

In response to the editor’s comment, we additionally performed triple-immunostaining for Nav1.1, Nav1.2 and ankyrinG (Figure 10 and Supplementary table S17). The analysis revealed that Nav1.1/Nav1.2 double-positive AISs were less than 0.5% of ankyrinG-positive AIS, and most of AIS were Nav1.2-positive in neocortex.

Accordingly, we revised the manuscript as follows;

Lines #237-262 at the Results section;

“Nav1.1 and Nav1.2 expressions are mutually exclusive in mouse brain

We previously reported that expressions of Nav1.1 and Nav1.2 seem to be mutually-exclusive in multiple brain regions including neocortex, hippocampal CA1, dentate gyrus, striatum, globus pallidus, and cerebellum in wild-type mice (Yamagata et al., 2017). To further confirm it, here we performed triple immunostaining for Nav1.1, Nav1.2 and ankyrinG, and counted Nav1.1- or Nav1.2-immunopositive AISs in the neocortex of *Scn1a*-GFP mice (Figure 10). The staining again showed that Nav1.1 and Nav1.2 expressions are mutually exclusive in brain regions including neocortex (Figure 10A). The counting revealed that 5% (L2/3), 6% (L5) and 3% (L6) of AISs at P15 were Nav1.1-positive and 78% (L2/3), 69% (L5) and 69% (L6) of AISs at P15 were Nav1.2-positive in the neocortex (Figure 10B and Supplementary table S17). Of note, less than 0.5% of AISs are Nav1.1/Nav1.2-double positive confirming that Nav1.1 and Nav1.2 do not co-express. These results are consistent with our previous study (Yamagata et al., 2017) and further support that expressions of Nav1.1 and Nav1.2 are mutually exclusive in mouse neocortex at least at immunohistochemical level.

As mentioned, GFP signals in *Scn1a*-GFP mouse can represent even moderate or low Nav1.1 expressions which cannot be detected by immunohistochemical staining, so some of GFP-positive cells may still express Nav1.2. To investigate whether and if so how much of GFP-positive cells have Nav1.2-positive AISs in *Scn1a*-GFP mouse neocortex, we performed triple immunohistochemical staining for Nav1.2, GFP and ankyrinG (Figure 11 and Supplementary tables S18-S20). The staining showed that AISs of GFP-positive cells are largely negative for Nav1.2, and cells with Nav1.2-positive AISs are mostly GFP-negative (Figure 11A). Cell counting revealed that 88% (L2/3), 90% (L5) and 95% (L6) of cells with Nav1.2-positive AISs at P15 were GFP-negative (Figure 11B-middle panel and Supplementary table S19), and 69% (L2/3), 83% (L5), and 86% (L6) of AISs of GFP-positive cells at P15 were Nav1.2-negative (Figure 11B-right panel and Supplementary table S20). These results indicate that the co-expression of GFP and Nav1.2 would be minimal if any.”

Lines #520-534 at the Materials and methods section;

"Fluorescence and Immunofluorescence quantification

For quantification of inhibitory neurons in GFP-positive cells, we used *Scn1a*-GFP/*Vgat*-Cre/*Rosa26*-tdTomamto mice at 4W. We acquired multiple color images of primary motor cortex and hippocampus from three parasagittal sections per animal. Six images per region of interest were manually counted and summarized using Adobe Photoshop Elements 10 and Excel (Microsoft). On immunofluorescence quantification, we used *Scn1a*-GFP mice at P15 and/or 4W for the quantification of immunosignals. For quantification of GFP, NeuN, PV, SST, Nav1.1, Nav1.2, FEZF2 or TBR1-positive cells, we acquired multiple color images of primary motor cortex and hippocampus from three parasagittal sections per animal. Six ~ nine images per region of interest were manually quantified and summarized. For quantification of PV, SST, FEZF2 or TBR1-positive cells, intensity and area size of GFP fluorescent signals were measured by Fiji software. Statistical analyses were performed by one-way ANOVA followed by Tukey–Kramer post-hoc multiple comparison test using Kyplot 6.0 (KyensLab Inc). P-value smaller than 0.05 was considered statistically significant. Data

9. In figures S1A, S1E, and S1AF, some cells with "dense" and "less intense" staining appear to be stained for GFP and an inhibitory marker. Likewise, some cells have prominent GFP staining but are not labeled for either SST or PV. To strengthen your claim, the data presented in supplementary figure S1 should include quantification of brain slices that are triple labeled for PV, SST and GFP. Cell counts should then be included for the fraction of GFP cells that are PV+ and SST+.

In response to the reviewer’s comment, we performed triple immunostaining for GFP, PV and SST, and its quantification using *Scn1a*-GFP mouse line #233. In this revised version, the original Supplementary figure S1 was changed to Figure 8 and then we performed quantification of immunostaining for GFP and PV or SST (Figure 8B and Supplementary tables S11-S13). In addition, in order to complement the results of Figure 8, we performed analysis for the intensity of GFP immunosignals in PV+, GFP+ and SST+ GFP+ cells (Figure 9 and Supplementary tables S15, S16). These results indicated that higher expression of GFP in PV-positive neurons.

Accordingly, we revised the manuscript as follows;

Lines #215-235 at the Results section;

“We further performed immunohistochemical staining of PV and SST in neocortex and hippocampus of *Scn1a*-GFP mice at 4W (Figure 8). PV and SST do not co-express in cells and do not overlap. PV-INs and SST-INs were both GFP-positive, and especially GFP signals in PV-INs were intense (Figure 8A). Cell counting revealed that 21% (L2/3), 37% (L5), 37% (L6), 58% (CA1), 42% (CA2/3), and 41% (DG) of GFP-positive cells were PV or SST-positive depending on regions in neocortex and hippocampus (Figure 8B and Supplementary tables S11-S13). All PV-INs were GFP-positive (Figure 8B – middle), and most of SST-INs were GFP-positive (Figure 8B – right). Comparison of these results with those of *Vgat*-Cre mouse (Figure 7) suggests that GFP-positive GABAergic neurons in neocortex are mostly PV- or SST-positive, while in hippocampus a half of those are PV/SST-negative GABAergic neurons. Higher ratios of PV- or SST-positive cells (Figure 8B) compared with those of *Vgat*-Cre-positive cells (Figure 7C) among GFP-positive cells would be explained by that we counted PV-positive cells even if their PV-immunosignals are moderate and a significant subpopulation of such cells are known to be excitatory neurons (Jinno et al., 2004; Tanahira et al., 2009; Matho et al., 2021). Quantitative analysis of GFP signal intensity and area size of cells revealed that GFP signal intensities in PV-positive cells were significantly higher than those in PV-negative cells and GFP signal intensities in SST-positive cells were lower than those in PV-positive cells but similar to PV/SST-double negative cells (Figure 9 and Supplementary tables S14-S16). These results indicate that Nav1.1 expression level in PV-INs is significantly higher than those in excitatory neurons and PV-negative GABAergic neurons including SST-INs.”

Lines #520-534 at the Materials and methods section;

"Fluorescence and Immunofluorescence quantification

For quantification of inhibitory neurons in GFP-positive cells, we used *Scn1a*-GFP/*Vgat*-Cre/*Rosa26*-tdTomamto mice at 4W. We acquired multiple color images of primary motor cortex and hippocampus from three parasagittal sections per animal. Six images per region of interest were manually counted and summarized using Adobe Photoshop Elements 10 and Excel (Microsoft). On immunofluorescence quantification, we used *Scn1a*-GFP mice at P15 and/or 4W for the quantification of immunosignals. For quantification of GFP, NeuN, PV, SST, Nav1.1, Nav1.2, FEZF2 or TBR1-positive cells, we acquired multiple color images of primary motor cortex and hippocampus from three parasagittal sections per animal. Six~nine images per region of interest were manually quantified and summarized. For quantification of PV, SST, FEZF2 or TBR1-positive cells, intensity and area size of GFP fluorescent signals were measured by Fiji software. Statistical analyses were performed by one-way ANOVA followed by Tukey–Kramer post-hoc multiple comparison test using Kyplot 6.0 (KyensLab Inc). P-value smaller than 0.05 was considered statistically significant. Data are presented as the mean ± standard error of the mean (SEM)."

10. Objective and quantitative definitions for "dense" and "less" intense" GFP staining should be established. Next, the fraction of inhibitory, and excitatory cells that have "dense" and "less" intense" GFP staining should been quantified. Without such criteria, it is difficult to conclude that FEZF2 or TBR1 signals were found in cells with less intense GFP signals.

In response to the reviewer’s comment, we performed quantifications of intensity and area size of GFP immunofluorescence for GFP only and GFP and PV (or SST)-double positive (Figure 9) or FEZF2 (or TBR1)-double positive cells (Figures 13 and 15).

Accordingly, we revised the manuscript as follows;

Lines #229-233 at the Results section;

“Quantitative analysis of GFP signal intensity and area size of cells revealed that GFP signal intensities in PV-positive cells were significantly higher than those in PV-negative cells and GFP signal intensities in SST-positive cells were lower than those in PV-positive cells but similar to PV/SST-double negative cells (Figure 9 and Supplementary tables S14-S16).”

Lines #282-291 at the Results section;

“Quantitative analyses revealed that FEZF2/GFP-double positive cells in L5 showed significantly lower GFP immunosignal intensities and larger signal areas (soma sizes) compared to those of FEZF2-negative/GFP-positive cells (Figure 13 and Supplementary table S24), indicating that L5 PT neurons showed lower Nav1.1 expression compared to other neurons such as PV-INs (see also Figure 9). Soma sizes of GFP-positive cells in L6 were overall smaller than those of FEZF2/GFP-double positive cells in L5, and there was no statistically significant difference in size between the FEZF2-positive/negative subpopulations. However, FEZF2/GFP-double positive cells still showed lower intensity of GFP signals compared to FEZF2-negative/GFP-positive cells (Figure 13 and Supplementary table S24).”

Lines #321-336 at the Results section;

“To further elucidate the distributions of GFP (Nav1.1)-expressing neurons in neocortex, we performed immunohistochemical staining of TBR1 on *Scn1a*-GFP mice (Figure 14 and Supplementary tables S33-S35) and quantitated GFP signal intensities and area sizes of cells (Figure 15 and Supplementary table S36). TBR1-positive cells were predominant in neocortical L6, some in L5 and a few in L2/3. In L5, contrary to the high ratio of GFP-positive cells among FEZF2-positive cells (83% at P15, and 96% at 4W) (Figure 12B), the ratios of GFP-positive cells among TBR1-positive cells are quite low (11% at P15, and 5% at 4W) (Figure 14B-middle panels and Supplementary table S34). Soma sizes of TBR1/GFP-double positive cells were smaller than those of TBR1-negative/GFP-positive cells (Figure 15-middle panel and Supplementary table S36). In L6 where a major population of TBR1-positive neurons locate, the ratios of GFP-positive cells among TBR1-positive cells are still low (15% at P15, and 26% at 4W) (Figure 14B-middle panels and Supplementary table S34). Soma sizes of TBR1/GFP-double positive cells were also smaller than those of TBR1-negative/GFP-positive cells, and TBR1/GFP-double positive cells showed lower intensity of GFP immunosignals compared to TBR1-negative/GFP-positive cells (Figure 15-right panel and Supplementary table S36).”

Lines #520-534 at the Materials and methods section;

"Fluorescence and Immunofluorescence quantification

For quantification of inhibitory neurons in GFP-positive cells, we used *Scn1a*-GFP/*Vgat*-Cre/*Rosa26*-tdTomamto mice at 4W. We acquired multiple color images of primary motor cortex and hippocampus from three parasagittal sections per animal. Six images per region of interest were manually counted and summarized using Adobe Photoshop Elements 10 and Excel (Microsoft). On immunofluorescence quantification, we used *Scn1a*-GFP mice at P15 and/or 4W for the quantification of immunosignals. For quantification of GFP, NeuN, PV, SST, Nav1.1, Nav1.2, FEZF2 or TBR1-positive cells, we acquired multiple color images of primary motor cortex and hippocampus from three parasagittal sections per animal. Six~nine images per region of interest were manually quantified and summarized. For quantification of PV, SST, FEZF2 or TBR1-positive cells, intensity and area size of GFP fluorescent signals were measured by Fiji software. Statistical analyses were performed by one-way ANOVA followed by Tukey–Kramer post-hoc multiple comparison test using Kyplot 6.0 (KyensLab Inc). P-value smaller than 0.05 was considered statistically significant. Data are presented as the mean ± standard error of the mean (SEM)."

11. Please provide quantification to support the statement made in the manuscript that most AISs of TBR1-positive neurons are Nav1.2-positive.

In response to the reviewer’s comment, we added the results of quantification of Nav1.2 positive AIS of TBR1 positive cells (Supplementary figure S12 and Supplementary tables S40-S42) which support that most AISs of TBR1-positive neurons are Nav1.2-positive.

Accordingly, we revised the manuscript as follows;

Lines #342-350 at the Results section;

“To investigate whether TBR1-positive cells express Nav1.2, we performed triple immunostaining of TBR1, Nav1.2 and ankyrinG on *Scn1a*-GFP mice (Supplementary figure S12 and Supplementary tables S40-S43). In L5, contrary to the low ratios of GFP-positive cells among TBR1-positive cells (11% at P15 and 5% at 4W) (Figure 14B-middle panels and Supplementary table S34), the ratios of Nav1.2-positive cells among TBR1-positeve cells are high (69% (L2/3), 69%(L5) and 69% (L6) at P15) (Supplementary figure S12B-right-upper panel and Supplementary table S42). The ratios of TBR1-positive cells among Nav1.2-positive cells are 29% (L2/3), 53% (L5) and 62% (L6) at P15 (Supplementary figure S12B-middle upper panel and Supplementary table S41).”

Lines #520-534 at the Materials and methods section;

"Fluorescence and Immunofluorescence quantification

For quantification of inhibitory neurons in GFP-positive cells, we used *Scn1a*-GFP/*Vgat*-Cre/*Rosa26*-tdTomamto mice at 4W. We acquired multiple color images of primary motor cortex and hippocampus from three parasagittal sections per animal. Six images per region of interest were manually counted and summarized using Adobe Photoshop Elements 10 and Excel (Microsoft). On immunofluorescence quantification, we used *Scn1a*-GFP mice at P15 and/or 4W for the quantification of immunosignals. For quantification of GFP, NeuN, PV, SST, Nav1.1, Nav1.2, FEZF2 or TBR1-positive cells, we acquired multiple color images of primary motor cortex and hippocampus from three parasagittal sections per animal. Six~nine images per region of interest were manually quantified and summarized. For quantification of PV, SST, FEZF2 or TBR1-positive cells, intensity and area size of GFP fluorescent signals were measured by Fiji software. Statistical analyses were performed by one-way ANOVA followed by Tukey–Kramer post-hoc multiple comparison test using Kyplot 6.0 (KyensLab Inc). P-value smaller than 0.05 was considered statistically significant. Data are presented as the mean ± standard error of the mean (SEM)."

Reviewer #2 (Recommendations for the authors):There is an excessive use of abbreviations which makes the reading of both the results and the discussion quite difficult. This can be minimized as much as possible.

In response to the reviewer’s comment, we corrected the positions of abbreviations or removed some of those and fully spelled out the words.

For example;

Lines #34-35 at the Abstract section;

“Although a distinct subpopulation of layer V (L5) neocortical excitatory neurons were also reported …”

Lines #41-45 at the Abstract section;

“By using neocortical excitatory projection neuron markers including FEZF2 for L5 pyramidal tract (PT) and TBR1 for layer VI (L6) cortico-thalamic (CT) projection neurons, we further show that most L5 PT neurons and a minor subpopulation of layer II/III (L2/3) cortico-cortical (CC) neurons express Nav1.1 while the majority of L6 CT, L5/6 cortico-striatal (CS) and L2/3 CC neurons express Nav1.2.”

Line #59-60 at the Introduction section;

“… such as epilepsy, autism spectrum disorder (ASD) and intellectual disability, …”

Line #66 at the Introduction section;

“…. expressed in medial ganglionic eminence-derived parvalbumin-positive …”

Line #74 at the Introduction section;

“… a minor amount is expressed in caudal ganglionic eminence-derived …”

Line #75 at the Introduction section;

“… inhibitory neurons such as vasoactive intestinal polypeptide (VIP)-positive ones …”

Lines #385 at the Discussion section;

“… features, ataxia and increased risk of sudden unexpected death in epilepsy (SUDEP).“

The arrow heads in Figure 3 do not clearly point towards what authors indicate they are pointing towards.

We have replaced the original Figure 3 to a new one Figure 11 in which the indicators were carefully arranged.

Figure 3 shows that Nav1.2 is not expressed in the axon initial segments of GRP-positive cells in the cortex and other brain regions. Indeed the authors state that data for the "other brain regions" are not shown but it will be great if they could mention of some of these brain regions.

In the present study, we generated *Scn1a*-GFP mouse and investigated the distribution of GFP signals mainly in neocortex and hippocampus. Although we surely reported that expressions of Nav1.1 and Nav1.2 are mutually-exclusive in striatum, globus pallidus, and cerebellum in addition to neocortex and hippocampus (Yamagata et al., 2017), we are afraid that the inclusion of data for mutually exclusiveness of GFP and Nav1.1 distributions in other brain regions such as striatum, globus pallidus, and cerebellum are too much for the current manuscript.

The authors state that "most AIS of TBRI-positive neurons are Nav1.2 positive" and go on to show only one supplemental figure (Supplemental Figure S2). The conclusions drawn from Figure 5 and Figure 6 were supported by percentage expression patterns in Supplemental Table 1. The authors should include a similar supplementary table to support the data shown in Supplemental Figure S2, since a conclusive statement is being made from that figure.

We newly performed a triple-staining of TBR1, Nav1.2 and ankyrinG and counted cell numbers (Supplementary figure S12 and Supplementary tables S40-S42). These results would comply with the reviewer's request.

Accordingly, we revised the manuscript as follows;

Lines #342-350 at the Results section;

“To investigate whether TBR1-positive cells express Nav1.2, we performed triple immunostaining of TBR1, Nav1.2 and ankyrinG on *Scn1a*-GFP mice (Supplementary figure S12 and Supplementary tables S40-S43). In L5, contrary to the low ratios of GFP-positive cells among TBR1-positive cells (11% at P15 and 5% at 4W) (Figure 14B-middle panels and Supplementary table S34), the ratios of Nav1.2-positive cells among TBR1-positeve cells are high (69% (L2/3), 69%(L5) and 69% (L6) at P15) (Supplementary figure S12B-right-upper panel and Supplementary table S42). The ratios of TBR1-positive cells among Nav1.2-positive cells are 29% (L2/3), 53% (L5) and 62% (L6) at P15 (Supplementary figure S12B-middle upper panel and Supplementary table S41).”

Lines #520-534 at the Materials and methods section;

"Fluorescence and Immunofluorescence quantification

For quantification of inhibitory neurons in GFP-positive cells, we used *Scn1a*-GFP/*Vgat*-Cre/*Rosa26*-tdTomamto mice at 4W. We acquired multiple color images of primary motor cortex and hippocampus from three parasagittal sections per animal. Six images per region of interest were manually counted and summarized using Adobe Photoshop Elements 10 and Excel (Microsoft). On immunofluorescence quantification, we used *Scn1a*-GFP mice at P15 and/or 4W for the quantification of immunosignals. For quantification of GFP, NeuN, PV, SST, Nav1.1, Nav1.2, FEZF2 or TBR1-positive cells, we acquired multiple color images of primary motor cortex and hippocampus from three parasagittal sections per animal. Six~nine images per region of interest were manually quantified and summarized. For quantification of PV, SST, FEZF2 or TBR1-positive cells, intensity and area size of GFP fluorescent signals were measured by Fiji software. Statistical analyses were performed by one-way ANOVA followed by Tukey–Kramer post-hoc multiple comparison test using Kyplot 6.0 (KyensLab Inc). P-value smaller than 0.05 was considered statistically significant. Data are presented as the mean ± standard error of the mean (SEM)."

There are a few grammatical errors that need addressing:Page 5 Paragraph 1: "Human has nine alphas (Nav1.1~Nav1.9) and four betas (β-1~β-4)."

We revised the manuscript as follows;

Lines #54-55 at the Introduction section;

“Human has nine α (Nav1.1~Nav1.9) and four β (β-1~β-4) subunits.”

Results Paragraph 2: "….and GFP fluorescent signals are well negative in CA pyramidal and Dentate granule cells….."

We revised the manuscript as follows;

Lines #120-123 at the Results section;

“As such, GFP signals are fibrous but do not form cell shapes in the CA pyramidal cell layer (Figures 2C, G, K and Supplementary figures S2C, G), and therefore these CA pyramidal cells themselves are assumed to be GFP-negative.”

Results Paragraph 4: "We therefore examined whether Nav1.2 is expressed in of cells expressing GFP and found that….."

The sentence pointed out by the reviewer has been removed from the revised text.

Results, Final Paragraph: The first sentence needs a rewrite. The use of "mostly half" is inappropriate and does not lend to understanding. The last sentence also needs a rewrite "…..the remained Nav1.1-negative are assumed to be…"

The sentence pointed out by the reviewer has been removed from the revised text.

Discussion: The last two paragraphs of the discussion could benefit from a rewrite which seeks to simplify the ideas being conveyed. These are really long sentences which reduce clarity of the statements being made. There are also multiple grammatical errors.

In response to the reviewer's comment and the comment #7 in Essential Revision, we revised the manuscript as follows;

Lines #406-409 at the Discussion section;

“Kalume and colleagues (2013) reported that parasympathetic hyperactivity is observed in Nav1.1 haplo-deficient mice and it causes ictal bradycardia and finally result in seizure-associated sudden death.”

Lines #410-415 at the Discussion section;

“Our present finding of Nav1.1 expression in L5 pyramidal tract projection neurons which innervate the vagus nerve may possibly elucidate the ameliorating effects of Nav1.1 haploinsufficiency in neocortical excitatory neurons for sudden death of Nav1.1 haplo-deficient mice and may contribute to the understanding of the neural circuit for SUDEP in patients with Dravet syndrome. Further studies including retrograde tracing and electrophysiological analyses are awaited.”

Reviewer #3 (Recommendations for the authors):This new transgenic mice is very exciting and an important contribution to the field of epilepsy mechanisms.

Thank you so much for your intensive reviews and generous comments.

My main suggestion for the manuscript is to remove the portion about how SUDEP might happen in Dravet Syndrome, including Figure 7. This section is purely speculative and is not necessary for the main objectives of the paper: Presenting the transgenic mouse model and further characterizing the excitatory cortical neurons in which Nav1.1 is expressed. I think the hypothesis proposed is intriguing, and as mentioned in the public review the transgenic mice will be useful in future experiments testing it. But right now I feel it weakens the paper by ending on a note that is not directly supported by the data presented in the manuscript (we don't see confirmation that the problem is cortical disinhibition and not dysfunction of hypothalamic neurons that also express Nav1.1 and would represent the bulk of input to the autonomic brainstem centers with the corticobulbar projections representing a minority and might even just go to the portion of the nucleus ambiguus in charge of pharyngeal muscles).

We agree with the reviewer's comment and therefore eliminated the discussion for Figure 7 and SUDEP from the revised manuscript as we responded to the Essential Revisions: Editor's comment #7.

While I agree with the authors that statistical methods are not necessary for a characterization study, the methods section would benefit from additional details. How many sections were reviewed per animal per age?

In response to the reviewer’s comment, we added the details in Method section of the revised manuscript.

Lines #483-486 at the Materials and methods section;

"The intensity of the Nav1.1 immunosignals were quantified using the Image Studio Lite software (LI-COR, Lincoln, Nebraska USA) and normalized to the level of β tubulin or Glyceraldehyde 3-phosphate dehydrogenase."

Lines #492-493 at the Materials and methods section;

"… brains were cryoprotected with 30% sucrose in PBS, cut in 30 µm parasagittal sections, and mounted on glass slides. "

Lines #495 at the Materials and methods section;

"For immunostaining, frozen parasagittal sections (30 µm) were blocked with …"

Lines #499-516 at the Materials and methods section;

"The antibody-antigen complexes were visualized using the Vectastain Elite ABC kit (Vector Laboratories) with Metal enhanced DAB substrate (34065, PIERCE). For immunofluorescent staining, we prepared 6 µm parasagittal sections from paraffin embedded PLP-fixed brains of mice. The sections were processed as previously described (Yamagata et al., 2017). Following antibodies were used to detect GFP, Nav1.1, Nav1.2, TBR1, FEZF2, ankyrinG, NeuN, parvalbumin and somatostatin; mouse anti-GFP antibodies (1:500; 11814460001, Roche Diagnostics), anti-Nav1.1 antibodies (1:10,000; rabbit IO1, 1:500; goat SC-16031, Santa Cruz Biotechnology), anti-Nav1.2 antibodies (1:1,000; rabbit ASC-002, Alomone Labs; goat SC-31371, Santa Cruz Biotechnology), rabbit anti-TBR1 antibody (1:1,000; ab31940, Abcam or 1:500; SC-376258, Santa Cruz Biotechnology), rabbit anti-FEZF2 antibody (1:500; #18997, IBL), ankyrinG antibodies (1:500; mouse SC-12719, rabbit SC-28561; goat, SC-31778, Santa Cruz Biotechnology), mouse anti-NeuN biotin conjugated antibody (1:2,000; MAB377B, Millipore), rabbit anti-parvalbumin (1:5,000; PC255L, Merck) and rabbit anti-somatostatin (1:5,000; T-4103, Peninsula Laboratories, 1:1,000; SC-7819, Santa Cruz Biotechnology) antibodies. As secondary antibodies, Alexa Fluor Plus 488, 555, 594 and 647 conjugated antibodies (1:1,000; A32723, A32766, A32794, A32754, A32849, A32795, A32787, Thermo Fisher Scientific) were used. To detect NeuN, Alexa-647 conjugated streptavidin (1:1,000; S21374, Thermo Fisher Science) was used.”

How many images per section?

In response to the reviewer’s comment, we added requested information in Method section of the revised manuscript.

There is a mention in the Results section of the FEZF2 neurons being quantified in the primary motor cortex. Were all images taken in this region? Were the sections sagital or coronal?

We used parasagittal sections for all immunostainings in this study. For the neocortex, we used regions of primary motor cortex in all experiments except for NeuN staining in which two regions (primary motor and somatosensory cortices) were investigated (Supplementary table S5).

Accordingly, we revised the manuscript as follows;

Lines #520-534 at the Materials and methods section;

"Fluorescence and Immunofluorescence quantification

For quantification of inhibitory neurons in GFP-positive cells, we used *Scn1a*-GFP/*Vgat*-Cre/*Rosa26*-tdTomamto mice at 4W. We acquired multiple color images of primary motor cortex and hippocampus from three parasagittal sections per animal. Six images per region of interest were manually counted and summarized using Adobe Photoshop Elements 10 and Excel (Microsoft). On immunofluorescence quantification, we used *Scn1a*-GFP mice at P15 and/or 4W for the quantification of immunosignals. For quantification of GFP, NeuN, PV, SST, Nav1.1, Nav1.2, FEZF2 or TBR1-positive cells, we acquired multiple color images of primary motor cortex and hippocampus from three parasagittal sections per animal. Six~nine images per region of interest were manually quantified and summarized. For quantification of PV, SST, FEZF2 or TBR1-positive cells, intensity and area size of GFP fluorescent signals were measured by Fiji software. Statistical analyses were performed by one-way ANOVA followed by Tukey–Kramer post-hoc multiple comparison test using Kyplot 6.0 (KyensLab Inc). P-value smaller than 0.05 was considered statistically significant. Data are presented as the mean ± standard error of the mean (SEM).”

The subheading "Nav1.1 is expressed in pyramidal tract while Nav1.2 in cortico-striatal and cortico-thalamic projection neurons" of the Results section would benefit from a quick review of the main projections from each layer at the top. Referring to figure 6 earlier would also make it easier for the reader to follow the results presented later in the section. Figure 6 is a great way to summarize the main points of the latter experiments by the way.

According to the Essential Revisions: Editor's comment #6, we eliminated the original Figure 6.

The authors assume that the FEZF2 and TBR1 positive neurons that are GFP negative are Nav1.2 positive. I assume this was not confirmed experimentally because the antibody for both transcription factors and Nav1.2 are all rabbit derived, but if other antibodies are available this would be an important point to confirm through IHC.

In the revised manuscript, we additionally performed triple stainings for FEZF2, Nav1.2 and ankyrinG (Supplementary figure S9), FEZF2, GFP and Nav1.2 (Supplementary figure S10), TBR1, Nav1.2 and ankyrinG (Supplementary figure S12) and TBR1, GFP and Nav1.2 (Supplementary figure S13) and counted cell numbers. These results should comply with the reviewer's request.

Accordingly, we revised the manuscript as follows;

Lines #298-313 at the Results section;

"We also performed triple immunostaining of FEZF2, Nav1.2 and ankyrinG on *Scn1a*-GFP mice at P15 (Supplementary figure S9 and Supplementary tables S28-S31). The staining showed that 20% of neurons (cells with ankyrinG-positive AISs) in L5 are FEZF2-positive and a half of L5 FEZF2-positive cells have Nav1.2-positive AISs. Together with the observation that most of FEZF2-positive cells are GFP-positive (Figure 12), these results indicate that a subpopulation of FEZF2-positive PT neurons may express both Nav1.1 and Nav1.2.

We additionally performed triple immunostaining of FEZF2, GFP, and Nav1.2 on *Scn1a*-GFP mice at P15 (Supplementary figure S10 and Supplementary table S32), showing that in L5 74% of FEZF2-positive cells are GFP-positive but a majority of their AISs are Nav1.2-negative. The ratios of Nav1.2-positive cells among FEZF2-positive cells obtained in the triple-immunostaining of FEZF2, GFP and Nav1.2 (Supplementary figure S10) are 27% (L5) and 40% (L6). These results further support the above notion that a subpopulation of FEZF2-positive PT neurons may express both Nav1.1 and Nav1.2. Although further studies such retrograde tracking analyses are required to confirm and figure out the detailed circuits, all these results propose that the majority of L5 PT neurons express Nav1.1.”

Lines #342-353 at the Results section;

"To investigate whether TBR1-positive cells express Nav1.2, we performed triple immunostaining of TBR1, Nav1.2 and ankyrinG on *Scn1a*-GFP mice (Supplementary figure S12 and Supplementary tables S40-S43). In L5, contrary to the low ratios of GFP-positive cells among TBR1-positive cells (11% at P15 and 5% at 4W) (Figure 14B-middle panels and Supplementary table S34), the ratios of Nav1.2-positive cells among TBR1-positeve cells are high (69% (L2/3), 69%(L5) and 69% (L6) at P15) (Supplementary figure S12B-right-upper panel and Supplementary table S42). The ratios of TBR1-positive cells among Nav1.2-positive cells are 29% (L2/3), 53% (L5) and 62% (L6) at P15 (Supplementary figure S12B-middle upper panel and Supplementary table S41).

We further performed triple immunostaining for TBR1, Nav1.2 and GFP on *Scn1a*-GFP mice (Supplementary figure S13 and Supplementary table S44) and found that most (88%) of L6 TBR1-positive cells are GFP-negative.”

I would suggest doing a double labelling of PV and SST in Supplemental Figure 1 to show that all GFP cells in the hippocampus are inhibitory.

In response to the reviewer’s comment, we newly performed triple-staining of GFP, PV and SST and counted cell numbers (Figure 8). In addition, to investigate the ratio of inhibitory neurons in GFP-positive cells or vice versa more accurately we further generated and examined *Scn1a*-GFP and vesicular GABA transporter (*Vgat*)-Cre double transgenic mice in which *Vgat*-Cre is expressed in all GABAergic inhibitory neurons (Figure 7 and Supplementary tables S9 and S10). These experiments surely verified that all GFP cells in the hippocampus were inhibitory.

Accordingly, we revised the manuscript as follows;

Lines #198-235 at the Results section;

“To investigate the ratio of inhibitory neurons in GFP-positive cells, we generated and examined *Scn1a*-GFP and vesicular GABA transporter (*Vgat*)-Cre (Ogiwara et al., 2013) double transgenic mice in which *Vgat*-Cre is expressed in all GABAergic inhibitory neurons and visualized by floxed tdTomato transgene (Figure 7). In the neocortex at 4W, 23% (L2/3), 28% (L5) and 27% (L6) of GFP-positive cells were Tomato-positive inhibitory neurons and 73% (L2/3), 77% (L5) and 83% (L6) of Tomato-positive cells were GFP-positive (Figure 7C and Supplementary tables S9, S10). These results suggest that a significant subpopulation of neocortical excitatory neurons also express Nav1.1. Our previous observation that Nav1.1 is expressed in callosal axons of neocortical excitatory neurons (Ogiwara et al., 2013) supports that a subpopulation of L2/3 CC neurons express Nav1.1. Unlike in neocortex, in the hippocampus most of GFP-positive cells were Tomato-positive, 98% (CA1) and 94% (DG), and majorities of Tomato-positive GABAergic neurons are GFP-positive, 93% (CA1) and 77% (DG). These results further confirmed that in hippocampus Nav1.1 is expressed in inhibitory neurons but not in excitatory neurons. Although somata of pyramidal cells in CA2/3 region are weakly GFP-positive in this and some other experiments (Figure 7B, Supplementary figure S2G), those were GFP-negative in other experiments (Figures 2K, 5A, D) and therefore the Nav1.1 expression in CA2/3 pyramidal cells would be minimal if any.

We further performed immunohistochemical staining of PV and SST in neocortex and hippocampus of *Scn1a*-GFP mice at 4W (Figure 8). PV and SST do not co-express in cells and do not overlap. PV-INs and SST-INs were both GFP-positive, and especially GFP signals in PV-INs were intense (Figure 8A). Cell counting revealed that 21% (L2/3), 37% (L5), 37% (L6), 58% (CA1), 42% (CA2/3), and 41% (DG) of GFP-positive cells were PV or SST-positive depending on regions in neocortex and hippocampus (Figure 8B and Supplementary tables S11-S13). All PV-INs were GFP-positive (Figure 8B – middle), and most of SST-INs were GFP-positive (Figure 8B – right). Comparison of these results with those of *Vgat*-Cre mouse (Figure 7) suggests that GFP-positive GABAergic neurons in neocortex are mostly PV- or SST-positive, while in hippocampus a half of those are PV/SST-negative GABAergic neurons. Higher ratios of PV- or SST-positive cells (Figure 8B) compared with those of *Vgat*-Cre-positive cells (Figure 7C) among GFP-positive cells would be explained by that we counted PV-positive cells even if their PV-immunosignals are moderate and a significant subpopulation of such cells are known to be excitatory neurons (Jinno et al., 2004; Tanahira et al., 2009; Matho et al., 2021). Quantitative analysis of GFP signal intensity and area size of cells revealed that GFP signal intensities in PV-positive cells were significantly higher than those in PV-negative cells and GFP signal intensities in SST-positive cells were lower than those in PV-positive cells but similar to PV/SST-double negative cells (Figure 9 and Supplementary tables S14-S16). These results indicate that Nav1.1 expression level in PV-INs is significantly higher than those in excitatory neurons and PV-negative GABAergic neurons including SST-INs.”

Lines #520-534 at the Materials and methods section;

"Fluorescence and Immunofluorescence quantification

For quantification of inhibitory neurons in GFP-positive cells, we used *Scn1a*-GFP/*Vgat*-Cre/*Rosa26*-tdTomamto mice at 4W. We acquired multiple color images of primary motor cortex and hippocampus from three parasagittal sections per animal. Six images per region of interest were manually counted and summarized using Adobe Photoshop Elements 10 and Excel (Microsoft). On immunofluorescence quantification, we used *Scn1a*-GFP mice at P15 and/or 4W for the quantification of immunosignals. For quantification of GFP, NeuN, PV, SST, Nav1.1, Nav1.2, FEZF2 or TBR1-positive cells, we acquired multiple color images of primary motor cortex and hippocampus from three parasagittal sections per animal. Six~nine images per region of interest were manually quantified and summarized. For quantification of PV, SST, FEZF2 or TBR1-positive cells, intensity and area size of GFP fluorescent signals were measured by Fiji software. Statistical analyses were performed by one-way ANOVA followed by Tukey–Kramer post-hoc multiple comparison test using Kyplot 6.0 (KyensLab Inc). P-value smaller than 0.05 was considered statistically significant. Data are presented as the mean ± standard error of the mean (SEM)."

In Figure 1, it looks like the left arrowhead in the GFP panel of E is right on top of the neuron's AIS; unlike in the Merge panel where the neurons seems to be completely to the right of the arrowhead

The original Figure 1E has been replaced with new figures Figure 4B and C, in which the indicators were carefully arranged.

In figure 2, the stratum pyramidale in CA1 and CA3 od panel C seems to have a lot of background and it is very hard to see the truly positive cells

In response to the reviewer’s comment, we adjusted the contrast and brightness of the images in Figure 2C to clarify the immunostaining of GFP.

In Figure 3, because the GFP is mostly somatic and Nav1.2/ankyrin G are located in the AIS, the current magnification of the figure makes it difficult to interpret. Consider using the same magnification as in Figure 1 panels E and F. With the current magnification it seems like one cell in the top left corner is positive for both GFP and Nav1.2

In response to the reviewer’s comment, we have replaced the original Figure 3 with a new Figure 11 of higher magnification and counted cell numbers.

Accordingly, we revised the manuscript as follows;

Lines #251-262 at the Results section;

“As mentioned, GFP signals in *Scn1a*-GFP mouse can represent even moderate or low Nav1.1 expressions which cannot be detected by immunohistochemical staining, so some of GFP-positive cells may still express Nav1.2. To investigate whether and if so how much of GFP-positive cells have Nav1.2-positive AISs in *Scn1a*-GFP mouse neocortex, we performed triple immunohistochemical staining for Nav1.2, GFP and ankyrinG (Figure 11 and Supplementary tables S18-S20). The staining showed that AISs of GFP-positive cells are largely negative for Nav1.2, and cells with Nav1.2-positive AISs are mostly GFP-negative (Figure 11A). Cell counting revealed that 88% (L2/3), 90% (L5) and 95% (L6) of cells with Nav1.2-positive AISs at P15 were GFP-negative (Figure 11B-middle panel and Supplementary table S19), and 69% (L2/3), 83% (L5), and 86% (L6) of AISs of GFP-positive cells at P15 were Nav1.2-negative (Figure 11B-right panel and Supplementary table S20). These results indicate that the co-expression of GFP and Nav1.2 would be minimal if any.”

Lines #520-534 at the Materials and methods section;

"Fluorescence and Immunofluorescence quantification

For quantification of inhibitory neurons in GFP-positive cells, we used *Scn1a*-GFP/*Vgat*-Cre/*Rosa26*-tdTomamto mice at 4W. We acquired multiple color images of primary motor cortex and hippocampus from three parasagittal sections per animal. Six images per region of interest were manually counted and summarized using Adobe Photoshop Elements 10 and Excel (Microsoft). On immunofluorescence quantification, we used *Scn1a*-GFP mice at P15 and/or 4W for the quantification of immunosignals. For quantification of GFP, NeuN, PV, SST, Nav1.1, Nav1.2, FEZF2 or TBR1-positive cells, we acquired multiple color images of primary motor cortex and hippocampus from three parasagittal sections per animal. Six~nine images per region of interest were manually quantified and summarized. For quantification of PV, SST, FEZF2 or TBR1-positive cells, intensity and area size of GFP fluorescent signals were measured by Fiji software. Statistical analyses were performed by one-way ANOVA followed by Tukey–Kramer post-hoc multiple comparison test using Kyplot 6.0 (KyensLab Inc). P-value smaller than 0.05 was considered statistically significant. Data are presented as the mean ± standard error of the mean (SEM)."

In the legend of Figure 5 for consistency keep the order of labeling consistent when describing the arrows and asterisks (i.e. mention the GFP status first in all of them). In the description of the asterisks it mentions CFP/TBR1-double positive cells. That CFP should be GFP.

In this revision, we have moved the original Figure 5 to a new Figure 14, and we edited/corrected the legend.

We revised the manuscript as follows;

Lines #934-951 at the Figure legend section:

“Figure 14. GFP-positive cells were mostly negative for TBR1 at L5 of *Scn1a*-GFP mouse neocortex.

(A) Double immunostaining of TBR1 and GFP in neocortex of 4W *Scn1a*-GFP mouse (line #233) detected by mouse rabbit anti-TBR1 (magenta) and anti-GFP (green) antibodies. Arrows indicate TBR1-negative/GFP-positive cells. Magnified images outlined in (A) are shown in (A1-A6). Note that at L5 GFP-positive cells were mostly TBR1-negative but at L2/3 more than half of GFP-positive cells were TBR1-positive. Scale bars; 100 μm (A), 50 μm (A1-A6). (B) Bar graphs indicating the percentage of cells with TBR1- and GFP-positive/negative nuclei and somata per GFP-positive cells and/or TBR1-positive nuclei (left panels) (see also Supplementary table S33), the percentage of cells with GFP-positive/negative somata per cells with TBR1-positive nuclei (middle panels) (see also Supplementary table S34), and the percentage of cells with TBR1-positive/negative nuclei per cells with GFP-positive somata (right panels) (see also Supplementary table S35) in L2/3, L5, and L6. Cells in primary motor cortex of *Scn1a*-GFP mouse at P15 and 4W were counted. Note that 86% (P15) and 95% (4W) of cells with TBR1-positive cells are GFP-negative in L5 (middle panels), and 90% (P15) and 96% (4W) of cells with GFP-positive cells are TBR1-negative in L5 (right panel). L2/3, L5, L6: neocortical layer II/III, V, VI. +, positive; -, negative.”

In Supplemental Table S1 please add the conventions for the a, b, and c superscripts. For consistency remove the line between P15 and 4W in the L2/3 rows and consider adding a line between the L5 and L6 rows.

In the revised manuscript, the original Supplementary table S1 has been split into 4 Supplementary tables S22, S23, S34 and S35. As suggested, we have edited the conventions for the superscripts a, b, and c in rows of each Supplementary table. We have also added lines in the tables appropriately.

There are occasional grammar and misspelling issues throughout the manuscript.

As suggested, we have intensively revised the manuscript.